# SGD WITH MEMORY: FUNDAMENTAL PROPERTIES AND STOCHASTIC ACCELERATION

**Dmitry Yarotsky**
Skoltech,
Steklov Mathematical Institute
d.yarotsky@skoltech.ru

**Maksim Velikanov**
Technology Innovation Institute,
CMAP, Ecole Polytechnique
maksim.velikanov@tii.ae

## ABSTRACT

An important open problem is the theoretically feasible acceleration of mini-batch SGD-type algorithms on quadratic problems with power-law spectrum. In the non-stochastic setting, the optimal exponent $\xi$ in the loss convergence $L_t \sim C_L t^{-\xi}$ is double that in plain GD and is achievable using Heavy Ball (HB) with a suitable schedule; this no longer works in the presence of mini-batch noise. We address this challenge by considering first-order methods with an arbitrary fixed number $M$ of auxiliary velocity vectors (*memory-$M$ algorithms*). We first prove an equivalence between two forms of such algorithms and describe them in terms of suitable characteristic polynomials. Then we develop a general expansion of the loss in terms of *signal and noise propagators*. Using it, we show that losses of stationary stable memory-$M$ algorithms always retain the exponent $\xi$ of plain GD, but can have different constants $C_L$ depending on their *effective learning rate* that generalizes that of HB. We prove that in memory-1 algorithms we can make $C_L$ arbitrarily small while maintaining stability. As a consequence, we propose a memory-1 algorithm with a time-dependent schedule that we show heuristically and experimentally to improve the exponent $\xi$ of plain SGD.

## 1 INTRODUCTION

Optimization is a central component of deep learning, with accelerated training algorithms providing both better final results and reducing the large amounts of compute used to train modern large models. The key aspects of this optimization is a huge number of degrees of freedom and the batch-wise learning process. The theoretical understanding of even simple gradient-based algorithms such as Stochastic Gradient Descent (SGD) is currently limited, and their acceleration challenging.

Deep neural networks can be approximated with linear models when close to convergence or in wide network regimes (Lee et al., 2019; Fort et al., 2020). For regression problems with MSE loss, this leads to quadratic optimization problem. The resulting spectral distributions – the eigenvalues $\lambda_k$ of the Hessian and the coefficients $c_k$ of the expansion of the optimal solution $\mathbf{w}_*$ over the Hessian eigenvectors – are often well described by power-laws when numerically computed on real-world data (Cui et al. (2021); Bahri et al. (2021); Wei et al. (2022), see more details and related work in Sec. A). With this motivation, we focus on infinite-dimensional quadratic problems, and especially (but not exclusively) those with a power-law asymptotic form of eigenvalues $\lambda_k$ and coefficients $c_k$:

$$\lambda_k = \Lambda k^{-\nu}(1 + o(1)), \quad k \to \infty; \qquad \sum_{k:\lambda_k < \lambda} \lambda_k c_k^2 = Q\lambda^\zeta(1 + o(1)), \quad \lambda \to 0. \qquad (1)$$

These power laws can be thought of as stricter versions of the classical capacity and source conditions (Caponnetto & De Vito, 2007). Convergence rates of different first-order algorithms are determined by the eigenvalue and target exponents $\nu, \zeta > 0$.

Theoretically more tractable are *stationary* algorithms, i.e. with iteration-independent parameters. In the non-stochastic setting, both stationary GD and Heavy Ball (HB) exhibit the same loss rate $O(t^{-\zeta})$ (Nemirovskiy & Polyak, 1984). In the presence of mini-batch noise, analysis of convergence becomes more challenging, with only the recent works Berthier et al. (2020); Varre et al. (2021) showing that the rate of stationary GD remains $O(t^{-\zeta})$ for sufficiently hard targets with $\zeta < 2 - \frac{1}{\nu}$,

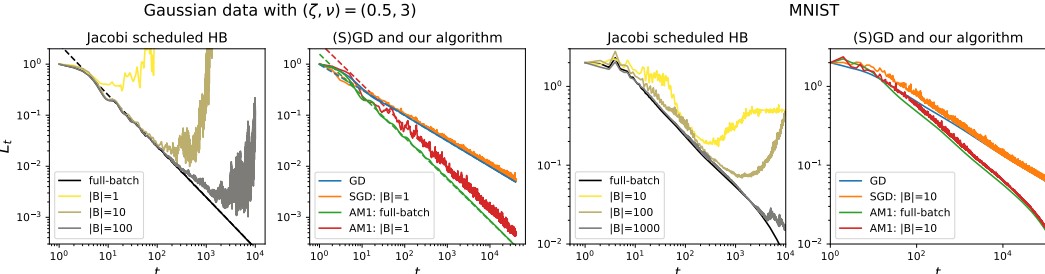

Figure 1: *Divergence of Jacobi accelerated HB vs. stability of our accelerated memory-1 (AM1) method.* For both synthetic Gaussian data **(left)** and MNIST classification with shallow ReLU network **(right)**, Jacobi HB enjoys the accelerated rate $L_t = O(t^{-2\zeta})$ in full-batch setting, but eventually diverges in stochastic one. Increasing batch size only delays the loss explosion. In contrast, our AM1 algorithm, although having weaker acceleration than full-batch Jacobi HB, is stable for small batch sizes and a long training duration. For Gaussian data with an ideal power-law spectrum, dashed lines show theoretical loss power laws with predicted exponents ($t^{-\zeta}$ for SGD and $t^{-\zeta(1+\overline{\alpha})}$ for AM1) and asymptotically match the experimental loss trajectories. Here $\overline{\alpha} > 0$ is the schedule exponent of the effective learning rate $\alpha_{\text{eff},t} \approx t^{\overline{\alpha}}$ used by AM1 algorithm. See precise definition of the AM1 schedule and parameters in sec. 6. Also, see sec. M for figure details and extra experiments.

while slowing down by noise to $O(t^{-(2-\frac{1}{\nu})})$ for easier targets with $\zeta > 2 - \frac{1}{\nu}$. In the rest of the paper, we will refer to these two regimes as *signal-* and *noise-dominated*, respectively.

With non-stationary algorithms, the rate in non-stochastic setting can be accelerated to $O(t^{-2\zeta})$ using HB with a schedule of learning rate and momentum based on Jacobi polynomials (Brakhage, 1987). However, in the stochastic mini-batch setting we are not aware of similar acceleration results for general $\zeta, \nu$ in the literature. Moreover, in Fig. 1 we show that direct application of HB with Jacobi schedule to noisy problems consistently leads to divergence of the optimization trajectory.

Previous works point to a connection between the observed divergence of Jacobi scheduled HB and a weaker (constant-level) acceleration of the stationary algorithms. Under power law spectrum (1), stationary Heavy Ball in both non-stochastic (Velikanov & Yarotsky, 2024) and signal-dominated stochastic (Velikanov et al., 2023) settings has the loss asymptotic $L_t \sim C_L t^{-\zeta}$. While the exponent $\zeta$ is fixed, the constant $C_L$ can vary greatly with learning rate $\alpha$ and momentum $\beta$ parameters of HB as $C_L \propto (\alpha_{\text{eff}})^{-\zeta}$, where $\alpha_{\text{eff}} = \frac{\alpha}{1-\beta}$ is the *effective learning rate*. On the one hand, non-stochastic setting allows to take $\beta \to 1$ without destroying the stability of the algorithm, making the constant $C_L$ arbitrarily small. In a sense, this mechanism is behind the Jacobi scheduled HB with the accelerated rate $O(t^{-2\zeta})$ which slowly takes $\beta_t \to 1$ along the optimization trajectory. On the other hand, Velikanov et al. (2023) shows that maintaining stability under a mini-batch sampling noise requires $\alpha_{\text{eff}}$ to be bounded, thus putting a lower bound on the loss constant $C_L$.

**Our contributions.** In this work, we overcome the above limitation by developing a general framework to analyze loss trajectories of a broad family of first-order methods. Then, we identify a region in the space of update rules that achieves acceleration while being stable. Specifically:

1. **Memory-$M$ algorithms.** We define a broad family of first-order algorithms that allow to store an arbitrary fixed amount $M$ of auxiliary vectors (i.e. generalized velocities), and use any linear combinations of them in gradient computation and parameter update. First, we connect these memory-$M$ algorithms with other multistep first-order methods. Then, in the mini-batch setting, we provide a unified description of their loss trajectories in terms of expansion in various combinations of elementary signal and noise *propagators*.

2. **Combinatorial approach to loss asymptotics.** We use the obtained propagator expansion to enrich in several directions the phase diagram of mini-batch SGD converges rates, for stationary algorithms. First, we show that the same signal-dominated $O(t^{-\zeta})$ and noise-dominated $O(t^{-(2-\frac{1}{\nu})})$ rates remain valid for any memory-$M$ algorithm. Second, we give a simple interpretation of the two phases in terms of our propagator expansion: the loss in

the signal- (resp, noise-) dominated phase is asymptotically mostly determined by configurations including one long signal (resp, noise) propagator. Finally, we generalize *effective learning rate* $\alpha_{\text{eff}}$ from HB to all memory-$M$ algorithms, and show that it largely determines the constant $C_L$ in the loss asymptotic.

3. **Accelerated update rule.** We then focus on general memory-1 algorithms. First, we find explicit expressions for loss asymptotic and full stability conditions of general stationary memory-1 algorithms. Then, in this space of stable algorithms, we find a region with arbitrary large effective learning rate $\alpha_{\text{eff}}$, which translates into arbitrary small constant $C_L$ in the loss asymptotic. The update rule in this region can be obtained by adding an extra learning rate to HB. Finally, we design a schedule that travels deep within the region in a power-law manner, and demonstrate that it achieves accelerated convergence rates up to $O(t^{-\zeta(2-\frac{1}{\nu})})$ in the signal-dominated phase.

## 2 THE SETTING

We consider the regression problem of fitting a target function $y(\mathbf{x})$ with a linear model $f(\mathbf{w}, \mathbf{x}) = \langle \mathbf{x}, \mathbf{w} \rangle$. Inputs $\mathbf{x}$ and parameters $\mathbf{w}$ are elements of a (infinite-dimensional) Hilbert space $\mathcal{H}$, and $\langle \cdot, \cdot \rangle$ denotes the scalar product in $\mathcal{H}$. Inputs $\mathbf{x}$ are drawn from a distribution $\rho$ on $\mathcal{H}$, and we measure the loss of the model as $L(\mathbf{w}) = \frac{1}{2} \mathbb{E}_{\mathbf{x} \sim \rho}(\langle \mathbf{x}, \mathbf{w} \rangle - y(\mathbf{x}))^2$. Importantly, we assume that the target function is representable in $\mathcal{H}$ as $y(\mathbf{x}) = \langle \mathbf{x}, \mathbf{w}_* \rangle$, and require only the norm $E_{\mathbf{x} \sim \rho}[y(\mathbf{x})^2]$ of the target but not of the optimal parameters $\mathbf{w}_*$ to be finite. This setting is relevant to kernel methods or overparametrized neural networks, with $\mathbf{x}$ being the mapping of original inputs to either kernel feature space or gradients of the model, respectively; see Sec. A for more details. This brings us to the quadratic objective

$$L(\mathbf{w}) = \frac{1}{2}\langle \Delta \mathbf{w}, \mathbf{H} \Delta \mathbf{w} \rangle, \qquad \mathbf{H} = \mathbb{E}_{\mathbf{x} \sim \rho}[\mathbf{x} \otimes \mathbf{x}], \tag{2}$$

where $\mathbf{H}$ is the model's Hessian and $\Delta \mathbf{w} = \mathbf{w} - \mathbf{w}_*$.

We will always assume that the Hessian $\mathbf{H}$ has a discrete spectrum with eigenvalues $\lambda_k \searrow 0, \lambda_k \neq 0$. We will also often (but not always) assume that the eigenvalues $\lambda_k$ and the respective eigenexpansion coefficients $c_k = \langle \mathbf{e}_k, \mathbf{w}_* \rangle$ of $\mathbf{w}_*$ obey power laws (1) with some $\nu, \zeta > 0$. We allow the "infeasible" case $0 < \zeta < 1$ when $\langle \mathbf{w}_*, \mathbf{H} \mathbf{w}_* \rangle < \infty$ but $\|\mathbf{w}_*\|^2 = \langle \mathbf{w}_*, \mathbf{w}_* \rangle = \infty$.

For stochastic optimization, we consider the random mini-batch setting. At each iteration $t$ we choose a batch $B = \{\mathbf{x}_i\}_{i=1}^b$ of $|B|$ i.i.d. samples from $\rho$, and compute the gradient on the respective empirical loss $L_B(\mathbf{w}) = \frac{1}{2|B|} \sum_{i=1}^{|B|} (\langle \mathbf{x}_i, \Delta \mathbf{w} \rangle)^2$. We focus on the evolution of the mean loss $L_t \equiv \mathbb{E}[L(\mathbf{w}_t)]$ averaged w.r.t. the choice of all the batches $B_1, \ldots, B_t$ along the optimization trajectory.

## 3 GRADIENT DESCENT WITH MEMORY

**General form of the algorithm.** We start with describing our gradient descent with memory in the non-stochastic setting. While later we will specialize to quadratic objectives $L$ as in Eq. (2), GD with memory can be defined as the first-order minimization method applicable to any loss $L$:

$$\begin{pmatrix} \mathbf{w}_{t+1} - \mathbf{w}_t \\ \mathbf{u}_{t+1} \end{pmatrix} = \begin{pmatrix} -\alpha_t & \mathbf{b}_t^T \\ \mathbf{c}_t & D_t \end{pmatrix} \begin{pmatrix} \nabla L(\mathbf{w}_t + \mathbf{a}_t^T \mathbf{u}_t) \\ \mathbf{u}_t \end{pmatrix}, \quad t = 0, 1, 2, \ldots \tag{3}$$

Here, $\mathbf{w}_t$ is the step-$t$ approximation to an optimal vector $\mathbf{w}_*$, and $\mathbf{u}_t$ is an auxiliary vector representing the "memory" of the optimizer. We assume that these auxiliary vectors have the form $\mathbf{u} = (\mathbf{u}^{(1)}, \ldots \mathbf{u}^{(M)})^T$ with $\mathbf{u}^{(m)} \in \mathcal{H}$; in other words, $\mathbf{u}_t$ is a size-$M$ column with each component belonging to the same space $\mathcal{H}$ as the vector $\mathbf{w}_t$. We will refer to $M$ as the *memory size*. The parameters $\alpha_t$ (learning rates) are scalar, the parameters $\mathbf{a}_t, \mathbf{b}_t, \mathbf{c}_t$ are $M$-dimensional column vectors, and $D_t$ are $M \times M$ scalar matrices. By $\mathbf{a}_t^T \mathbf{u}_t$ we mean $\sum_{m=1}^M a_t^{(m)} \mathbf{u}^{(m)}$, where $a_t^{(m)}$ are the scalar components of the vector $\mathbf{a}_t$. The algorithm can be viewed as a sequence of transformations of size-$(M+1)$ column vectors $\begin{pmatrix} \mathbf{w}_t \\ \mathbf{u}_t \end{pmatrix}$ with $\mathcal{H}$-valued components.

Note a few important special cases:

1. The basic gradient descent (GD) $\mathbf{w}_{t+1} = \mathbf{w}_t - \alpha_t \nabla L(\mathbf{w}_t)$ is the memory-0 case.

2. The Heavy Ball (a.k.a. GD with momentum, Polyak (1964)) can be defined by the two-step recurrence $\mathbf{w}_{t+1} = \mathbf{w}_t - \alpha_t \nabla L(\mathbf{w}_t) + \beta_t(\mathbf{w}_t - \mathbf{w}_{t-1})$. It can be cast in the form (3) with memory $M = 1$ by introducing the auxiliary vector $\mathbf{u}_t = \mathbf{w}_t - \mathbf{w}_{t-1}$:

$$\begin{pmatrix} \mathbf{w}_{t+1} - \mathbf{w}_t \\ \mathbf{u}_{t+1} \end{pmatrix} = \begin{pmatrix} -\alpha_t & \beta_t \\ -\alpha_t & \beta_t \end{pmatrix} \begin{pmatrix} \nabla L(\mathbf{w}_t) \\ \mathbf{u}_t \end{pmatrix}. \tag{4}$$

3. Averaged GD (Polyak & Juditsky, 1992) can be defined by $\mathbf{w}_t = \frac{1}{t} \sum_{s=1}^t \widetilde{\mathbf{w}}_s$, where $\widetilde{\mathbf{w}}_{t+1} = \widetilde{\mathbf{w}}_t - \alpha_t \nabla L(\widetilde{\mathbf{w}}_t)$ (i.e., the original GD trajectory $\widetilde{\mathbf{w}}_t$ is subsequently averaged, usually with the purpose of noise suppression). It can be cast in the form (3) with memory $M = 1$ by introducing the auxiliary vector $\mathbf{u}_t = \widetilde{\mathbf{w}}_t - \mathbf{w}_t$:

$$\begin{pmatrix} \mathbf{w}_{t+1} - \mathbf{w}_t \\ \mathbf{u}_{t+1} \end{pmatrix} = \begin{pmatrix} -\frac{\alpha_t}{t+1} & \frac{1}{t+1} \\ -\frac{\alpha_t t}{t+1} & \frac{t}{t+1} \end{pmatrix} \begin{pmatrix} \nabla L(\mathbf{w}_t + \mathbf{u}_t) \\ \mathbf{u}_t \end{pmatrix}. \tag{5}$$

Flammarion & Bach (2015) consider a wider family of memory-1 algorithms that also includes Nesterov's accelerated gradient (Nesterov, 1983). Some algorithms in the literature use memory size larger than 1 (Varre & Flammarion, 2022).

Iterations (3) thus represent a general framework encompassing various standard algorithms. The first entry in the l.h.s. is written as $\mathbf{w}_{t+1} - \mathbf{w}_t$ because it is natural to have all matrix elements in Eq. (3) *shift-invariant* while the sequence $\mathbf{w}_t$ is naturally *shift-equivariant* (see B.1).

Clearly, there is some redundancy in the definition of the algorithms (3). In particular, any linear transformation $\mathbf{u}'_t = C_t \mathbf{u}_t$ of the auxiliary vectors with some non-degenerate $M \times M$ matrices $C_t$ produces an equivalent algorithm, having new parameters $\mathbf{a}'_t, \mathbf{b}'_t, \mathbf{c}'_t, D'_t$, but preserving the trajectory $\mathbf{w}_t$. A large part of our work will be focused on the stationary setting, i.e. when $\alpha, \mathbf{a}, \mathbf{b}, \mathbf{c}, D$ are $t$-independent. In this setting we will be able to largely remove redundancy by going to an equivalent form of the dynamics.

**Quadratic losses.**   We assume now a quadratic objective $L$ as in Eq. (2). In this case $\nabla L(\mathbf{w}) = \mathbf{H}\Delta\mathbf{w}$ and iterations (3) acquire the form

$$\begin{pmatrix} \Delta\mathbf{w}_{t+1} \\ \mathbf{u}_{t+1} \end{pmatrix} = \left[ \begin{pmatrix} 1 & \mathbf{b}_t^T \\ 0 & D_t \end{pmatrix} + \mathbf{H} \otimes \begin{pmatrix} -\alpha_t \\ \mathbf{c}_t \end{pmatrix} (1, \mathbf{a}_t^T) \right] \begin{pmatrix} \Delta\mathbf{w}_t \\ \mathbf{u}_t \end{pmatrix}. \tag{6}$$

Here, if $\begin{pmatrix} \Delta\mathbf{w}_t \\ \mathbf{u}_t \end{pmatrix}$ is viewed as an element of the tensor product $\mathcal{H} \otimes \mathbb{R}^{M+1}$, the operator $\mathbf{H}$ acts in the first factor, and the rank-1 matrix $\begin{pmatrix} -\alpha_t \\ \mathbf{c}_t \end{pmatrix}(1\ \mathbf{a}_t^T)$ in the second. In particular, if $\Delta\mathbf{w}_t$ and $\mathbf{u}_t$ are eigenvectors of $\mathbf{H}$ with eigenvalue $\lambda$, then

$$\begin{pmatrix} \Delta\mathbf{w}_{t+1} \\ \mathbf{u}_{t+1} \end{pmatrix} = S_{\lambda,t} \begin{pmatrix} \Delta\mathbf{w}_t \\ \mathbf{u}_t \end{pmatrix}, \quad S_{\lambda,t} = \left[ \begin{pmatrix} 1 & \mathbf{b}_t^T \\ 0 & D_t \end{pmatrix} + \lambda \begin{pmatrix} -\alpha_t \\ \mathbf{c}_t \end{pmatrix} (1, \mathbf{a}_t^T) \right], \tag{7}$$

and the new vectors $\Delta\mathbf{w}_{t+1}, \mathbf{u}_{t+1}$ are again eigenvectors of $\mathbf{H}$ with eigenvalue $\lambda$. As a result, performing the spectral decomposition of $\Delta\mathbf{w}_t, \mathbf{u}_t$ reduces the original dynamics (3) acting in $H \otimes \mathbb{R}^{M+1}$ to a $\lambda$-indexed collection of independent dynamics each acting in $\mathbb{R}^{M+1}$.

**Equivalence with multistep recurrences.**   As noted earlier, Heavy Ball can be alternatively written either in the single-step matrix form (4) with auxiliary vectors $\mathbf{u}_t$, or as the two-step recurrence $\mathbf{w}_{t+1} = \mathbf{w}_t - \alpha_t \nabla L(\mathbf{w}_t) + \beta_t(\mathbf{w}_t - \mathbf{w}_{t-1})$ not involving auxiliary vectors, but involving two previous vectors $\mathbf{w}_t, \mathbf{w}_{t-1}$. A natural question is whether a similar equivalence holds for general algorithms (3). It appears that there is no such equivalence for a general loss function $L$. However, if $L$ is quadratic as in Eq. (2) and, moreover, the parameters $\alpha, \mathbf{a}, \mathbf{b}, \mathbf{c}, D$ are $t$-independent, we can prove an equivalence of (3) to multistep iterations

$$\mathbf{w}_{t+M+1} = \sum_{m=0}^M p_m \mathbf{w}_{t+m} + \sum_{m=0}^M q_m \nabla L(\mathbf{w}_{t+m}), \quad t = 0, 1, \ldots, \tag{8}$$

where $\sum_{m=0}^M p_m = 1$. Such iterations can be viewed as general stationary shift-invariant algorithms linearly depending on 0'th and 1'st order information from previous $M$ steps.

**Theorem 1** (B.2). *Suppose that the loss is quadratic as in Eq.* (2).

1. *Given any $t$-independent parameters $\alpha \in \mathbb{R}, \mathbf{a}, \mathbf{b}, \mathbf{c} \in \mathbb{R}^M, D \in \mathbb{R}^{M \times M}$, define constants $(p_m)_{m=0}^M, (q_m)_{m=0}^M$ to be the coefficients in the polynomial expansions*

$$\sum_{m=0}^{M} p_m x^m = x^{M+1} - (x-1)\det(x-D), \tag{9}$$

$$\sum_{m=0}^{M} q_m x^m = \det \begin{pmatrix} \mathbf{a}^T \mathbf{c} - \alpha & \mathbf{a}^T(1-D) - \mathbf{b}^T \\ \mathbf{c} & x - D \end{pmatrix}. \tag{10}$$

*Then $\sum_{m=0}^M p_m = 1$ and any solution $\left(\begin{smallmatrix} \mathbf{w}_t \\ \mathbf{u}_t \end{smallmatrix}\right)$ of matrix dynamics (3) obeys recurrence (8).*

2. *Conversely, given constants $(p_m)_{m=0}^M, (q_m)_{m=0}^M$ such that $\sum_{m=0}^M p_m = 1$, there exist parameters $\alpha \in \mathbb{R}, \mathbf{a}, \mathbf{b}, \mathbf{c} \in \mathbb{R}^M, D \in \mathbb{R}^{M \times M}$ such that any vector sequence $\mathbf{w}_t$ generated by recurrence (8) satisfies the respective $t$-independent memory-$M$ matrix dynamics (3) with suitable auxiliary vectors $\mathbf{u}_t$. Moreover, we can set $\mathbf{a} = \mathbf{0}$ and $\mathbf{b} = (1, 0, \ldots, 0)^T$.*

Theorem 1 along with its proof provide a constructive correspondence between the two forms of iterations. The fact that we can set $\mathbf{a} = 0$ when converting recurrence (8) to matrix iterations (3) (statement 2) shows that for quadratic losses and stationary dynamics the possibility of shifting the argument in $\nabla L$ with $\mathbf{a} \neq 0$ does not increase the expressiveness of the algorithm.

The natural non-stationary analog of Theorem 1 fails to hold in general:

**Proposition 1** (B.3). *There exist a non-stationary dynamics (3) with $M = 1$ that cannot be written in the form $\mathbf{w}_{t+2} = p_{t,1}\mathbf{w}_{t+1} + p_{t,0}\mathbf{w}_t + q_{t,1}\nabla L(\mathbf{w}_{t+1}) + q_{t,0}\nabla L(\mathbf{w}_t)$, even for quadratic problems.*

## 4 BATCH SGD WITH MEMORY

**Evolution of second moments.** SGD with memory is defined similarly to GD with memory (3), but with the batch loss $L_B$ instead of $L$.

For quadratic loss functions, we saw the deterministic GD with memory reduced to a simple evolution (6) of the state $\left(\begin{smallmatrix} \Delta \mathbf{w}_t \\ \mathbf{u}_t \end{smallmatrix}\right)$. Such a description is no longer feasible for SGD, but we can describe deterministically the evolution of the second moments $\mathbf{M}_t$ of the state $\left(\begin{smallmatrix} \Delta \mathbf{w}_t \\ \mathbf{u}_t \end{smallmatrix}\right)$

$$\mathbf{M}_t \equiv \begin{pmatrix} \mathbf{C}_t & \mathbf{J}_t \\ \mathbf{J}_t^T & \mathbf{V}_t \end{pmatrix}, \quad \mathbf{C}_t \equiv \mathbb{E}[\Delta \mathbf{w}_t \otimes \Delta \mathbf{w}_t], \quad \mathbf{J}_t \equiv \mathbb{E}[\Delta \mathbf{w}_t \otimes \mathbf{u}_t^T], \quad \mathbf{V}_t \equiv \mathbb{E}[\mathbf{u}_t \otimes \mathbf{u}_t^T]. \tag{11}$$

The moments $\mathbf{C}_t$ determine the loss as $L_t = \frac{1}{2}\text{Tr}(\mathbf{H}\mathbf{C}_t)$. A direct computation (see sec. C.1) yields

$$\mathbf{M}_{t+1} = \left[ \begin{pmatrix} 1 & \mathbf{b}_t^T \\ 0 & D_t \end{pmatrix} + \mathbf{H} \begin{pmatrix} -\alpha_t \\ \mathbf{c}_t \end{pmatrix} (1, \mathbf{a}_t^T) \right] \mathbf{M}_t \left[ \begin{pmatrix} 1 & \mathbf{b}_t^T \\ 0 & D_t \end{pmatrix} + \mathbf{H} \begin{pmatrix} -\alpha_t \\ \mathbf{c}_t \end{pmatrix} (1, \mathbf{a}_t^T) \right]^T \tag{12}$$

$$+ \frac{1}{|B|} \Sigma_\rho \left( (1, \mathbf{a}_t^T) \mathbf{M}_t (1, \mathbf{a}_t^T)^T \right) \otimes \begin{pmatrix} -\alpha_t \\ \mathbf{c}_t \end{pmatrix} (-\alpha_t \quad \mathbf{c}_t), \tag{13}$$

where

$$\Sigma_\rho(\mathbf{C}) = \mathbb{E}_{\mathbf{x} \sim \rho}[\langle \mathbf{x}, \mathbf{C}\mathbf{x} \rangle \mathbf{x} \otimes \mathbf{x}] - \mathbf{H}\mathbf{C}\mathbf{H}. \tag{14}$$

The term in line (12) corresponds to the noiseless dynamics, while the term in line (13) reflects the stochastic noise vanishing in the limit of batch size $|B| \to \infty$. The map $\Sigma_\rho(\mathbf{C})$ preserves the positive semi-definiteness of $\mathbf{C}$ (see Lemma 1), so the term in line (13) is also positive semi-definite, and $L_t$ is always not less than its counterpart for the deterministic GD with memory.

For general $\rho$, the first term in $\Sigma_\rho$ cannot be expressed via $\mathbf{H}$, thus preventing us from describing the loss evolution in terms of the spectral information $\lambda_k, c_k$. Following Velikanov et al. (2023), we overcome this by replacing $\Sigma_\rho$ with the *Spectrally Expressible (SE)* approximation

$$\Sigma_{\text{SE}}^{(\tau_1, \tau_2)}(\mathbf{C}) \equiv \tau_1 \text{Tr}[\mathbf{H}\mathbf{C}]\mathbf{H} - \tau_2 \mathbf{H}\mathbf{C}\mathbf{H}, \tag{15}$$

with some parameters $\tau_1, \tau_2$. We list basic properties of $\Sigma_{\text{SE}}^{(\tau_1, \tau_2)}$ in Lemma 1. In particular, $\Sigma_{\text{SE}}^{(\tau_1, \tau_2)}$ preserves semi-definiteness of $\mathbf{C}$ whenever $0 \leq \tau_1 \geq \tau_2$.

We can look at this replacement in two ways. First, Velikanov et al. (2023) show that, for certain choices of $\tau_1, \tau_2$, the equality $\Sigma_\rho = \Sigma_{\text{SE}}^{(\tau_1,\tau_2)}$ holds *exactly* in several natural special scenarios (gaussian and translation-invariant data) and *approximately* for a number of real-world problems such as MNIST classification. Second, generalizing earlier ideas of Bordelon & Pehlevan (2021) and "uniform kurtosis" assumptions (Varre & Flammarion, 2022), we show that the loss obtained with $\Sigma_{\text{SE}}^{(\tau_1,\tau_2)}$ can serve as an upper or lower bound for the loss obtained with $\Sigma_\rho$:

**Proposition 2** (C.2). *Denote by $L_t^{(\rho)}$ and $L_t^{(\tau_1,\tau_2)}$ the losses obtained using the same initial condition $\mathbf{M}_0 \geq 0$ and either the map $\Sigma_\rho$ or the map $\Sigma_{\text{SE}}^{(\tau_1,\tau_2)}$ in iterations* (13)*, respectively. Suppose that $0 \leq \tau_1 \geq \tau_2$. Then, if $\Sigma_\rho(\mathbf{C}) \leq \Sigma_{\text{SE}}^{(\tau_1,\tau_2)}(\mathbf{C})$ for all $\mathbf{C} \geq 0$, then $L_t^{(\rho)} \leq L_t^{(\tau_1,\tau_2)}$ for all $t$. Conversely, if $\Sigma_\rho(\mathbf{C}) \geq \Sigma_{\text{SE}}^{(\tau_1,\tau_2)}(\mathbf{C})$ for all $\mathbf{C} \geq 0$, then $L_t^{(\rho)} \geq L_t^{(\tau_1,\tau_2)}$ for all $t$.*

**Loss expansion.** The key role in the subsequent analysis is played by an expansion of $L_t$ in terms of scalar quantities that we call *propagators*. To simplify the exposition, we assume now and in the next section the stationary setting (see Sec. D for the non-stationary generalization). Consider the linear operators $A_\lambda : \mathbb{R}^{(1+M)\times(1+M)} \to \mathbb{R}^{(1+M)\times(1+M)}$ associated with the eigenvalues $\lambda$ of $\mathbf{H}$:

$$A_\lambda Z = S_\lambda Z S_\lambda^T - \tfrac{\tau_2}{|B|}\lambda^2 \left(\begin{smallmatrix}-\alpha\\\mathbf{c}\end{smallmatrix}\right)\left(\begin{smallmatrix}1\\\mathbf{a}\end{smallmatrix}\right)^T Z \left(\begin{smallmatrix}1\\\mathbf{a}\end{smallmatrix}\right)\left(\begin{smallmatrix}-\alpha\\\mathbf{c}\end{smallmatrix}\right)^T, \tag{16}$$

where $S_\lambda$ are the noiseless eigencomponent evolution matrices defined in Eq. (7). The operators $A_\lambda$ describe the $\mathbf{M}$ matrix iterations (12)–(13) in their diagonal eigencomponents w.r.t. $\mathbf{H}$. If the SE parameter $\tau_2 = 0$, then $A_\lambda$ simply implements the adjoint action of $S$ on matrices, $A_\lambda = S_\lambda \otimes S_\lambda$, but in general $A_\lambda$ also includes the effect of the $\tau_2 \mathbf{HCH}$ term in the SE noise (15).

Enumerate the eigenvalues of $\mathbf{H}$ as $\lambda_k$, and let $c_k^2 = \langle \mathbf{e}_k, \mathbf{w}_* \rangle^2$ be the respective amplitudes of the target $\mathbf{w}^*$. We define the *signal propagators* $V_t, V_t'$ by

$$V_t = \sum_k \lambda_k c_k^2 \langle \left(\begin{smallmatrix}1&0\\0&0\end{smallmatrix}\right), A_{\lambda_k}^{t-1}\left(\begin{smallmatrix}1&0\\0&0\end{smallmatrix}\right)\rangle, \quad V_t' = \sum_k \lambda_k c_k^2 \langle \left(\begin{smallmatrix}1\\\mathbf{a}\end{smallmatrix}\right)\left(\begin{smallmatrix}1\\\mathbf{a}\end{smallmatrix}\right)^T, A_{\lambda_k}^{t-1}\left(\begin{smallmatrix}1&0\\0&0\end{smallmatrix}\right)\rangle, \tag{17}$$

where $\langle A, B \rangle \equiv \text{Tr}(AB^*)$ is the standard inner product between matrices. Note that the difference between $V_t$ and $V_t'$ is only in taking this inner product either with $\left(\begin{smallmatrix}1&0\\0&0\end{smallmatrix}\right)$ or with$\left(\begin{smallmatrix}1\\\mathbf{a}\end{smallmatrix}\right)\left(\begin{smallmatrix}1\\\mathbf{a}\end{smallmatrix}\right)^T$.

We also define *noise propagators* $U_t, U_t'$ (with a similar difference between them) by

$$U_t = \tfrac{\tau_1}{|B|}\sum_k \lambda_k^2 \langle \left(\begin{smallmatrix}1&0\\0&0\end{smallmatrix}\right), A_{\lambda_k}^{t-1}\left(\begin{smallmatrix}-\alpha\\\mathbf{c}\end{smallmatrix}\right)\left(\begin{smallmatrix}-\alpha\\\mathbf{c}\end{smallmatrix}\right)^T\rangle, \quad U_t' = \tfrac{\tau_1}{|B|}\sum_k \lambda_k^2 \langle \left(\begin{smallmatrix}1\\\mathbf{a}\end{smallmatrix}\right)\left(\begin{smallmatrix}1\\\mathbf{a}\end{smallmatrix}\right)^T, A_{\lambda_k}^{t-1}\left(\begin{smallmatrix}-\alpha\\\mathbf{c}\end{smallmatrix}\right)\left(\begin{smallmatrix}-\alpha\\\mathbf{c}\end{smallmatrix}\right)^T\rangle. \tag{18}$$

**Theorem 2** (D). *Assuming $\mathbf{w}_0 = 0, \mathbf{u}_0 = 0$, for any $t = 0, 1, \ldots$*

$$L_t = \frac{1}{2}\Big(V_{t+1} + \sum_{m=1}^{t}\sum_{0<t_1<\ldots<t_m<t+1} U_{t+1-t_m}U'_{t_m-t_{m-1}}U'_{t_{m-1}-t_{m-2}}\cdots U'_{t_2-t_1}V'_{t_1}\Big). \tag{19}$$

This expansion results from considering all the configurations in which the iterations are divided into those where we apply the operators $A_\lambda$, and into remaining $t_1, \ldots, t_m$ where we apply the first term $\tau_1 \text{Tr}[\mathbf{HC}]\mathbf{H}$ of the SE noise (15). In the former iterations the states evolve independently in the eigenspaces of $\mathbf{H}$, whereas the latter act as rank-1 operators aggregating the states across all eigenspaces. Thanks to the rank-1 property, the loss of the configuration is factored as the product of scalar signal- and noise propagators that describe the eigenspace evolutions taken over iteration intervals $(t_k, t_{k+1})$ and aggregated at their ends. Each product includes one signal propagator associated with the evolution of the initial residual $\Delta\mathbf{w}_0 = -\mathbf{w}_*$, and some number of noise propagators associated with the evolution of the noise injected at $t_1, \ldots, t_m$. The propagators $U, V$ end at the loss evaluation time $t$, while the propagators $U', V'$ end at one of the noise injection times $t_1, \ldots, t_m$. Note that if $\mathbf{a} = 0$, then $U_t$ can be identified with $U_t'$ and $V_t$ with $V_t'$.

## 5 Stability and convergence of stationary memory-$M$ SGD

**Preliminaries.** We will now use loss expansion (19) to establish various general stability and convergence properties of stationary SGD with memory. It will be convenient to assume henceforth that all the propagators $U_t', U_t, V_t', V_t$ are nonnegative. A particular sufficient condition for this is $\tau_1 \geq 0, \tau_2 \leq 0$, since for $\tau_2 \leq 0$ the maps $A_\lambda$ preserve positive semi-definiteness (see Eq. (16)).

**Proposition 3.** *If $\tau_1 \geq 0$ and $\tau_2 \leq 0$, then all $U_t', U_t, V_t', V_t \geq 0$.*

**Immediate divergence.** Recall that $\dim \mathcal{H} = \infty$ and so the Hessian $\mathbf{H}$ has infinitely many eigenvalues. In this case, even single propagators $U'_t, U_t, V'_t, V_t$ may diverge. Note that $U'_1 = \frac{\tau_1}{|B|}(\mathbf{a}^T \mathbf{c} - \alpha)^2 \sum_k \lambda_k^2$ and $V'_1 = \sum_k \lambda_k c_k^2$. Accordingly, necessary conditions to avoid divergence in one step for generic $\mathbf{a}, \mathbf{c}, \alpha$ are $\sum_k \lambda_k^2 < \infty$ and $\sum_k \lambda_k c_k^2 < \infty$. These conditions are clearly also sufficient to ensure $U_t, U'_t, V_t, V'_t < \infty$ for all $t$. Under power law spectral assumptions (1) we automatically have $\sum_k \lambda_k c_k^2 < \infty$, while $\sum_k \lambda_k^2 < \infty$ is fulfilled whenever $\nu > \frac{1}{2}$.

**Convergence of the propagator expansion.** We analyze now loss expansion (19) and relate convergence of $L_t$ to properties of the propagators. Introduce the sum $U_\Sigma = \sum_{t=1}^{\infty} U_t$ and analogously the sums $U'_\Sigma, V_\Sigma, V'_\Sigma$ of $U'_t, V_t, V'_t$. To simplify exposition, we state the next theorem for the case when $U_t = U'_t$ and $V_t = V'_t$ for all $t$, which holds if $\mathbf{a} = 0$. See Section G for the general version.

**Theorem 3** (F). *Let numbers $L_t$ be given by expansion (19) with some $U_t = U'_t \geq 0, V_t = V'_t \geq 0$.*

1. *[**Convergence**] Suppose that $U_\Sigma < 1$. At $t \to \infty$, if $V_t = O(1)$ (respectively, $V_t = o(1)$), then also $L_t = O(1)$ (respectively, $L_t = o(1)$).*

2. *[**Divergence**] If $U_\Sigma > 1$ and $V_t > 0$ for at least one $t$, then $\sup_{t=1,2,\ldots} L_t = \infty$.*

3. *[**Signal-dominated regime**] Suppose that there exist constants $\xi_V, C_V > 0$ such that $V_t = C_V t^{-\xi_V}(1 + o(1))$ as $t \to \infty$. Suppose also that $U_\Sigma < 1$ and $U_t = O(t^{-\xi_U})$ with some $\xi_U > \max(\xi_V, 1)$. Then*

$$L_t = \frac{C_V}{2(1 - U_\Sigma)} t^{-\xi_V}(1 + o(1)). \tag{20}$$

4. *[**Noise-dominated regime**] Suppose that there exist constants $\xi_V > \xi_U > 1, C_U > 0$ such that $U_t = C_U t^{-\xi_U}(1 + o(1))$ and $V_t = O(t^{-\xi_V})$ as $t \to \infty$. Suppose also that $U_\Sigma < 1$. Then*

$$L_t = \frac{V_\Sigma C_U}{2(1 - U_\Sigma)^2} t^{-\xi_U}(1 + o(1)). \tag{21}$$

The first two statement characterize convergence in general, while the last two describe the regimes occurring under power-law propagator decay assumptions. Note that in the non-stochastic GD setting the noise propagators $U_t$ vanish so that $L_t = \frac{1}{2} V_{t+1}$. Then, statements 1 and 2 say that convergence of SGD with memory is equivalent to that of the respective non-stochastic GD plus the condition $U_\Sigma < 1$. In the general case of $\mathbf{a} \neq 0$, when $U_\Sigma \neq U'_\Sigma$, the condition is $U'_\Sigma < 1$ (Sec. G).

Formulas (20), (21) have a simple meaning (see Fig. 2, right). In the signal-dominated case the leading contribution to $L_t$ in expansion (19) comes from terms having one long signal propagator and several short noise propagators. The factor $C_V t^{-\xi_V}$ in Eq. (20) comes from the long signal propagator, while the coefficient $(1 - U_\Sigma)^{-1}$ is the accumulated contribution of the noise propagator products, summed over all possible partitions. In the noise-dominated case, the leading terms have one long noise propagator contributing the factor $C_U t^{-\xi_U}$ in Eq. (21). The coefficient $\frac{V_\Sigma}{(1-U_\Sigma)^2}$ is the accumulated effect of remaining short propagators before and after the long noise propagator.

**Stability of the maps $A_\lambda$.** Now we start analyzing the particular form (17), (18) of our propagators $U_t, U'_t, V_t, V'_t$. Note first that the linear matrix maps $A = (A_\lambda)$ defined in Eq. (16) may not be stable in the sense of having eigenvalues larger than 1 in absolute value. As a result, this would generally lead to an exponential growth of the propagators and hence an eventual divergence of $L_t$.

At $\lambda = 0$, $A_{\lambda=0}$ acts as the tensor square $S_{\lambda=0} \otimes S_{\lambda=0}$, and its eigenvalues are the products $\mu_1 \mu_2$, where $\mu_1, \mu_2$ are the eigenvalues of $S_{\lambda=0}$. The eigenvalues of $S_{\lambda=0}$ are just 1 and the eigenvalues of $D$. We will assume henceforth that all the eigenvalues of $D$ are strictly less than 1 in absolute value. We will refer to matrices with this property as *strictly stable*. When $D$ is strictly stable, $A_{\lambda=0}$ has the simple leading eigenvalue 1 with the eigenvector $\mathbf{M}_0 = \begin{pmatrix} 1 & 0 \\ 0 & 0 \end{pmatrix}$. All the other eigenvalues of $A_{\lambda=0}$ are less than 1 in absolute value. This condition generalizes the standard condition $|\beta| < 1$ for the momentum (Tugay & Tanik, 1989; Roy & Shynk, 1990).

As $\lambda$ increases, $A_\lambda$ acquires a more complicated form, and its eigenvalues continuously deform from the eigenvalues of $A_{\lambda=0}$. It is hard to give an exact condition of stability for general $\lambda$ and $\frac{\tau_2}{|B|}$, but we prove the following:

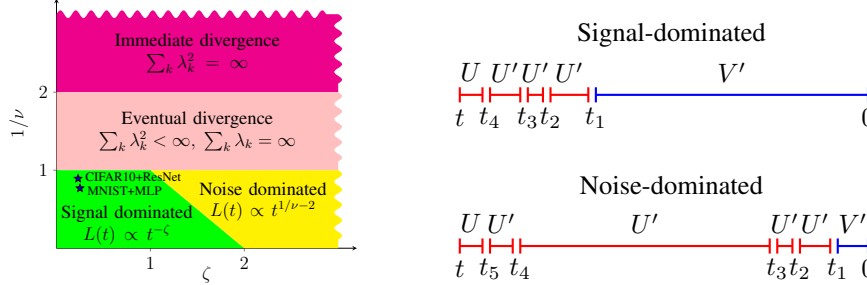

Figure 2: **Left:** The phase diagram of stationary SGD with momentum from Velikanov et al. (2023). Theorems 3, 5 show that it remains valid for general memory size $M$. **Right:** Leading terms in the loss expansion (19). In the signal-dominated phase the leading terms have one long signal propagator $V'$ and many short noise propagators $U'$. In the noise-dominated phase the leading terms have one long noise propagator $U'$, while the other propagators $U'$ and the signal propagators $V'$ are short.

**Theorem 4** (H). *Suppose that $D$ is strictly stable. Define effective learning rate $\alpha_{\text{eff}}$ by*

$$\alpha_{\text{eff}} = \alpha - \mathbf{b}^T (1-D)^{-1} \mathbf{c} = \frac{\sum_{m=0}^{M} q_m}{\sum_{m=0}^{M} m p_m - M - 1}. \tag{22}$$

*Then for sufficiently small $\lambda$ the leading eigenvalue of the linear maps $A_\lambda$ is $\mu_{A,\lambda} = 1 - 2\alpha_{\text{eff}}\lambda + O(\lambda^2)$. If $\alpha_{\text{eff}} > 0$, then for sufficiently small $\lambda > 0$ the linear maps $A_\lambda$ are strictly stable.*

**The power-law phase diagram.** We verify now the hypotheses of Theorem 3 under the power-law spectral assumptions (1).

**Theorem 5** (I). *Keep assumptions of Theorem 4 and suppose that $A_\lambda$ are strictly stable for all $0 < \lambda \le \lambda_{\max} \equiv \max_k \lambda_k$. Then, under spectral assumptions (1) with $\nu > \frac{1}{2}$, the values $V_t, V_t', U_t, U_t'$ given by Eqs. (17), (18) obey, as $t \to \infty$,*

$$V_t', V_t = (1 + o(1)) Q\Gamma(\zeta + 1)(2\alpha_{\text{eff}} t)^{-\zeta}, \tag{23}$$

$$U_t', U_t = (1 + o(1)) \frac{(\alpha_{\text{eff}} \Lambda)^{1/\nu} \tau_1 \Gamma(2 - 1/\nu)}{|B|\nu} (2t)^{1/\nu - 2}. \tag{24}$$

We see that the phase diagram for stationary SGD Berthier et al. (2020) remains valid for arbitrary memory size (see Fig. 2). Moreover, the algorithm parameters affect the leading terms in $V_t, V_t', U_t, U_t'$ only though the effective learning rate $\alpha_{\text{eff}}$ and the batch size $|B|$. However, loss asymptotics (20), (21) depend not only on the leading terms in $V_t, U_t$, but also on their sums $V_\Sigma, U_\Sigma$ (in general on $V_\Sigma', U_\Sigma, U_\Sigma'$, see Sec. G). In particular, in the signal-dominated scenario, if we wish to accelerate convergence of $L_t$, we need to increase $\alpha_{\text{eff}}$ while keeping $U_\Sigma' < 1$. In principle, if $U_\Sigma' < \infty$, we can always ensure $U_\Sigma' < 1$ by taking $|B|$ large enough. One can ask, however, if $\alpha_{\text{eff}}$ can be indefinitely increased while keeping $|B|$ and preserving the condition $U_\Sigma' < 1$. It was shown in Velikanov et al. (2023) that this is impossible for SGD with momentum. We will prove, however, that this can be done even with memory $M = 1$, by slightly generalizing the model.

## 6 DETAILED INVESTIGATION OF MEMORY-1 ALGORITHM

We aim to check whether memory-1 algorithm can achieve accelerated convergence in the signal-dominated phase $\zeta < 2 - \frac{1}{\nu}$ with a suitable choice of its five scalar hyperparameters $\alpha_t, a_t, b_t, c_t, D_t$. In this section, we will assume that $\tau_2 = 0$ to ensure factorization $A_\lambda = S_\lambda \otimes S_\lambda$, and strict stability of $A_\lambda$ reduces to that of $S_\lambda$.

Let us start with the stationary case. According to theorems 3 and 5, stable loss trajectories have the loss asymptotic $L_t \sim C_L t^{-\zeta}$ with $C_L \propto \alpha_{\text{eff}}^{-\zeta}$. Hence, we want to maximize $\alpha_{\text{eff}}(\alpha, a, b, c, D)$ under the stability constraints. The latter consist of the noiseless stability condition $|S_\lambda| < 1$ followed by

the noisy condition $U'_\Sigma < 1$. To proceed, we find it convenient to denote $\delta = 1 - D$ and use multi-step recurrence parametrization of Theorem 1 with $p_1 = 2 - \delta$ and $q_0, q_1$ given by (10). In particular, applying (22) to memory-1 case gives $\alpha_{\text{eff}} = -\frac{q_0 + q_1}{\delta}$. Then, we characterize the stability with

**Theorem 6** (J). *Stability of all memory-1 algorithms is fully determined by the triplet* $(\delta, \alpha_{\text{eff}}, q_0)$.

1. *Given* $\lambda_{\max} > 0$, *the matrix $D$ and the maps $S_\lambda$ are strictly stable for all* $0 < \lambda \le \lambda_{\max}$ *if and only if* $0 < \delta < 2$, $-q_0 < \frac{\delta}{\lambda_{\max}}$, *and* $0 < \delta\alpha_{\text{eff}} < \frac{4 - 2\delta}{\lambda_{\max}} - 2q_0$.

2. *Assume* $0 < \delta < 2$ *and* $\alpha_{\text{eff}} > 0$. *If* $q_1 \le -\alpha_{\text{eff}}$, *then the eigenvalues of $S_\lambda$ are real for all* $\lambda > 0$. *If* $q_1 > -\alpha_{\text{eff}}$, *then they are either real or lie on the circle* $\{x \in \mathbb{C} : |q_1 x + q_0|^2 = \delta^2 \alpha_{\text{eff}}(\alpha_{\text{eff}} + q_1)\}$ *(see Fig. 3).*

3. *Denote* $R_\lambda = \sum_{t=0}^{\infty} \langle \binom{1}{a} \binom{1}{a}^T, A_\lambda^t \binom{-\alpha}{c} \binom{-\alpha}{c}^T \rangle$ *so that* $U'_\Sigma = \frac{\tau_1}{|B|} \sum_k \lambda_k^2 R_{\lambda_k}$. *Then for strictly stable $S_\lambda$ and $\tau_2 = 0$,*

$$R_\lambda = \frac{2\lambda q_0^2 + (\lambda q_0 + 2 - \delta)\delta\alpha_{\text{eff}}}{\lambda(\delta + \lambda q_0)(4 - 2\delta - \lambda(2q_0 + \delta\alpha_{\text{eff}}))}. \tag{25}$$

Let us discuss implications of the above result. First, taking into account $-q_0 < \frac{\delta}{\lambda_{\max}}$, the last stability condition in statement 1 gives the effective learning rate bound $\alpha_{\text{eff}} < \delta^{-1}\frac{4}{\lambda_{\max}}$. Therefore, the only possibility to have $\alpha_{\text{eff}} \to \infty$ is to take $\delta \to 0$. Now consider the noise stability condition $U'_\Sigma < 1$. In the limit $\delta \to 0$, (25) can be simplified, leading to three spectral regions separated by $\lambda_1^{\text{cr}} = \frac{\delta}{q_0}$ and $\lambda_2^{\text{cr}} = \frac{\delta\alpha_{\text{eff}}}{q_0^2}$ and having a distinct impact on the stability sum $\sum_{k=1}^{\infty} \lambda_k^2 R_{\lambda_k}$:

$$\lambda^2 R_\lambda \asymp \lambda \frac{\delta\alpha_{\text{eff}} + \lambda q_0^2}{\delta + \lambda q_0} \asymp \begin{cases} \alpha_{\text{eff}}\lambda, & 0 < \lambda < \lambda_1^{\text{cr}} \\ \delta\alpha_{\text{eff}}/q_0, & \lambda_1^{\text{cr}} < \lambda < \lambda_2^{\text{cr}} \\ q_0\lambda, & \lambda_2^{\text{cr}} < \lambda < \lambda_{\max} - \varepsilon, \end{cases} \tag{26}$$

where $\epsilon$ is a small constant and $f \asymp g$ denotes $cg < f < c'g$ for some $c, c' > 0$. In the last equivalence of (26) we assumed $q_0 > 0$, which turns out to be necessary for $\sum_{k=1}^{\infty} \lambda_k^2 R_{\lambda_k} < \infty$. These observations lead to a simple criterion of stable acceleration:

**Theorem 7** (K). *Consider a spectrum $\lambda_k$ s.t.* $\sum_k \lambda_k < \infty, \lambda_k < \lambda_{\max} = 1$, *and assume $\tau_2 = 0$.*

1. *Suppose we are adjusting the parameters $\delta, \alpha_{\text{eff}}, q_0$ so that $\alpha_{\text{eff}} \to \infty$ while maintaining the strict stability of $S_\lambda$ as described in Theorem 6, part 1. Then, the necessary and sufficient conditions for having* $\sum_{k=1}^{\infty} \lambda_k^2 R_{\lambda_k} = O(1)$ *are*

$$\delta \to 0; \quad q_0 > 0; \quad \sum_{k:\lambda_k < \frac{\delta}{q_0}} \lambda_k = O(\tfrac{1}{\alpha_{\text{eff}}}); \quad \left|\{k : \tfrac{\delta}{q_0} < \lambda_k < \tfrac{\delta\alpha_{\text{eff}}}{q_0^2}\}\right| = O\left(\tfrac{q_0}{\delta\alpha_{\text{eff}}}\right).$$

2. *Conditions from item 1 can always be satisfied. In the particular case of power law $\lambda_k = (1 + o(1))\Lambda k^{-\nu}, \nu > 1$, this can be achieved by choosing $\alpha_{\text{eff}} = \delta^{-h}, q_0 = \delta^g$ with any $0 \le g < 1$ and $h \le h_{\max} = (1 - \frac{1}{\nu})(1 - g)$. Moreover, for $g > 0$ and $h < h_{\max}$, the noisy stability condition $U'_\Sigma < 1$ can be satisfied with any batch size $|B|$ as $\delta \to 0$.*

This result fully settles our initial question of having arbitrarily small constant $C_L$ in the loss asymptotic, while maintaining stability.

Now, let us look closer at the parameters of the obtained accelerated algorithm. First, we see from the second item of theorem 7 that, at given $\delta$, the maximal $\alpha_{\text{eff}} = \delta^{-h}$ is reached with $h = 1 - \frac{1}{\nu}$ by choosing $g = 0$, corresponding to $q_0 = const$. Although these values are specific to power-law spectrum with exponent $\nu$, they suggest that in the space of algorithm parameters $(\delta, \alpha_{\text{eff}}, q_0) \in \mathbb{R}^3$ the region $\delta^{-1} \gg \alpha_{\text{eff}} \gg q_0 = const$ is the most promising for stable acceleration.

**Power-law schedule with accelerated rate.** Jacobi scheduled HB uses momentum $1 - \beta_t = \delta_t \propto t^{-1}$ to achieve accelerated rate $O(t^{-2\varsigma})$ in the noiseless setting (Brakhage, 1987). Heuristically, we can view this non-stationary algorithm as a sequence of stationary algorithms gradually changing so as to increase the effective learning rate $\alpha_{\text{eff}} = \frac{\alpha}{\delta} \to \infty$ while maintaining stability. Mimicking this connection, we take the accelerated stationary algorithm obtained above and construct

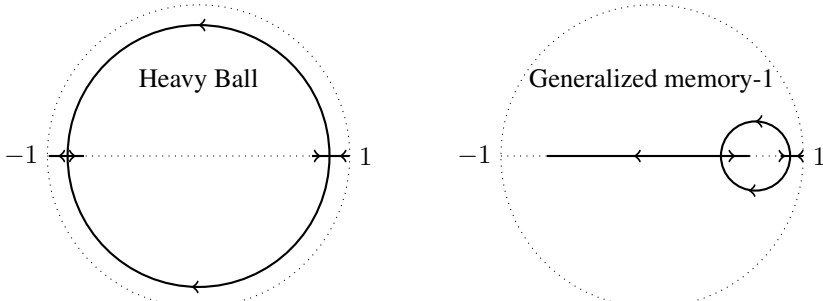

Figure 3: The geometric position and movement of complex eigenvalues of a memory-1 matrix $S_\lambda$ as $\lambda$ increases from 0 to 1 (see Theorem 6, part 2). **Left:** Classical Heavy Ball ($q_0 = 0$, Polyak (1964); Tugay & Tanik (1989)) with $\delta = 0.25, \alpha_{\text{eff}} = 14$. The circle of non-real eigenvalues is centered at the origin and has radius $r = \sqrt{1 - \delta}$. Acceleration ($\alpha_{\text{eff}} \gg 1$) requires $\delta \ll 1$ and hence a large circle, $r \approx 1$. **Right:** Our generalized memory-1 algorithm with $\delta = 0.15, \alpha_{\text{eff}} = 4, q_0 = 1.3$. The accelerated regime of Theorem 7 corresponds to small circles close to point 1.

a schedule $(\delta_t, \alpha_{\text{eff},t}, q_{0,t})$ that also achieves improved convergence rate. We call the result *accelerated memory-1 algorithm* (AM1), heuristically derive its convergence rate below, and verify it empirically in figures 1 and 4. We additionally discuss intuition behind AM1 update rule in sec. M.

We construct AM1 using the power-law ansatz $(\delta_t, \alpha_{\text{eff},t}, q_{0,t}) \sim (t^{-\overline{\delta}}, t^{\overline{\alpha}}, const)$. Assuming an approximate momentary stationarity as above and substituting this ansatz in Theorem 7 part 2, we get the stability condition $\overline{\alpha} \le \overline{\delta}(1 - \frac{1}{\nu})$. (We emphasize, however, that, strictly speaking, Theorem 7 only holds for stationary algorithms.)

Now we examine the loss convergence rate for these stable $\overline{\delta}, \overline{\alpha}$. By Theorem 3, the rate of $L_t$ in the signal-dominated phase is the same as that of the noiseless loss $V_t$. We assume that this is also true for our non-stationary algorithm, so we only need to estimate $V_t$. In sec. L, we replace evolution matrix $A_\lambda$ with its maximal eigenvalue $|\mu_{\max}(A_\lambda)|$ in eq. (17), and obtain $V_t = O(t^{-\zeta(1+\overline{\alpha})})$ for $\overline{\delta} < 1$. Summarizing, we arrive at the following convergence rate of AM1 algorithm:

$$L_t = O(t^{-\zeta(1+\overline{\alpha})}), \qquad 0 < \overline{\alpha} \le \overline{\delta}(1 - \tfrac{1}{\nu}), \quad 0 < \overline{\delta} < 1. \tag{27}$$

The maximal limiting convergence rate $O(t^{-\zeta(2-\frac{1}{\nu})})$ is obtained by setting $\overline{\delta} = 1, \overline{\alpha} = 1 - \frac{1}{\nu}$.

## 7 DISCUSSION

We have developed a general framework of SGD algorithms with size-$M$ memory and analyzed their convergence properties. We developed a loss expansion in terms of signal and noise propagators, and generalized the existing SGD phase diagram of power-law regimes to arbitrary $M$. In the signal-dominated regime, we have found the asymptotic loss convergence to depend on the algorithm mostly through the batch size and a scalar parameter $\alpha_{\text{eff}}$ that we call the effective learning rate. We have proved that already within memory-1 algorithms the effective learning rate can be arbitrarily increased while maintaining stability. As a consequence, we proposed a time-dependent schedule for memory-1 SGD which is heuristically expected to improve the standard SGD convergence $L_t = O(t^{-\zeta})$ up to $O(t^{-\zeta(2-\frac{1}{\nu})})$. This conjecture was confirmed by experiments with MNIST and synthetic problems.

Interestingly, while our framework allows shifts $\mathbf{w}_t + \mathbf{a}^T \mathbf{u}_t$ in the arguments of computed gradients $\nabla L$, and these shifts are important for Nesterov momentum (Nesterov, 1983) and averaging (Polyak & Juditsky, 1992), we have not found $\mathbf{a} \ne 0$ to be useful in our setting of quadratic problems and mini-batch noise. The special regime of stably increasing $\alpha_{\text{eff}}$ that we identified in Theorem 7 requires separation of scales between the three main memory-1 parameters unrelated to $\mathbf{a}$, and does not seem to be easily interpretable in terms of averaging or related general ideas.

Our results leave two obvious open problems: rigorously proving the accelerated convergence $L_t = O(t^{-\zeta(2-\frac{1}{\nu})})$ and further improving convergence by considering larger memory sizes $M$.

ACKNOWLEDGMENT

We thank Ivan Oseledets and the anonymous reviewers for useful discussions and suggestions that helped improve the paper.

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

CONTENTS

## A  RELATED WORK

**Linearization of neural networks.**  The relevance of quadratic optimization problems to deep learning is given by various scenarios where a neural network function $f(\mathbf{w}, \mathbf{x})$ can be linearized with respect to parameters $\mathbf{w}$

$$f_{\text{lin}}(\mathbf{w}, \mathbf{x}) = f(\mathbf{w}_0, \mathbf{x}) + \langle \mathbf{w} - \mathbf{w}_0, \nabla_{\mathbf{w}} f(\mathbf{w}_0, \mathbf{x}) \rangle, \qquad (28)$$

often referred to as kernel regime since $f_{\text{lin}}(\mathbf{w}, \mathbf{x})$ can be described in terms of respective neural tangent kernel (NTK) $K(\mathbf{x}, \mathbf{x}') = \langle \nabla_{\mathbf{w}} f(\mathbf{w}_0, \mathbf{x}), \nabla_{\mathbf{w}} f(\mathbf{w}_0, \mathbf{x}') \rangle$. One of such scenario is infinite width limit in NTK regime Jacot et al. (2018); Lee et al. (2019); Chizat et al. (2019), where linearization becomes exact. Another scenario, is an approximate linearization of practical neural network at certain stage of its training Fort et al. (2020); Maddox et al. (2021); Ortiz-Jimenez et al. (2021); Lee et al. (2020). We cover this scenario by linearized network features $\nabla_{\mathbf{w}} f(\mathbf{w}_0, \mathbf{x})$ as inputs $\mathbf{x}$ in (2).

**Power laws.**  Both classical kernels as well as NTK often display power-law shapes of their spectral distributions when applied to real world data. Typical examples include MNIST and CIFAR on either classical kernels or NTK of different architectures such as MLP or resnet (Cui et al., 2021; Bahri et al., 2021; Lee et al., 2020; Canatar et al., 2021; Kopitkov & Indelman, 2020; Dou & Liang, 2020; Atanasov et al., 2021; Bordelon & Pehlevan, 2021; Basri et al., 2020; Bietti, 2021; Wei et al., 2022; Atanasov et al., 2021). Power-law spectrum also appears theoretically in settings involing highly oscilating Fourier harmonics (Yang & Salman, 2020; Spigler et al., 2020; Basri et al., 2020; Velikanov & Yarotsky, 2021). Finally, power-law spectrum is often used as an assumption (Caponnetto & De Vito, 2007) for different optimization and generalization results on kernel methods.

Importantly, the exponent $\zeta$ in the power law spectra of real world problems is typically $\zeta < 1$, for example for MNIST different works report $\zeta \in [0.2, 0.4]$. This implies that the respective infinite dimensional problem with $\zeta < 1$ will have divergent series of coefficients $\|\mathbf{w}_*\|^2 = \sum_k c_k^2 = \infty$, so we can refer to respective targets as *infeasible*. Such setting makes vacuous many classical results relying on norm $\|\mathbf{w}_*\|^2$ (Wei et al., 2022; Velikanov & Yarotsky, 2024). Results in the present paper were developed to include the case $\zeta < 1$.

**Quadratic optimization under mini-batch noise.**  The properties of stochastic optimization can greatly vary with the structure of the noise. In particular, a common setting of additive noise (Polyak & Juditsky, 1992; Wu et al., 2020; Zhu et al., 2018) differs substantially from multiplicative noise (Moulines & Bach, 2011; Bach & Moulines, 2013) considered in the current paper. The mini-batch setting for power-law spectrum and its phase diagram with signal-dominated $\zeta < 2 - \frac{1}{\nu}$ and noise-dominated $\zeta > 2 - \frac{1}{\nu}$ regimes was identified in (Berthier et al., 2020; Varre et al., 2021) on the level of rigorous upper bounds, and in (Velikanov et al., 2023) on the level of loss symptotics under spectrally expressible (SE) approximation. The latter work relies on a generating function technique and provides the asymptotic form for the partial sums $\sum_{t \leq T} t L_t$ rather than for the losses $L_t$. In contrast, the combinatorial approach of propagator expansions used in the present work allows to directly obtain the asymptotics of $L_t$.

As we mentioned in the main paper, SE approximation, or its $\tau_2 = 0$ version, is often used as an upper bound assumption on the noise covariance (Bordelon & Pehlevan, 2021; Flammarion & Bach, 2017; Dieuleveut et al., 2017; Zou et al., 2021; Varre & Flammarion, 2022; Wu et al., 2022). Another series of work (Paquette & Paquette, 2021; Paquette et al., 2021; Lee et al., 2022; Paquette et al., 2022; Collins-Woodfin et al., 2023) consider a different high-dimensional setting where SGD converges deterministic dynamics described by convolution Volterra equation. This setting differes from ours in that eigenvalues are described by a limiting density, leading to the regime of immediate divergence $\sum_k \lambda_k^2 = \infty$, which has to be overcome by scaling the learning rate with dimension as $d^{-1}$. Apart from this difference, algebraic structure of continuous time Volterra equation is close to our discrete time propagator expansion (19), and we expect some of our results to be extendable to the aforementioned setting.

**Accelerated algorithms.** It is a standard textbook material that Heavy Ball accelerates convergence of plain non-stochastic gradient descent on quadratic problems (Polyak, 1987). In the setting of non-stochastic optimization of non strictly convex quadratic problems with power law spectral data, convergence rates of different GD-related methods were first analized by Nemirovskiy & Polyak (1984); Nemirovsky & Polyak (1984). In particular, they showed that under the source condition corresponding to the second equation in (1) with some $\zeta$, the loss convergence rate of standard stationary GD was $L_t = O(t^{-\zeta})$, and also gave a number of other upper and lower bounds for different algorithms. Brakhage (1987) introduced Heavy Ball with a time-dependent schedule based on Jacobi polynomials and showed that such schedules provide acceleration of the standard GD rate $L_t = O(t^{-\zeta})$ to $L_t = O(t^{-2\zeta})$ for any spectral exponent $\zeta$. This rate is optimal among algorithms with predefined schedule, though, as shown already by Nemirovskiy & Polyak (1984), Conjugate Gradients can provide an even faster convergence $L_t = O(t^{-(2+\nu)\zeta})$ (but in this algorithm the update coefficients are not predefined). See Hanke (1991; 1996); Velikanov & Yarotsky (2024) for several other results concerning acceleration of non-stochastic gradient descent on non-strongly convex quadratic problems under power law spectral assumptions.

A series of works (Flammarion & Bach, 2015; Varre & Flammarion, 2022) consider a special schedule of memory-1 SGD parameters such that after rescaling the iterates $\Delta \widetilde{\mathbf{w}}_t = t \Delta \mathbf{w}_t$ the update rule become stationary. Then, the stability of $\widetilde{\mathbf{w}}_t$ leads to $O(t^{-2})$ convergence of original iterates $\Delta \mathbf{w}_t$ as long as $\|\Delta \mathbf{w}_0\|^2 = \|\mathbf{w}_*\|^2 < \infty$. In particular, for mini-batch SGD under uniform kurtosis assumption (i.e. noise is bound by our SE term), Varre & Flammarion (2022) gives an upper bound $O(\frac{d}{t^2})$, where problem dimension $d$ appears from downscaling learning rate as $\alpha = O(d^{-1})$ to ensure stability. Potentially, the necessity to downscale learning rate of the algorithm of Varre & Flammarion (2022) can be traced to third stability condition of theorem 7, with the set $\left\{ k : \frac{\delta}{q_0} < \lambda_k < \frac{\delta \alpha_{\text{eff}}}{q_0^2} \right\}$ covering the whole spectrum, thus requiring small learning rate inversely proportional to the size of the set.

# B  GD WITH MEMORY

## B.1  SHIFT-INVARIANT AND EQUIVARIANT VECTORS

The vectors $\mathbf{w}_t$ in the evolution (3) are *shift-equivariant* while $\mathbf{u}_t$ are *shift-invariant*, in the following sense. Given a loss function $L(\cdot)$ and an initial approximation $\mathbf{w}_0$, consider their shifts by some vector $\mathbf{v}$ : $L'(\cdot) = L(\cdot - \mathbf{v}), \mathbf{w}_0' = \mathbf{w}_0 + \mathbf{v}$. Then running the algorithm (3) with the same parameters $\alpha_t, \mathbf{a}_t, \mathbf{b}_t, \mathbf{c}_t, D_t$ yields the shifted trajectory $\mathbf{w}_t' = \mathbf{w}_t + \mathbf{v}$, as is commonly expected, while preserving the auxiliary trajectory $\mathbf{u}_t' = \mathbf{u}_t$. This can be seen as a consequence of using the difference $\mathbf{w}_{t+1} - \mathbf{w}_t$ in the l.h.s., since $\mathbf{w}_t$ being shift-equivariant for all $t$ is equivalent to $\mathbf{w}_{t+1} - \mathbf{w}_t$ being shift-invariant for all $t$. Since the gradients $\nabla L(\mathbf{w}_t + \mathbf{a}_t^T \mathbf{u}_t)$ are also shift-invariant, all the matrix elements appearing in Eq. (4) are shift-invariant, which is consistent with the mentioned in- and equi-variance of the trajectories $\mathbf{u}_t$ and $\mathbf{w}_t$.

## B.2   PROOF OF THEOREM 1

*Proof.* 1. Consider the characteristic polynomial

$$\chi(x, \lambda) = \det(x - S_\lambda) = \det\left[x - \left(\begin{pmatrix} 1 & \mathbf{b}^T \\ 0 & D \end{pmatrix} + \lambda \begin{pmatrix} -\alpha \\ \mathbf{c} \end{pmatrix} (1, \mathbf{a}^T)\right)\right] = \sum_{m,k} r_{mk} x^m \lambda^k. \quad (29)$$

We can express $\chi$ in terms of $\alpha, \mathbf{a}, \mathbf{b}, \mathbf{c}, D$ by conjugating $S_\lambda$ with the matrices

$$U = \begin{pmatrix} 1 & \mathbf{a}^T \\ 0 & \mathbf{1} \end{pmatrix}, \quad U^{-1} = \begin{pmatrix} 1 & -\mathbf{a}^T \\ 0 & \mathbf{1} \end{pmatrix}, \quad (30)$$

namely

$$\chi(x, \lambda) = \det(x - US_\lambda U^{-1}) \quad (31)$$

$$= \det\left[x - \left(\begin{pmatrix} 1 & \mathbf{b}^T + \mathbf{a}^T(D-1) \\ 0 & D \end{pmatrix} + \lambda \begin{pmatrix} \mathbf{a}^T\mathbf{c} - \alpha \\ \mathbf{c} \end{pmatrix} (1, \mathbf{0})\right)\right] \quad (32)$$

$$= \det\begin{pmatrix} x - 1 - \lambda(\mathbf{a}^T\mathbf{c} - \alpha) & -(\mathbf{b}^T + \mathbf{a}^T(D-1)) \\ -\lambda\mathbf{c} & x - D \end{pmatrix} \quad (33)$$

$$= \det\begin{pmatrix} x - 1 & -(\mathbf{b}^T + \mathbf{a}^T(D-1)) \\ 0 & x - D \end{pmatrix} - \lambda \det\begin{pmatrix} \mathbf{a}^T\mathbf{c} - \alpha & -(\mathbf{b}^T + \mathbf{a}^T(D-1)) \\ \mathbf{c} & x - D \end{pmatrix} \quad (34)$$

$$= (x - 1)\det(x - D) - \lambda\det\begin{pmatrix} \mathbf{a}^T\mathbf{c} - \alpha & \mathbf{a}^T(1 - D) - \mathbf{b}^T \\ \mathbf{c} & x - D \end{pmatrix}. \quad (35)$$

This shows, in particular, that $\chi$ can be written in the form

$$\chi(x, \lambda) = x^{M+1} - \sum_{m=0}^{M} p_m x^m - \lambda \sum_{m=0}^{M} q_m x^m \quad (36)$$

with some coefficients $(p_m)_{m=0}^{M}, (q_m)_{m=0}^{M}$ such that $\sum_{m=0}^{M} p_m = 1$.

On the other hand, by the Cayley-Hamilton theorem, $\chi(S_\lambda, \lambda) = 0$. In the time-independent scenario $\lambda$-eigenvectors $\begin{pmatrix} \Delta\mathbf{w}_t \\ \mathbf{u}_t \end{pmatrix}$ are evolved by

$$\begin{pmatrix} \Delta\mathbf{w}_{t+m} \\ \mathbf{u}_{t+m} \end{pmatrix} = S_\lambda^m \begin{pmatrix} \Delta\mathbf{w}_t \\ \mathbf{u}_t \end{pmatrix}. \quad (37)$$

Recalling the expansion $\chi(x, \lambda) = \sum_{m,k} r_{mk} x^m \lambda^k$ of the characteristic polynomial, it follows that

$$0 = \left(\sum_{m,k} r_{mk} S_\lambda^m \lambda^k\right) \begin{pmatrix} \Delta\mathbf{w}_t \\ \mathbf{u}_t \end{pmatrix} = \sum_{m,k} r_{mk} \lambda^k \begin{pmatrix} \Delta\mathbf{w}_{t+m} \\ \mathbf{u}_{t+m} \end{pmatrix} = \sum_{m,k} r_{mk} \mathbf{H}^k \begin{pmatrix} \Delta\mathbf{w}_{t+m} \\ \mathbf{u}_{t+m} \end{pmatrix}. \quad (38)$$

In particular, taking the top row,

$$\sum_{m,k} r_{mk} \mathbf{H}^k \Delta\mathbf{w}_{t+m} = 0. \quad (39)$$

Since this equation is $\lambda$-independent, it holds not only for eigenvectors $\Delta\mathbf{w}_{t+m}$, but, by linearity, for any vectors $\Delta\mathbf{w}_{t+m}$ connected by the matrix dynamics (3).

Finally, we obtain the desired identity (8) by substituting the expression (36) for $\chi$, recalling that $\sum_{m=0}^{1} p_m = 1$, and noting that $\mathbf{H}\Delta\mathbf{w} = \nabla L(\mathbf{w})$ :

$$0 = \Delta\mathbf{w}_{t+M+1} - \sum_{m=0}^{M} p_m \Delta\mathbf{w}_{t+m} - \sum_{m=0}^{M} q_m \mathbf{H}\Delta\mathbf{w}_{t+m} \quad (40)$$

$$= \mathbf{w}_{t+M+1} - \sum_{m=0}^{M} p_m \mathbf{w}_{t+m} - \sum_{m=0}^{M} q_m \nabla L(\mathbf{w}_{t+m}). \quad (41)$$

2. The proof will consist of two components:

a) Construct $\alpha, \mathbf{a}, \mathbf{b}, \mathbf{c}, D$ so as to satify Eqs. (9),(10). By part 1) of the theorem, this will ensure that the sequence $\mathbf{w}_t$ satisfies the recurrence (8).

b) Show that for any sequence $\mathbf{w}_0, \mathbf{w}_1, \ldots, \mathbf{w}_M$ required to initialize recurrence (8) one can find an initial condition $\left(\begin{smallmatrix}\mathbf{w}_0\\\mathbf{u}_0\end{smallmatrix}\right)$ for the matrix evolution (3) to produce this initial sequence.

For a), we first satisfy Eq. (9), i.e. construct a matrix $D$ such that

$$\det(x - D) = \frac{x^{M+1} - \sum_{m=0}^{M} p_m x^m}{x - 1} = x^M + \sum_{m=0}^{M-1} p'_m x^m, \tag{42}$$

where

$$p'_m = \sum_{k=0}^{m} p_k, \quad m = 0, \ldots, M - 1. \tag{43}$$

A $M \times M$ matrix $D$ with such $\det(x - D)$ can be given as

$$D = \begin{pmatrix} -p'_{M-1} & 1 & 0 & \cdots & 0 \\ -p'_{M-2} & 0 & 1 & \cdots & 0 \\ & & \cdots & & \\ -p'_1 & 0 & 0 & \cdots & 1 \\ -p'_0 & 0 & 0 & \cdots & 0 \end{pmatrix}. \tag{44}$$

Now to satisfy Eq. (10) we choose $\mathbf{a} = \mathbf{0}, \mathbf{b} = (1, 0, \ldots, 0)^T$, so that this condition becomes

$$\sum_{m=0}^{M} q_m x^m = \det \begin{pmatrix} -\alpha & -\mathbf{b}^T \\ \mathbf{c} & x - D \end{pmatrix} \tag{45}$$

$$= -\alpha \det(x - D) + \begin{pmatrix} c_1 & -1 & 0 & \cdots & 0 \\ c_2 & x & -1 & \cdots & 0 \\ & & \cdots & & \\ c_{M-1} & 0 & 0 & \cdots & -1 \\ c_M & 0 & 0 & \cdots & x \end{pmatrix} \tag{46}$$

$$= -\alpha \det(x - D) + \sum_{m=0}^{M-1} c_{M-1-m} x^m. \tag{47}$$

Observe that there is a unique assignment of $\alpha$ and $\mathbf{c}$ that satisfies this identity: first we assign $\alpha = -q_M$ so that $\sum_{m=0}^{M} q_m x^m + \alpha \det(x - D)$ is a polynomial of degree not greater than $M - 1$, and its coefficients are then exactly the components of the vector $\mathbf{c}$.

Having constructed $\alpha, \mathbf{a}, \mathbf{b}, \mathbf{c}, D$, we check now the completeness condition b). Note that since we have set $\mathbf{a} = 0$, the gradients $\nabla L(\mathbf{w}_t + \mathbf{a}^T \mathbf{u}_t) = \nabla L(\mathbf{w}_t)$ appearing in Eq. (3) only depend on the target sequence $\mathbf{w}_t$ and not on the auxiliary sequence $\mathbf{u}_t$. In the first step of matrix iterations we have

$$\mathbf{w}_1 - \mathbf{w}_0 = -\alpha \nabla L(\mathbf{w}_0) + \mathbf{b}^T \mathbf{u}_0 = \mathbf{v}_1 + \mathbf{u}_0^{(1)} \tag{48}$$

with some $\mathbf{v}_1$ that depends only on $\mathbf{w}_0$ and not on $\mathbf{u}_0$. In the second step, using the explicit form (44) of the matrix $D$,

$$\mathbf{w}_2 - \mathbf{w}_1 = -\alpha \nabla L(\mathbf{w}_1) + \mathbf{b}^T \mathbf{u}_1 \tag{49}$$

$$= -\alpha \nabla L(\mathbf{w}_1) + \mathbf{b}^T (\nabla L(\mathbf{w}_0)\mathbf{c} + D\mathbf{u}_0) \tag{50}$$

$$= -\alpha \nabla L(\mathbf{w}_1) + \nabla L(\mathbf{w}_0)c_1 - p'_{M-1}\mathbf{u}_0^{(1)} + \mathbf{u}_0^{(2)} \tag{51}$$

$$= \mathbf{v}_2 + \mathbf{u}_0^{(2)}, \tag{52}$$

where $\mathbf{v}_2$ is a vector depending on $\mathbf{w}_0, \mathbf{w}_1$ and the first component $\mathbf{u}_0^{(1)}$ of $\mathbf{u}_0$, but not on the second component $\mathbf{u}_0^{(2)}$.

By repeating this process, we see that for all $m = 1, 2, \ldots M$ we can write

$$\mathbf{w}_m - \mathbf{w}_{m-1} = \mathbf{v}_m + \mathbf{u}_0^{(m)} \tag{53}$$

with some vector $\mathbf{v}_m$ that depends on $\mathbf{w}_0, \ldots, \mathbf{w}_{m-1}$ and on $\mathbf{u}_0^{(1)}, \ldots, \mathbf{u}_0^{(m-1)}$, but not on $\mathbf{u}_0^{(m)}$. This shows that by sequentially adjusting the components $\mathbf{u}_0^{(m)}$ we can ensure the desired values for all the differences $\mathbf{w}_m - \mathbf{w}_{m-1}, m = 1, \ldots, M$, i.e. we can find the initial condition $\left( \begin{smallmatrix} \mathbf{w}_0 \\ \mathbf{u}_0 \end{smallmatrix} \right)$ producing the desired sequence $\mathbf{w}_0, \ldots, \mathbf{w}_M$.

□

### B.3 Proof of Proposition 1

Suppose that $L$ is quadratic and consider iterations (7) in the eigenspace representation, starting from some $\left( \begin{smallmatrix} \Delta \mathbf{w}_0 \\ \mathbf{u}_0 \end{smallmatrix} \right)$. Observe that $\left( \begin{smallmatrix} \Delta \mathbf{w}_t \\ \mathbf{u}_t \end{smallmatrix} \right)$ is polynomial in $\lambda$, with degree at most $t$ and the associated leading term resulting from applying the rank-1 $\lambda$-term in each step:

$$\begin{pmatrix} \Delta \mathbf{w}_t \\ \mathbf{u}_t \end{pmatrix} = \lambda^t \begin{pmatrix} -\alpha_t \\ c_t \end{pmatrix} \Big[ \prod_{s=1}^{t-1} (c_s a_{s+1} - \alpha_s) \Big] (\Delta \mathbf{w}_0 + a \mathbf{u}_0) + O(\lambda^{t-1}). \tag{54}$$

In particular,

$$\Delta \mathbf{w}_1 = \lambda(-\alpha_1)(\Delta \mathbf{w}_0 + a_1 \mathbf{u}_0) + O(1) \tag{55}$$

$$\Delta \mathbf{w}_2 = \lambda^2(-\alpha_2)(c_1 a_2 - \alpha_1)(\Delta \mathbf{w}_0 + a_1 \mathbf{u}_0) + O(\lambda) \tag{56}$$

Suppose now that $\alpha_1 = 0$ while $\alpha_2 \neq 0$ and $c_1 a_2 \neq 0$. Then the leading term in $\Delta \mathbf{w}_1$ vanishes and the degree of $\Delta \mathbf{w}_1$ drops to 0, while the degree of $\Delta \mathbf{w}_2$ is exactly 2. Then, it is impossible for generic $\Delta \mathbf{w}_0, \mathbf{u}_0$ to represent $\Delta \mathbf{w}_2$ as

$$\Delta \mathbf{w}_2 = \lambda q_1 \Delta \mathbf{w}_1 + \lambda q_0 \Delta \mathbf{w}_0 + p_1 \Delta \mathbf{w}_1 + p_0 \Delta \mathbf{w}_0, \tag{57}$$

since the r.h.s. is a polynomial in $\lambda$ of degree at most 1. Representation (57) is equivalent to

$$\mathbf{w}_2 = p_1 \mathbf{w}_1 + p_0 \mathbf{w}_0 + q_1 \nabla L(\mathbf{w}_1) + q_0 \nabla L(\mathbf{w}_0) - (p_1 + p_0 - 1) \mathbf{w}_*. \tag{58}$$

We can assume $\mathbf{w}_* = 0$, then this has exactly the form as in the statement of the proposition.

## C SGD with memory

### C.1 Evolution of second moments

In a single step of SGD, the parameters and memory vectors are updated with stochastic batch Hessian $\mathbf{H}_t = \frac{1}{|B_t|} \sum_{\mathbf{x}_i \in B_t} \mathbf{x}_i \otimes \mathbf{x}_i$ as

$$\begin{pmatrix} \Delta \mathbf{w}_{t+1} \\ \mathbf{u}_{t+1} \end{pmatrix} = \mathbf{S}_t \begin{pmatrix} \Delta \mathbf{w}_t \\ \mathbf{u}_t \end{pmatrix}, \quad \mathbf{S}_t \equiv \begin{pmatrix} 1 & \mathbf{b}_t^T \\ 0 & D_t \end{pmatrix} + \mathbf{H}_t \otimes \begin{pmatrix} -\alpha_t \\ \mathbf{c}_t \end{pmatrix} (1, \mathbf{a}_t^T) \tag{59}$$

To lighten notation, we will sometimes drop the tensor product sign between Hessian and the algorithm parameter matrix, writing simply as $\mathbf{H}_t \left( \begin{smallmatrix} -\alpha_t \\ \mathbf{c}_t \end{smallmatrix} \right)(1, \mathbf{a}_t^T)$.

To obtain the update of second moments (11), we simply substitute the update of parameters above, and take the expectation with respect to draw of batches $B_1, B_2, \ldots B_t$

$$\mathbf{M}_{t+1} = \mathbb{E}_{B_1, \ldots, B_{t-1}, B_t} \left[ \begin{pmatrix} \Delta \mathbf{w}_{t+1} \\ \mathbf{u}_{t+1} \end{pmatrix} \begin{pmatrix} \Delta \mathbf{w}_{t+1} \\ \mathbf{u}_{t+1} \end{pmatrix}^T \right]$$

$$= \mathbb{E}_{B_1, \ldots, B_{t-1}, B_t} \left[ \mathbf{S}_t \begin{pmatrix} \Delta \mathbf{w}_t \\ \mathbf{u}_t \end{pmatrix} \begin{pmatrix} \Delta \mathbf{w}_t \\ \mathbf{u}_t \end{pmatrix}^T \mathbf{S}_t^T \right] = \mathbb{E}_{B_t} \left[ \mathbf{S}_t \mathbf{M}_t \mathbf{S}_t^T \right]. \tag{60}$$

To calculate the last expectation, we use that samples $\mathbf{x}_i \in B_t$ are drawn i.i.d. from distribution $\rho$. Then, for any fixed matrix $\mathbf{G}$ in $\mathcal{H}$, the expectations involving two batch Hessians are computed as

$$\mathbb{E}_{B_t}[\mathbf{H}_t \mathbf{G} \mathbf{H}_t] = \mathbb{E}_{B_t} \left[ \frac{1}{|B_t|^2} \sum_{i,j=1}^{|B_t|} \langle \mathbf{x}_i, \mathbf{G} \mathbf{x}_j \rangle \mathbf{x}_i \otimes \mathbf{x}_j \right]$$

$$= \frac{|B_t| - 1}{|B_t|} \mathbf{H} \mathbf{G} \mathbf{H} + \frac{1}{|B_t|} \mathbb{E}_{\mathbf{x} \sim \rho}[\langle \mathbf{x}, \mathbf{G} \mathbf{x} \rangle \mathbf{x} \otimes \mathbf{x}] \tag{61}$$

$$= \mathbf{H} \mathbf{G} \mathbf{H} + \frac{1}{|B_t|} \Sigma_\rho(\mathbf{G}),$$

where the sampling noise variance $\Sigma_\rho(\mathbf{G})$ is defined by eq. (14).

Using the above, we finish the computation of second moments update

$$
\begin{aligned}
\mathbf{M}_{t+1} =& \mathbb{E}_{B_t}\left(\left(\begin{pmatrix} 1 & \mathbf{b}_t^T \\ 0 & D_t \end{pmatrix} + \mathbf{H}_t\begin{pmatrix} -\alpha_t \\ \mathbf{c}_t \end{pmatrix}(1, \mathbf{a}_t^T)\right)\mathbf{M}_t\left(\begin{pmatrix} 1 & \mathbf{b}_t^T \\ 0 & D_t \end{pmatrix} + \mathbf{H}_t\begin{pmatrix} -\alpha_t \\ \mathbf{c}_t \end{pmatrix}(1, \mathbf{a}_t^T)\right)^T\right) \\
=& \left(\begin{pmatrix} 1 & \mathbf{b}_t^T \\ 0 & D_t \end{pmatrix} + \mathbf{H}\begin{pmatrix} -\alpha_t \\ \mathbf{c}_t \end{pmatrix}(1, \mathbf{a}_t^T)\right)\mathbf{M}_t\left(\begin{pmatrix} 1 & \mathbf{b}_t^T \\ 0 & D_t \end{pmatrix} + \mathbf{H}\begin{pmatrix} -\alpha_t \\ \mathbf{c}_t \end{pmatrix}(1, \mathbf{a}_t^T)\right)^T \\
&+ \frac{1}{|B_t|}\Sigma_\rho\left(\mathbf{C}_t + 2\mathbf{J}_t\mathbf{a}_t + \mathbf{a}_t^T\mathbf{V}\mathbf{a}_t^T\right) \otimes \begin{pmatrix} -\alpha_t \\ \mathbf{c}_t \end{pmatrix}(-\alpha_t, \mathbf{c}_t).
\end{aligned}
\tag{62}
$$

## C.2 SPECTRALLY-EXPRESSIBLE APPROXIMATION

As usual, for two Hermitian operators $A, B$ we write $A \geq B$ (or $A \leq B$) meaning that $A - B$ is positive (negative) semi-definite.

Recall the maps $\Sigma_\rho$ and $\Sigma_{\text{SE}}^{(\tau_1,\tau_2)}$ defined in Eqs. (14), (15):

$$
\Sigma_\rho(\mathbf{C}) = \mathbb{E}_{\mathbf{x}\sim\rho}[\langle \mathbf{x}, \mathbf{C}\mathbf{x}\rangle \mathbf{x} \otimes \mathbf{x}] - \mathbf{HCH},
\tag{63}
$$

$$
\Sigma_{\text{SE}}^{(\tau_1,\tau_2)}(\mathbf{C}) = \tau_1 \operatorname{Tr}[\mathbf{HC}]\mathbf{H} - \tau_2\mathbf{HCH}.
\tag{64}
$$

In the following lemma we collect a few useful properties of these maps.

**Lemma 1.** *Suppose that $\mathbf{H} \geq 0$.*

1. *For any $\mathbf{C} \geq 0$ we have $\mathbf{HCH} \leq \operatorname{Tr}(\mathbf{HC})\mathbf{H}$.*

2. *For any rank-1 Hermitian $\mathbf{H}$ we have $\mathbf{HCH} = \operatorname{Tr}(\mathbf{HC})\mathbf{H}$.*

3. *If $0 \leq \tau_1 \geq \tau_2$, then $\Sigma_{\text{SE}}^{(\tau_1,\tau_2)}(\mathbf{C}) \geq 0$ for any $\mathbf{C} \geq 0$, and, more generally, $\Sigma_{\text{SE}}^{(\tau_1,\tau_2)}(\mathbf{C}) \geq \Sigma_{\text{SE}}^{(\tau_1,\tau_2)}(\mathbf{C}')$ for any $\mathbf{C} \geq \mathbf{C}'$.*

4. *If $\tau_1 < 0$ or $\tau_2 > \tau_1$, then it is no longer true in general that $\Sigma_{\text{SE}}^{(\tau_1,\tau_2)}(\mathbf{C}) \geq 0$ for any $\mathbf{C} \geq 0$.*

5. *$\Sigma_{\text{SE}}^{(\tau_1+\tau,\tau_2+\tau)}(\mathbf{C}) \geq \Sigma_{\text{SE}}^{(\tau_1,\tau_2)}(\mathbf{C})$ for any $\mathbf{C} \geq 0$ and $\tau \geq 0$.*

6. *$\Sigma_\rho(\mathbf{C}) \geq 0$ for any $\mathbf{C} \geq 0$ and, more generally, $\Sigma_\rho(\mathbf{C}) \geq \Sigma_\rho(\mathbf{C}')$ for any $\mathbf{C} \geq \mathbf{C}'$.*

*Proof.*

1. We need to show that for any vector $\mathbf{u}$

$$
\langle \mathbf{u}, \mathbf{HCHu}\rangle \leq \operatorname{Tr}(\mathbf{HC})\langle \mathbf{u}, \mathbf{Hu}\rangle.
\tag{65}
$$

By considering $\mathbf{u} = \mathbf{H}^{\frac{1}{2}}\mathbf{u}$ and rescaling $\mathbf{v} = \mathbf{u}/\|\mathbf{u}\|$, it is sufficient to show that $\langle \mathbf{v}, \mathbf{H}^{\frac{1}{2}}\mathbf{CH}^{\frac{1}{2}}\mathbf{v}\rangle \leq \operatorname{Tr}(\mathbf{HC})$ for any unit vector $\mathbf{v}$. But this just follows from the positive semi-definiteness of $\mathbf{H}^{\frac{1}{2}}\mathbf{CH}^{\frac{1}{2}}$ and the cyclic property of the trace:

$$
\langle \mathbf{v}, \mathbf{H}^{\frac{1}{2}}\mathbf{CH}^{\frac{1}{2}}\mathbf{v}\rangle \leq \operatorname{Tr}(\mathbf{H}^{\frac{1}{2}}\mathbf{CH}^{\frac{1}{2}}) = \operatorname{Tr}(\mathbf{HC}).
\tag{66}
$$

2. Immediate.

3. For $\mathbf{C} \geq 0$ we have $\Sigma_{\text{SE}}^{(\tau_1,\tau_2)}(\mathbf{C}) = \tau_1[\operatorname{Tr}(\mathbf{HC})\mathbf{H} - \mathbf{HCH}] + (\tau_2 - \tau_1)\mathbf{HCH} \geq 0$ by statement 1 and positive semi-definiteness of $\mathbf{HCH}$.

   Monotonicity follows from the linearity of $\Sigma_{\text{SE}}^{(\tau_1,\tau_2)}$.

4. If $\tau_1 < 0$, consider $\mathbf{H} = \mathbf{1}$ and a rank-1 $\mathbf{C}$. Then $\Sigma_{\text{SE}}^{(\tau_1,\tau_2)}(\mathbf{C}) = \tau_1(\operatorname{Tr}\mathbf{C})\mathbf{1} - \tau_2\mathbf{C}$, which is not positive semi-definite unless the space is 1-dimensional.

   If $\tau_2 > \tau_1$, consider a rank-1 $\mathbf{H}$. By Statement 2, $\Sigma_{\text{SE}}^{(\tau_1,\tau_2)}(\mathbf{C}) = (\tau_1 - \tau_2)\mathbf{HCH}$, which is negative semi-definite.

5. Follows from statement 1.

6. It suffices to show that for any two vectors $\mathbf{u}, \mathbf{v}$ we have $\langle \mathbf{u}, \Sigma_\rho(\mathbf{v} \otimes \mathbf{v})\mathbf{u} \rangle \geq 0$, i.e.

$$\mathbb{E}_{\mathbf{x}\sim\rho}[\langle \mathbf{u}, \mathbf{x} \rangle^2 \langle \mathbf{x}, \mathbf{v} \rangle^2] \geq \mathbb{E}_{\mathbf{x}\sim\rho}^2[\langle \mathbf{u}, \mathbf{x} \rangle \langle \mathbf{x}, \mathbf{v} \rangle]. \tag{67}$$

But this is just a special case of the Cauchy inequality $\mathbb{E}_{\mathbf{x}\sim\rho}^2[f] \leq \mathbb{E}_{\mathbf{x}\sim\rho}[f^2]$ with $f(\mathbf{x}) = \langle \mathbf{u}, \mathbf{x} \rangle \langle \mathbf{x}, \mathbf{v} \rangle$.

Monotonicity again follows from the linearity of $\Sigma_\rho$.

$\square$

We can now prove Proposition 2.

*Proof of Proposition 2.* By statements 3 and 6 of Lemma, both $\Sigma_{\text{SE}}^{(\tau_1,\tau_2)}$ and $\Sigma_\rho$ preserve positive semi-definiteness and are monotone in the operator sense.

Let us prove that $\Sigma_\rho(\mathbf{C}) \leq \Sigma_{\text{SE}}^{(\tau_1,\tau_2)}(\mathbf{C})$ for all $\mathbf{C} \geq 0$ implies $L_t^{(\rho)} \leq L_t^{(\tau_1,\tau_2)}$ for all $t$. It suffices to show that for all $t = 0, 1 \ldots$ we have $0 \leq \mathbf{C}_t \leq \mathbf{C}_t'$, where $\mathbf{C}_t$ and $\mathbf{C}_t'$ denote the $\mathbf{C}$-components of the matrices $\mathbf{M}_t, \mathbf{M}_t'$ corresponding to the dynamics with $\Sigma_\rho$ and $\Sigma_{\text{SE}}^{(\tau_1,\tau_2)}$, respectively. In turn, it is sufficient to show that $0 \leq \mathbf{M}_t \leq \mathbf{M}_t'$ for all $t$. By induction, suppose that $0 \leq \mathbf{M}_t \leq \mathbf{M}_t'$ holds for some $t$. Then, in particular,

$$0 \leq (1, \mathbf{a}_t^T)\mathbf{M}_t(1, \mathbf{a}_t^T)^T \leq (1, \mathbf{a}_t^T)\mathbf{M}_t'(1, \mathbf{a}_t^T)^T. \tag{68}$$

Then, using the hypothesis of the proposition and the monotonicity of $\Sigma_{\text{SE}}^{(\tau_1,\tau_2)}$,

$$0 \leq \Sigma_\rho\left((1, \mathbf{a}_t^T)\mathbf{M}_t(1, \mathbf{a}_t^T)^T\right) \tag{69}$$

$$\leq \Sigma_{\text{SE}}^{(\tau_1,\tau_2)}\left((1, \mathbf{a}_t^T)\mathbf{M}_t(1, \mathbf{a}_t^T)^T\right) \tag{70}$$

$$\leq \Sigma_{\text{SE}}^{(\tau_1,\tau_2)}\left((1, \mathbf{a}_t^T)\mathbf{M}_t'(1, \mathbf{a}_t^T)^T\right). \tag{71}$$

It follows that the noise term (13) of the $\Sigma_\rho$-dynamics is positive semi-definite and dominated by the respective term of the $\Sigma_{\text{SE}}^{(\tau_1,\tau_2)}$-dynamics. The same holds for the signal terms (12). It follows that $0 \leq \mathbf{M}_{t+1} \leq \mathbf{M}_{t+1}'$, as desired.

The proof of the converse statement is completely analogous. $\square$

## D    PROOF OF THEOREM 2

We will derive a more general, non-stationary version of loss expansion (19), which will imply the simplified stationary version.

Consider the $\mathbf{M}$ matrix iterations (12)–(13), in which we, as assumed, use the SE approximation $\Sigma_{\text{SE}}^{(\tau_1,\tau_2)}$ instead of $\Sigma_\rho$:

$$\mathbf{M}_{t+1} = \left[\begin{pmatrix} 1 & \mathbf{b}_t^T \\ 0 & D_t \end{pmatrix} + \mathbf{H}\begin{pmatrix} -\alpha_t \\ \mathbf{c}_t \end{pmatrix}(1, \mathbf{a}_t^T)\right]\mathbf{M}_t\left[\begin{pmatrix} 1 & \mathbf{b}_t^T \\ 0 & D_t \end{pmatrix} + \mathbf{H}\begin{pmatrix} -\alpha_t \\ \mathbf{c}_t \end{pmatrix}(1, \mathbf{a}_t^T)\right]^T \tag{72}$$

$$+ \frac{1}{|B|}\Sigma_{\text{SE}}^{(\tau_1,\tau_2)}\left((1, \mathbf{a}_t^T)\mathbf{M}_t(1, \mathbf{a}_t^T)^T\right) \otimes \begin{pmatrix} -\alpha_t \\ \mathbf{c}_t \end{pmatrix}(-\alpha_t \quad \mathbf{c}_t), \tag{73}$$

$$\Sigma_{\text{SE}}^{(\tau_1,\tau_2)}(\mathbf{C}) = \tau_1 \operatorname{Tr}[\mathbf{HC}]\mathbf{H} - \tau_2 \mathbf{HCH}. \tag{74}$$

Let us write these iterations in the form

$$\mathbf{M}_{t+1} = F_t \mathbf{M}_t = (A_t + P_t)\mathbf{M}_t, \tag{75}$$

where $F_t, A_t, P_t$ are linear operators defining this transformation:

1. $F_t$ is the full transformation;
2. $P_t$ represents the part of $F_t$ associated with the term $\tau_1 \operatorname{Tr}[\mathbf{HC}]\mathbf{H}$ in SE Eq. (74) as part of the noisy term (73);
3. $A_t$ represents the remaining part of $F_t$, i.e. the noiseless term (72) and the part $\tau_2 \mathbf{HCH}$ of the noisy term (73).

The matrix $\mathbf{M}_t$ has a diagonal component $\mathbf{M}_{t,\text{diag}}$ w.r.t. the operator $\mathbf{H}$ (below we refer to such matrices as $\mathbf{H}$-*diagonal*): given the eigenbasis $\mathbf{e}_k$ of $\mathbf{H}$, we can represent $\mathbf{M}_t = \sum_{k,l}(\mathbf{e}_k \otimes \mathbf{e}_l) \otimes Z_{t,k,l}$ with some matrices $Z_{t,k,l} \in \mathbb{R}^{(1+M)\times(1+M)}$, and then $\mathbf{M}_{t,\text{diag}} = \sum_k(\mathbf{e}_k \otimes \mathbf{e}_k) \otimes Z_{t,k,k}$. Whereas the full matrices $\mathbf{M}_t$ live in the space $(\mathcal{H} \otimes \mathbb{R}^{1+M})^{\otimes 2}$, the matrices $\mathbf{M}_{t,\text{diag}}$ naturally live in the space $\mathcal{H} \otimes (\mathbb{R}^{1+M})^{\otimes 2}$. We will argue now that we need to only keep track of this diagonal component of $\mathbf{M}_t$ during the iterations.

Observe first that the diagonal part $\mathbf{M}_{t,\text{diag}}$ evolves independently of the off-diagonal part of $\mathbf{M}_t$. Indeed, if $\mathbf{M}_t = (\mathbf{e}_k \otimes \mathbf{e}_l) \otimes Z$ with some $Z \in \mathbb{R}^{(1+M)\times(1+M)}$, then also $A_t\mathbf{M}_t = (\mathbf{e}_k \otimes \mathbf{e}_l) \otimes Z'$ with some $Z' \in \mathbb{R}^{(1+M)\times(1+M)}$. On the other hand, $P_t\mathbf{M}_t$ depends on $\mathbf{M}_t$ only through $\mathbf{M}_{t,\text{diag}}$ and produces an $\mathbf{H}$-diagonal result.

Note also that the averaged loss $L_t$ only depends on the diagonal part of $\mathbf{M}_t$:

$$L_t = \tfrac{1}{2}\operatorname{Tr}[\mathbf{HC}_t] = \tfrac{1}{2}\operatorname{Tr}[\mathbf{HC}_{t,\text{diag}}]. \tag{76}$$

It follows that $L_t$ is completely determined by the self-consistent evolution of the diagonal components $\mathbf{M}_{t,\text{diag}}$.

Since $\mathbf{w}_0 = 0, \mathbf{u}_0 = 0$, and accordingly $\Delta\mathbf{w}_0 = -\mathbf{w}_*$, we can then take the initial second moment matrix $\mathbf{M}_0$ to only have a $\mathbf{H}$-diagonal nontrivial component $\mathbf{C}_0$, i.e. $\mathbf{M}_0 = \left(\begin{smallmatrix}\mathbf{C}_0 & 0 \\ 0 & 0\end{smallmatrix}\right)$, where $\mathbf{C}_0 = \operatorname{diag}(c_k^2), c_k^2 = \langle\mathbf{e}_k, \mathbf{w}_*\rangle^2$. Then,

$$L_t = \tfrac{1}{2}\operatorname{Tr}[\mathbf{HC}_t] = \tfrac{1}{2}\langle\boldsymbol{\lambda} \otimes \left(\begin{smallmatrix}1 & 0 \\ 0 & 0\end{smallmatrix}\right), F_t F_{t-1}\cdots F_1\left(\begin{smallmatrix}\mathbf{C}_0 & 0 \\ 0 & 0\end{smallmatrix}\right)\rangle, \tag{77}$$

where $\boldsymbol{\lambda} = \operatorname{diag}(\lambda_1, \lambda_2, \ldots)$ and the inner product is between matrices, $\langle A, B\rangle \equiv \operatorname{Tr}(AB^*)$.

We can now describe the action of the operators $P_t$ and $A_t$ on $\mathbf{H}$-diagonal matrices $\mathbf{M}$. The operator $P_t$ is a rank-1 operator that can be written as

$$P_t = \tfrac{\tau_1}{|B|}\boldsymbol{\lambda} \otimes \left(\begin{smallmatrix}-\alpha_t \\ \mathbf{c}_t\end{smallmatrix}\right)\left(-\alpha_t \; \mathbf{c}_t^T\right)\langle\boldsymbol{\lambda} \otimes \left(\begin{smallmatrix}1 \\ \mathbf{a}_t\end{smallmatrix}\right)\left(1 \; \mathbf{a}_t^T\right), \cdot\rangle. \tag{78}$$

The operator $A_t$ is a system of $(1+M)\times(1+M)$-matrices $A_{t,\lambda}$ acting independently in each $\lambda$-subspace:

$$A_t = (A_{t,\lambda}), \quad A_{t,\lambda}Z = S_{\lambda,t}ZS_{\lambda,t}^T - \tfrac{\tau_2}{|B|}\lambda^2\left(\begin{smallmatrix}-\alpha_t \\ \mathbf{c}_t\end{smallmatrix}\right)\left(\begin{smallmatrix}1 \\ \mathbf{a}_t\end{smallmatrix}\right)^T Z\left(\begin{smallmatrix}1 \\ \mathbf{a}_t\end{smallmatrix}\right)\left(\begin{smallmatrix}-\alpha_t \\ \mathbf{c}_t\end{smallmatrix}\right)^T, \tag{79}$$

with $S_{\lambda,t}$ defined in Eq. (7).

We obtain the desired loss expansion by substituting $F_t = P_t + A_t$ for each $t$ in Eq. (77) and expand the result over various choices of the iterations $t_1, \ldots, t_m$ in which we choose the term $P$ while in the other iterations we choose the term $A$. Since $P_t$ is rank-1, the contribution of each configuration is split into a product of scalar propagators:

$$L_t = \frac{1}{2}\left(V_{t+1} + \sum_{m=1}^{t}\sum_{0 < t_1 < \ldots < t_m < t+1} U_{t+1,t_m}U'_{t_m,t_{m-1}}U'_{t_{m-1},t_{m-2}}\cdots U'_{t_2,t_1}V'_{t_1}\right). \tag{80}$$

Here the *propagators* $U'_{t,s}, U_{t,s}, V'_t, V_t$ have the form

$$U'_{t,s} = \tfrac{\tau_1}{|B|}\langle\boldsymbol{\lambda} \otimes \left(\begin{smallmatrix}1 \\ \mathbf{a}_t\end{smallmatrix}\right)\left(1 \; \mathbf{a}_t^T\right), A_{t-1}A_{t-2}\cdots A_{s+1}[\boldsymbol{\lambda} \otimes \left(\begin{smallmatrix}-\alpha_s \\ \mathbf{c}_s\end{smallmatrix}\right)\left(-\alpha_s \; \mathbf{c}_s^T\right)]\rangle \tag{81}$$

$$= \tfrac{\tau_1}{|B|}\sum_k\lambda_k^2\langle\left(\begin{smallmatrix}1 \\ \mathbf{a}_t\end{smallmatrix}\right)\left(1 \; \mathbf{a}_t^T\right), A_{t-1,\lambda_k}\cdots A_{s+1,\lambda_k}\left(\begin{smallmatrix}-\alpha_s \\ \mathbf{c}_s\end{smallmatrix}\right)\left(-\alpha_s \; \mathbf{c}_s^T\right)\rangle, \tag{82}$$

$$V'_t = \langle\boldsymbol{\lambda} \otimes \left(\begin{smallmatrix}1 \\ \mathbf{a}_t\end{smallmatrix}\right)\left(1 \; \mathbf{a}_t^T\right), A_{t-1}A_{t-2}\cdots A_1\left(\begin{smallmatrix}\mathbf{C}_0 & 0 \\ 0 & 0\end{smallmatrix}\right)\rangle \tag{83}$$

$$= \sum_k\lambda_k c_k^2\langle\left(\begin{smallmatrix}1 \\ \mathbf{a}_t\end{smallmatrix}\right)\left(1 \; \mathbf{a}_t^T\right), A_{t-1,\lambda_k}\cdots A_{1,\lambda_k}\left(\begin{smallmatrix}1 & 0 \\ 0 & 0\end{smallmatrix}\right)\rangle, \tag{84}$$

and $U_{t,s}, V_t$ are obtained from $U'_{t,s}, V'_t$ by replacing $\left(\begin{smallmatrix} 1 \\ \mathbf{a}_t \end{smallmatrix}\right)\left(\begin{smallmatrix} 1 & \mathbf{a}_t^T \end{smallmatrix}\right)$ with $\left(\begin{smallmatrix} 1 & 0 \\ 0 & 0 \end{smallmatrix}\right)$ in the left arguments of scalar products in these formulas. The propagators $U, V$ are different from $U', V'$ in that the former end at the loss evaluation time $t$, whereas the latter end at one of the noise injection times $t_1, \ldots, t_m$. Each term in Eq. (80) contains exactly one signal propagator $V$ or $V'$ and some number of noise propagators $U$ or $U'$.

In the stationary case the above formulas simplify as given in Eqs. (17), (18), (19): we can use a single time index $U'_{t,s} \equiv U'_{t-s}, U_{t,s} \equiv U_{t-s}$, and replace the products $A_{t-1} \ldots A_{s+1}$ by the power $A^{t-s-1}$.

## E  PROPAGATOR IDENTITIES AND BOUNDS

In this section we collect auxiliary results that will be used in the proof of Theorem 3.

### E.1  GENERAL NON-STATIONARY CASE

Recall that in the general non-stationary case the loss expansion is given by Eqs. (80)-(84).

We will derive a convenient representation of the vector of loss values,

$$\mathbf{L} = (L_0, L_1, \ldots), \tag{85}$$

in terms of suitable linear operators. First, consider the values

$$L'_t = \frac{1}{2}\left(V'_{t+1} + \sum_{m=1}^{t} \sum_{0 < t_1 < \ldots < t_m < t+1} U'_{t+1,t_m} U'_{t_m,t_{m-1}} U'_{t_{m-1},t_{m-2}} \cdots U'_{t_2,t_1} V'_{t_1}\right) \tag{86}$$

and the respective vector

$$\mathbf{L}' = (L'_0, L'_1, \ldots)^T \tag{87}$$

Observe that

$$\mathbf{L} = \tfrac{1}{2}\mathbf{V} + \mathbb{U}\mathbf{L}', \tag{88}$$

where

$$\mathbf{V} = (V_1, V_2, \ldots)^T \tag{89}$$

and $\mathbb{U}$ is the infinite matrix

$$(\mathbb{U})_{ts} = \begin{cases} U_{t+1,s+1}, & t > s, \\ 0, & t \le s. \end{cases} \tag{90}$$

Note now that $\mathbf{L}'$ satisfies the equation

$$\mathbf{L}' = \tfrac{1}{2}\mathbf{V}' + \mathbb{U}'\mathbf{L}', \tag{91}$$

where

$$\mathbf{V}' = (V'_1, V'_2, \ldots)^T, \tag{92}$$

$$(\mathbb{U}')_{ts} = \begin{cases} U'_{t+1,s+1}, & t > s, \\ 0, & t \le s. \end{cases} \tag{93}$$

The representation (86) corresponds to the formal solution of Eq. (91):

$$\mathbf{L}' = \tfrac{1}{2}(1 - \mathbb{U}')^{-1}\mathbf{V}' = \tfrac{1}{2}(\mathbf{V}' + \mathbb{U}'\mathbf{V}' + \mathbb{U}'^2\mathbf{V}' + \ldots). \tag{94}$$

When viewed in suitable Banach spaces, these operator representations can be conveniently used to derive upper bounds on $L_t$. Specifically, let $a_t > 0$ be a positive sequence and consider the Banach space of sequences with the norm

$$\|\mathbf{f}\| = \sup_t |a_t f_t|, \quad \mathbf{f} = (f_0, f_1, \ldots). \tag{95}$$

Then

$$\|\mathbb{U}\| = \sup_t \sum_s |U_{t+1,s+1}|\frac{a_t}{a_s}, \tag{96}$$

$$\|\mathbb{U}'\| = \sup_t \sum_s |U'_{t+1,s+1}|\frac{a_t}{a_s}. \tag{97}$$

Assuming $\|\mathbb{U}'\| < 1$, we get

$$\|\mathbf{L}'\| = \tfrac{1}{2}\|(1 - \mathbb{U}')\mathbf{V}'\| \leq \frac{\|\mathbf{V}'\|}{2(1 - \|\mathbb{U}'\|)} \tag{98}$$

and then

$$\|\mathbf{L}\| = \|\tfrac{1}{2}\mathbf{V} + \mathbb{U}\mathbf{L}'\| \leq \tfrac{1}{2}\|\mathbf{V}\| + \|\mathbb{U}\|\|\mathbf{L}'\|. \tag{99}$$

For example, choosing $a_t = (t+1)^\xi, t = 0, 1, \ldots,$ with a constant exponent $\xi \geq 0$, we obtain a simple sufficient condition for the bound $L_t = O(t^{-\xi})$:

**Proposition 4.** *Suppose that*

$$|V_t| \leq C_V t^{-\xi}, \quad t = 1, 2, \ldots \tag{100}$$

$$|V'_t| \leq C_{V'} t^{-\xi}, \quad t = 1, 2, \ldots \tag{101}$$

$$C_U = \sup_{t \geq 1} \sum_{s=1}^{t-1} |U_{t,s}| \left(\frac{t}{s}\right)^\xi < \infty \tag{102}$$

$$C_{U'} = \sup_{t \geq 1} \sum_{s=1}^{t-1} |U'_{t,s}| \left(\frac{t}{s}\right)^\xi < 1 \tag{103}$$

*with some constants $C_V, C_{V'}, C_U, C_{U'}$. Then*

$$L_t \leq C_L (t+1)^{-\xi}, \quad C_L = \tfrac{1}{2}C_V + C_U \frac{C_{V'}}{2(1 - C_{U'})}. \tag{104}$$

In particular, by taking $\xi = 0$ we see that if $\sup_{t \geq 1} \sum_{s=1}^{t-1} |U'_{t,s}| < 1$ and $\sup_{t \geq 1} \sum_{s=1}^{t-1} |U_{t,s}| < \infty$, then uniform boundedness of $V_t, V'_t$ implies uniform boundedness of $L_t$.

### E.2 STATIONARY CASE

Recall the stationary version (19) of the loss expansion:

$$L_t = \frac{1}{2}\Big(V_{t+1} + \sum_{m=1}^{t} \sum_{0 < t_1 < \ldots < t_m < t+1} U_{t+1-t_m} U'_{t_m - t_{m-1}} U'_{t_{m-1} - t_{m-2}} \cdots U'_{t_2 - t_1} V'_{t_1}\Big). \tag{105}$$

It is will be occasionally convenient to slightly adjust the operator formalism of previous section by considering two-sided sequences. We adjust the definitions (85) and (89) of the vectors $\mathbf{L}$ and $\mathbf{V}$ by padding them by 0 to the left:

$$\mathbf{L} = (\ldots, 0, L_0, L_1, \ldots), \tag{106}$$

$$\mathbf{V} = (\ldots, 0, V_1, V_2, \ldots). \tag{107}$$

Similar convention applies to $\mathbf{L}', \mathbf{V}'$.

We extend the definition (90) of operator $\mathbb{U}$ to double-sided sequences, also taking into account that due to the stationarity $\mathbb{U}$ is now translation equivariant:

$$(\mathbb{U})_{ts} = \begin{cases} U_{t-s}, & t > s \\ 0, & t \leq s \end{cases} \equiv (\mathbb{U})_{t-s}. \tag{108}$$

Similar convention applies to $\mathbb{U}'$.

We now establish a useful general result showing that $U_t = O(t^{-\xi_U})$ with $\xi_U > 1$ implies a similar bound for the matrix elements of $(1 - \mathbb{U})^{-1}$. This can be easily done if we assume not just $U_\Sigma < 1$ and $U_t = O(t^{-\xi_U})$ as in Theorem 3, but rather $U_t \leq C't^{-\xi_U}$ with a sufficiently small $C'$:

**Lemma 2.** *Suppose that $U_t = 0$ for $t \leq 0$ and $0 \leq U_t \leq C't^{-\xi_U}$ for $t > 0$ with some $C'$ and $\xi_U > 1$. Then, if $C'$ is sufficiently small, we have $(\mathbb{U}^2)_t \leq \frac{1}{2}C't^{-\xi_U}$ for all $t > 0$. By iterating this bound, we get*

$$((1 - \mathbb{U})^{-1})_t \begin{cases} \leq 2C't^{-\xi_U}, & t > 0 \\ = 1, & t = 0 \\ = 0, & t < 0 \end{cases} \tag{109}$$

*Proof.* We have for $t > 0$

$$(\mathbb{U}^2)_t = \sum_{r=1}^{t-1} U_{t-r} U_r \tag{110}$$

$$\leq 2 \sum_{r=1}^{t/2} C'(t/2)^{-\xi_U} U_r \tag{111}$$

$$\leq C' t^{-\xi_U} 2^{1+\xi_U} \sum_{r=1}^{t/2} U_r \tag{112}$$

$$\leq \Big[ 2^{1+\xi_U} C' \sum_{r=1}^{\infty} r^{-\xi_U} \Big] C' t^{-\xi_U}. \tag{113}$$

The factor $2^{1+\xi_U} C' \sum_{r=1}^{\infty} r^{-\xi_U}$ can be made less than $1/2$ by taking $C'$ small enough.

The relation (109) follows by expanding $(1 - \mathbb{U})^{-1} = 1 + \mathbb{U} + \mathbb{U}^2 + \dots$ and repeatedly applying $\sum_{r=1}^{t-1} [C'(t-r)^{-\xi_U}][C' r^{-\xi_U}] \leq \frac{1}{2} C'(t-r)^{-\xi_U}$. $\qquad\square$

Now we prove a more subtle statement that only uses the weaker assumptions made in Theorem 3.

**Lemma 3.** *Suppose that $U_\Sigma < 1$ and $U_t = O(t^{-\xi})$ for some $\xi > 1$. Then $((1 - \mathbb{U})^{-1})_t = O(t^{-\xi})$.*

*Proof.* The operator $(1 - \mathbb{U})^{-1}$ is the operator of convolution with the kernel $((1 - U)^{-1})_t$ which can be expanded as

$$((1 - U)^{-1})_t = \delta_t + U_t + (U * U)_t + \dots, \tag{114}$$

where $*$ denotes the convolution of two sequences, i.e.

$$(A * B)_t = \sum_{r=0}^{t} A_{t-r} B_r, \quad r = 0, 1, \dots \tag{115}$$

for any one-sided sequences $A, B$. Let us split the sequence $U_t$ into a finite initial segment $A_t$ and a remainder $B_t$ with a small sum:

$$U = A + B, \tag{116}$$
$$A_t = U_t \mathbf{1}_{t \leq t_0}, \tag{117}$$
$$B_t = U_t \mathbf{1}_{t > t_0}, \tag{118}$$
$$\tag{119}$$

where $t_0$ is such that

$$\sum_t B_t < \epsilon \tag{120}$$

with a sufficiently small $\epsilon$ to be chosen later. We then have

$$(1 - U)^{-1} = \sum_{n=0}^{\infty} U^{*n} = \sum_{n=0}^{\infty} \sum_{k=0}^{n} \binom{n}{k} A^{*k} * B^{*(n-k)}, \tag{121}$$

where the binomial expansion is legitimate due to the commutativity and associativity of convolution. Consider the convolutional power $B^{*r}$. Recall that $B_t = O(t^{-\xi})$. We can now argue as in Lemma 2 and show by induction that for any $r = 1, 2, \dots$

$$(B^{*r})_t \leq C \epsilon_1^{r-1} t^{-\xi} \tag{122}$$

with $\epsilon_1$ that can be made arbitrarily small by making $\epsilon$ in Eq. (120) small enough. Indeed, for $r \geq 2$

$$(B^{*r})_t = \sum_{s=1}^{t-1} (B^{*(r-1)})_s B_{t-s} \tag{123}$$

$$= \sum_{s=1}^{t/2} (B^{*(r-1)})_s B_{t-s} + \sum_{s=t/2+1}^{t-1} (B^{*(r-1)})_s B_{t-s} \tag{124}$$

$$\leq \Big( \sum_{s=1}^{\infty} (B^{*(r-1)})_s \Big) C(t/2)^{-\xi} + C\epsilon_1^{r-2} (t/2)^{-\xi} \sum_{s=1}^{\infty} B_s \tag{125}$$

$$\leq \epsilon_1^{r-1} C(t/2)^{-\xi} + C\epsilon_1^{r-2} (t/2)^{-\xi} \epsilon \tag{126}$$

$$= C\epsilon_1^{r-1} t^{-\xi} ((\epsilon/\epsilon_1)^{r-1} 2^{\xi} + (\epsilon/\epsilon_1) 2^{\xi}) \tag{127}$$

$$\leq C\epsilon_1^{r-1} t^{-\xi} \tag{128}$$

by choosing $\epsilon$ so that $(\epsilon/\epsilon_1)^{r-1} 2^{\xi} + (\epsilon/\epsilon_1) 2^{\xi} < 1$ for $r \geq 2$. This proves (122).

Now let

$$a = \sum_t A_t. \tag{129}$$

By assumption, $a \leq \sum_t U_t < 1$. Observe that $(A^{*k})_t = 0$ for $t > kt_0$. It follows that

$$(A^{*k} * B^{*(n-k)})_t \leq \Big( \sum_{s=1}^{kt_0} (A^{*k})_s \Big) C\epsilon_1^{n-k-1} \max(1, t - kt_0)^{-\xi} \tag{130}$$

$$\leq a^k C\epsilon_1^{n-k-1} \max(1, t - kt_0)^{-\xi}. \tag{131}$$

Then from the binomial expansion (121)

$$((1 - \mathbb{U})^{-1} - 1)_t \leq \sum_{n=1}^{\infty} \Big( \sum_{k=0}^{n-1} \binom{n}{k} a^k C\epsilon_1^{n-1-k} \max(1, t - kt_0)^{-\xi} + a^n \mathbf{1}_{t \leq nt_0} \Big) \tag{132}$$

$$= \sum_{n=1}^{t/(2t_0)} + \sum_{n=t/(2t_0)+1}^{\infty} \tag{133}$$

$$\leq (t/2)^{-\xi} \sum_{n=1}^{t/(2t_0)} \sum_{k=0}^{n} \binom{n}{k} a^k C\epsilon_1^{n-1-k} + \sum_{n=t/(2t_0)+1}^{\infty} \sum_{k=0}^{n} \binom{n}{k} a^k C\epsilon_1^{n-1-k} \tag{134}$$

$$= (t/2)^{-\xi} C \sum_{n=1}^{t/(2t_0)} (a + \epsilon_1)^n + C \sum_{n=t/(2t_0)+1}^{\infty} (a + \epsilon_1)^n \tag{135}$$

$$\leq C(1 - (a + \epsilon_1))^{-1} ((t/2)^{-\xi} + (a + \epsilon_1)^{t/(2t_0)+1}). \tag{136}$$

By choosing $\epsilon_1$ sufficiently small so that $a + \epsilon_1 < 1$ we get the desired property $((1 - \mathbb{U})^{-1})_t = O(t^{-\xi})$. $\qquad\square$

## F  PROOF OF THEOREM 3

Throughout this section we will use the stationary-case representation of $L_t$ from Eq. (19), with the identified $U_t = U'_t, V_t = V'_t$:

$$L_t = \frac{1}{2} \Big( V_{t+1} + \sum_{m=1}^{t} \sum_{0 < t_1 < ... < t_m < t+1} U_{t+1-t_m} U_{t_m-t_{m-1}} U_{t_{m-1}-t_{m-2}} \cdots U_{t_2-t_1} V_{t_1} \Big). \tag{137}$$

We use the operator formalism and results from Section E.

## F.1 PART 1 (CONVERGENCE)

If $U_\Sigma < 1$, then $V_t = O(1)$ implies $L_t = O(1)$ by the remark after Proposition 4.

To conclude $L_t = o(1)$ from $V_t = o(1)$, write

$$L_t = \frac{1}{2} \sum_{s=1}^{t} ((1 - \mathbb{U})^{-1})_s V_{t-s+1}. \tag{138}$$

Here $((1 - \mathbb{U})^{-1})_s \geq 0$ and

$$\sum_{s=1}^{\infty} ((1 - \mathbb{U})^{-1})_s = \frac{1}{1 - U_\Sigma} < \infty. \tag{139}$$

For any $\epsilon > 0$ we can choose $r$ such that $\sum_{s=r}^{\infty} ((1 - \mathbb{U})^{-1})_s < \epsilon$. Then by Eq. (138)

$$L_t \leq \frac{\max_{s=1,\dots,r-1} V_{t-s+1}}{2(1 - U_\Sigma)} + \frac{\epsilon \max_s V_s}{2} \xrightarrow{t \to \infty} \frac{\epsilon \max_s V_s}{2}. \tag{140}$$

It follows that $L_t \to 0$.

## F.2 PART 2 (DIVERGENCE)

Since $U_t, V_t \geq 0$, we can lower bound, for any $k \geq 0$ and $s \geq 0$,

$$L_{t+s} \geq (\mathbb{U}^k)_t V_{s+1}. \tag{141}$$

We choose $s$ so that $V_{s+1} > 0$. Since $U_\Sigma > 1$, we can choose $t_0$ such that

$$\sum_{t=1}^{t_0} U_t = a > 1. \tag{142}$$

Consider the respective truncated sequence $A_t$:

$$A_t = \begin{cases} U_t. & t \leq t_0 \\ 0, & t > t_0. \end{cases} \tag{143}$$

It follows from Eq. (141) that

$$L_{t+s} \geq (A^{*k})_t V_{s+1}. \tag{144}$$

Note that $(A^{*k})_t = 0$ for $t > kt_0$ and $\sum_t (A^{*k})_t = (\sum_t A_t)^k = a^k$. It follows that

$$\sum_{t=s}^{kt_0+s} L_t \geq a^k V_{s+1}. \tag{145}$$

Hence

$$\sup_{t=0,1,\dots} L_t \geq \sup_{k \geq 0} \frac{a^k V_{s+1}}{kt_0 + s} = \infty. \tag{146}$$

## F.3 PART 3 (SIGNAL-DOMINATED REGIME)

The idea of the proof is that in the signal dominated case we expect the main contribution to $L_t$ to come from the terms with large time $t_1$ of the $V$-factor and small time increments of the other, $U$-factors. We then define the potential leading term $L_t^{(V)}$ by first replacing $V_{t_1}$ by $V_{t+1}$ in Eq. (137),

$$L_t^{(V,0)} = \frac{1}{2} \Big( \sum_{m=0}^{t} \sum_{0 < t_1 < \dots < t_m < t+1} U_{T+1-t_m} U_{t_m - t_{m-1}} U_{t_{m-1} - t_{m-2}} \cdots U_{t_2 - t_1} \Big) V_{t+1}, \tag{147}$$

and then extending summation to negative $t$:

$$L_t^{(V)} = \frac{1}{2}\Big(\sum_{m=0}^{\infty} \sum_{-\infty < t_1 < \ldots < t_m < t+1} U_{t+1-t_m} U_{t_m - t_{m-1}} U_{t_{m-1} - t_{m-2}} \cdots U_{t_2 - t_1}\Big) V_{t+1} \tag{148}$$

$$= \frac{1}{2}\Big(\sum_{m=0}^{\infty} \sum_{s_1,\ldots,s_m=1}^{\infty} U_{s_m} U_{s_{m-1}} \cdots U_{s_1}\Big) V_{t+1} \tag{149}$$

$$= \frac{V_{t+1}}{2(1 - U_\Sigma)} \tag{150}$$

$$= \frac{C_V}{2(1 - U_\Sigma)}(t+1)^{-\xi_V}(1 + o(1)). \tag{151}$$

Thus, to establish the desired asymptotics, it is sufficient to show that the differences $L_t^{(V)} - L_t^{(V,0)}$ and $L_t^{(V,0)} - L_t$ are sub-leading compared to the leading term $L_t^{(V)}$.

**The difference $L_t^{(V)} - L_t^{(V,0)}$.** In terms of the operator $\mathbb{U}$ we can write the vector representing the difference $L_t^{(V)} - L_t^{(V,0)}$ as

$$\mathbf{L}^{(V)} - \mathbf{L}^{(V,0)} = \tfrac{1}{2}\mathbf{V} \odot (1 - \mathbb{U})^{-1}\mathbf{1}_{t<0}, \tag{152}$$

where $\mathbf{1}_{t<0}(t) = 0$ if $t \geq 0$ and 1 otherwise, and $\odot$ is the pointwise product. Consider the norm (95) defined with

$$a_t = \begin{cases} 1, & t \leq 0 \\ (t+1)^\xi, & t \geq 0 \end{cases} \tag{153}$$

with some $\xi > 0$ to be chosen later. Then if we check that $\|\mathbb{U}\| < 1$, then $((1 - \mathbb{U})^{-1}\mathbf{1}_{t<0})_t = O(t^{-\xi})$ as $t \to +\infty$, in particular implying that

$$L_t^{(V)} - L_t^{(V,0)} = V_t \cdot o(1) = o(L_t^{(V)}), \tag{154}$$

as desired. We have

$$\|\mathbb{U}\| = \sup_t \sum_s |U_{t-s}|\frac{|a_t|}{|a_s|} \tag{155}$$

$$\leq \sum_{r>0} U_r \sup_t \frac{a_t}{a_{t-r}} \tag{156}$$

$$\leq \sum_{r>0} U_r (r+1)^\xi. \tag{157}$$

By assumption, $U_r = O(r^{-\xi_U})$ with some $\xi_U > 1$. Then, by the dominated convergence theorem,

$$\lim_{\xi \searrow 0} \sum_{r>0} U_r (r+1)^\xi = \sum_{r>0} U_r < 1, \tag{158}$$

by the assumption $U_\Sigma < 1$. Therefore, by choosing $\xi$ small enough, we can ensure $\|\mathbb{U}\| < 1$, as desired.

**The difference $L_t^{(V,0)} - L_t$.** We start by writing

$$L_t - L_t^{(V,0)} = \frac{1}{2}\sum_{m=1}^{t} \sum_{0 < t_1 < \ldots < t_m < t+1} U_{t+1-t_m} U_{t_m - t_{m-1}} \cdots U_{t_2 - t_1}(V_{t_1} - V_{t+1}) \tag{159}$$

or equivalently

$$\mathbf{L} - \mathbf{L}_V^{(V,0)} = \tfrac{1}{2}(1 - \mathbb{U})^{-1}\mathbf{V} - \tfrac{1}{2}\mathbf{V} \odot (1 - \mathbb{U})^{-1}\mathbf{1}_{t\geq 0}. \tag{160}$$

Let $k_t$ be any integer sequence such that $k_t \to \infty$ and $k_t = o(t)$. Using $V_t = C_V t^{-\xi_V}(1 + o(1))$ and Lemma 3, we have

$$2|L_t^{(V,0)} - L_t| \leq \sum_{s=1}^{t} ((1 - \mathbb{U})^{-1})_s |V_{t+1-s} - V_{t+1}| \tag{161}$$

$$= \sum_{s=1}^{k_t} + \sum_{s=k_t+1}^{t} \tag{162}$$

$$\leq \sum_{s=1}^{k_t} ((1 - \mathbb{U})^{-1})_s o(t^{-\xi_V}) + \sum_{s=k_t+1}^{t} O(s^{-\xi_U}) O((1 + t - s)^{-\xi_V}) \tag{163}$$

$$= o(t^{-\xi_V}) + \sum_{s=k_t}^{t/2} O(s^{-\xi_U}) O((t-s)^{-\xi_V}) + \sum_{s=t/2+1}^{t-1} O(s^{-\xi_U}) O((t-s)^{-\xi_V}) \tag{164}$$

$$= o(t^{-\xi_V}) + o(1) O(t^{-\xi_V}) + O(t^{-\xi_U}) \sum_{s=t/2+1}^{t-1} O((t-s)^{-\xi_V}) \tag{165}$$

$$= o(t^{-\xi_V}) + O(t^{-\xi_U}) \begin{cases} O(1), & \xi_V > 1 \\ O(\ln t), & \xi_V = 1 \\ O(t^{1-\xi_V}), & \xi_V < 1 \end{cases} \tag{166}$$

$$= o(t^{-\xi_V}) + \begin{cases} O(t^{-\xi_U}), & \xi_V > 1 \\ O(t^{-\xi_U} \ln t), & \xi_V = 1 \\ O(t^{(1-\xi_U)-\xi_V}), & \xi_V < 1 \end{cases} \tag{167}$$

$$= o(t^{-\xi_V}), \tag{168}$$

where we used the assumptions $\xi_V < \xi_U$ and $\xi_U > 1$.

Having established Eqs. (154), (168), we have completed the proof that

$$L_t = L_t^{(V)}(1 + o(1)) = \frac{C_V}{2(1 - U_\Sigma)} t^{-\xi_V}(1 + o(1)). \tag{169}$$

### F.4 PART 4 (NOISE-DOMINATED REGIME)

We transform the original loss $L_T$ to an approximation $L_T^{(U)}$ through the following three steps:

1. Isolate in $L_T$ a long factor $U_{t_{m_1+1}-t_{m_1}}$ of length $T(1 - o(1))$:

$$L_T^{(U,0)} = \frac{1}{2} \sum_{\substack{m_1=1 \\ m_2=0}}^{\infty} \sum_{\substack{0<t_1<...<t_{m_1}<T_1(T) \\ T-T_1(T)<t_{m_1+1}<...<t_{m_1+m_2}<T+1}} (U_{T+1-t_{m_1+m_2}} \cdots U_{t_{m_1+2}-t_{m_1+1}}) \tag{170}$$

$$\times U_{t_{m_1+1}-t_{m_1}}(U_{t_{m_1}-t_{m_1-1}} \cdots U_{t_2-t_1}) V_{t_1} \tag{171}$$

$$= \frac{1}{2} \sum_{\substack{0<t_1<t<T_1(T) \\ T-T_1(T)<s<T+1}} ((1 - \mathbb{U})^{-1})_{T+1-s} U_{s-t} ((1 - \mathbb{U})^{-1})_{t-t_1} V_{t_1}, \tag{172}$$

where $T_1(T) = T^{\frac{\xi_U + \xi_V}{2\xi_V}}$ so that in particular $T_1(T) \to \infty$ and $T_1(T) = o(T)$, due to $\xi_V > \xi_U$.

2. Replace in $L_T^{(U,0)}$ the long factor $U_{t_{m_1+1}-t_{m_1}}$ by $U_T$:

$$L_T^{(U,1)} = \frac{1}{2} \sum_{\substack{0<t_1<t<T_1(T) \\ T-T_1(T)<s<T+1}} ((1 - \mathbb{U})^{-1})_{T+1-s} U_T ((1 - \mathbb{U})^{-1})_{t-t_1} V_{t_1}. \tag{173}$$

3. Finally, extend summation in $L_T^{(U,1)}$, dropping the upper constraints on $t_1, t$ and the lower constraints on $s$:

$$L_T^{(U)} = \frac{1}{2} \sum_{\substack{0 < t_1 < t < \infty \\ -\infty < s < T+1}} ((1 - \mathbb{U})^{-1})_{T+1-s} U_T ((1 - \mathbb{U})^{-1})_{t-t_1} V_{t_1} \tag{174}$$

$$= \frac{V_\Sigma U_T}{2(1 - U_\Sigma)^2} \tag{175}$$

$$= \frac{V_\Sigma C_U}{2(1 - U_\Sigma)^2} (T + 1)^{-\xi_U} (1 + o(1)). \tag{176}$$

We argue now that each of these approximations introduces an error of order $o(T^{-\xi_U})$.

1. $[L \to L^{(U,0)}]$. Note that the difference $L_T - L_T^{(U,0)}$ consists of the terms where either $t_1 > T_1(T)$, or there exists $k > 1$ for which $T_1(T) < t_k < T - T_1(T)$. Let us bound the contribution of such terms.

   In the first case, the contribution is bounded by $O((T_1(T))^{-\xi_V}) = O(T^{-\frac{\xi_U + \xi_V}{2}}) = o(T^{-\xi_U})$, since $\xi_V > \xi_U$.

   In the second case using $((1 - \mathbb{U})^{-1})_t = O(t^{-\xi_U})$ and $V_t = O(t^{-\xi_U})$, the contribution of these terms can be upper bounded by

$$\sum_{t_k = T_1(T)}^{T - T_1(T)} O((T - t_k)^{-\xi_U}) O(t_k^{-\xi_U}) = O(T^{-\xi_U}) \sum_{t_k = T_1(T)}^{T/2} O(t_k^{-\xi_U}) \tag{177}$$

$$= O(T^{-\xi_U}) O((T_1(T))^{1-\xi_U} - T^{1-\xi_U}) \tag{178}$$

$$= o(T^{-\xi_U}), \tag{179}$$

   since $\xi_U > 1$.

2. $[L^{(U,0)} \to L^{(U,1)}]$. Since $s - t = T(1 + o(1))$, we have $U_{s-t} - U_T = o(T^{-\xi_U})$ and then also $|L_T^{(U,0)} - L_T^{(U,1)}| = o(T^{-\xi_U})$ by the boundedness of the summed products of the remaining factors.

3. $[L^{(U,1)} \to L^{(U)}]$. Here one can use similar arguments, taking into account the relation $U_T = O(T^{-\xi_U})$ and the tail bound

$$\sum_{t > T_1} ((1 - \mathbb{U})^{-1})_t = O(T_1^{1-\xi_U}), \tag{180}$$

   following from Lemma 3.

This completes the proof of this part of Theorem 3.

# G   GENERALIZATION OF THEOREM 3 TO $V_t \neq V_t', U_t \neq U_t'$

In this section we drop the assumption $V_t = V_t', U_t = U_t'$ adopted for Theorem 3. Accordingly, $V_t, U_t, L_t, U_\Sigma, V_\Sigma$ are now distinct from $V_t', U_t', L_t', U_\Sigma', V_\Sigma'$. The respective generalized version of Theorem 3 reads

**Theorem 8.** *Suppose that the numbers $L_t$ are given by expansion* (19) *with some numbers $U_t, U_t', V_t, V_t' \geq 0$.*

1. *[**Convergence**] Let $U_\Sigma' < 1$ and $U_\Sigma < \infty$. At $t \to \infty$, if $V_t, V_t' = O(1)$ (respectively, $V_t, V_t' = o(1)$), then also $L_t = O(1)$ (respectively, $L_t = o(1)$).*

2. *[**Divergence**] If $U_\Sigma' > 1$, $V_t' > 0$ at least for one $t$ and $U_t > 0$ at least for one $t$, then $\sup_{t=1,2,\dots} L_t = \infty$.*

3. **[Signal-dominated regime]** *Suppose that there exist constants $\xi_V, C_V > 0$ such that $V_t, V_t' = C_V t^{-\xi_V}(1 + o(1))$ as $t \to \infty$. Suppose also that $U_\Sigma' < 1$ and $U_t, U_t' = O(t^{-\xi_U})$ with some $\xi_U > \max(\xi_V, 1)$. Then*

$$L_t = \frac{(1 - U_\Sigma' + U_\Sigma)C_V}{2(1 - U_\Sigma')} t^{-\xi_V}(1 + o(1)). \tag{181}$$

4. **[Noise-dominated regime]** *Suppose that there exist constants $\xi_V > \xi_U > 1, C_U > 0$ such that $U_t, U_t' = C_U t^{-\xi_U}(1 + o(1))$ and $V_t, V_t' = O(t^{-\xi_V})$ as $t \to \infty$. Suppose also that $U_\Sigma' < 1$. Then*

$$L_t = \frac{(1 - U_\Sigma' + U_\Sigma)V_\Sigma' C_U}{2(1 - U_\Sigma')^2} t^{-\xi_U}(1 + o(1)). \tag{182}$$

Note that the only essential difference in the conclusions of Theorems 3 and 8 is the factor $1 - U_\Sigma' + U_\Sigma$ appearing in the leading terms. In the case $U_\Sigma = U_\Sigma'$ this factor evaluates to 1.

We remark also that theorem 8 assumes in item 3 that the quantities $V_t$ and $V_t'$, though generally different, have the same leading terms $C_V t^{-\xi_V}$; in particular, with the same constant $C_V$ rather than two different constants $C_V, C_{V'}$. This assumption is not important and is made simply because it is fulfilled by the actual $V_t, V_t'$ as analyzed in Theorem 5. The same remark applies to $U_t, U_t'$ in item 4.

*Proof of Theorem 8.* 1. [Convergence] The $O(1)$ part of the claim follows from the remark after Proposition 4.

For the $o(1)$ part, we already know from Theorem 3 that $U_\Sigma' < 1$ and $V_t' = o(1)$ imply $L_t' = o(1)$. Using the relation

$$L_t = \frac{1}{2}\left(V_{t+1} + \sum_{s=1}^{t} U_s L_{t-s}'\right) \tag{183}$$

and the conditions $V_t = o(1)$, $U_\Sigma < \infty$, we then get $L_t = o(1)$.

2. [Divergence] We already know from Theorem 3 that $\sup_t L_t' = \infty$. Since, for any $s > 0$, $L_t \geq U_s L_{t-s}'$, and $U_s > 0$ at least for one $s$, we also have $\sup_t L_t = \infty$.

3. [Signal-dominated regime] We already know by Theorem 3 that

$$L_t' = \frac{C_V}{2(1 - U_\Sigma')} t^{-\xi_V}(1 + o(1)). \tag{184}$$

Let $\gamma = \frac{\xi_U + \max(\xi_V, 1)}{2\xi_U} \in (0, 1)$ and observe that

$$L_t = \frac{1}{2}V_{t+1} + \sum_{s=0}^{t-1} U_{t-s} L_s' \tag{185}$$

$$= \frac{C_V}{2} t^{-\xi_V}(1 + o(1)) + \sum_{s=0}^{t-t^\gamma} U_{t-s} L_s' + \sum_{s=t-t^\gamma+1}^{t} U_{t-s} L_s' \tag{186}$$

$$= \frac{C_V}{2} t^{-\xi_V}(1 + o(1)) + O\left(t^{-\gamma\xi_U}\sum_{s=1}^{t} s^{-\xi_V}\right) + \frac{C_V}{2(1 - U_\Sigma')} t^{-\xi_V}(1 + o(1)) \sum_{s=t-t^\gamma+1}^{t} U_{t-s} \tag{187}$$

$$= \frac{1}{2}\left(C_V + \frac{C_V U_\Sigma}{1 - U_\Sigma'}\right) t^{-\xi_V}(1 + o(1)) + O(t^{-\frac{\xi_U + \max(\xi_V, 1)}{2}}) \begin{cases} O(1), & \xi_V > 1 \\ O(\ln t), & \xi_V = 1 \\ O(t^{1-\xi_V}), & \xi_V < 1 \end{cases} \tag{188}$$

$$= \frac{1}{2}\left(C_V + \frac{C_V U_\Sigma}{1 - U_\Sigma'}\right) t^{-\xi_V}(1 + o(1)) \tag{189}$$

as claimed.

4. [Noise-dominated regime] We know that, by Theorem 3,

$$L'_t = \frac{V'_\Sigma C_U}{2(1 - U'_\Sigma)^2} t^{-\xi_U}(1 + o(1)). \tag{190}$$

We have

$$L_t = \frac{1}{2}V_{t+1} + \sum_{s=0}^{t-1} U_{t-s}L'_s = \sum_{s=0}^{t-1} U_{t-s}L'_s + o(t^{-\xi_U}). \tag{191}$$

Arguing as earlier, the leading contribution to the sum $\sum_{s=0}^{t-1} U_{t-s}L'_s$ comes from terms with $s$ close to 0 or $t$, since the terms in the middle are $O(t^{-2\xi_U})$. It follows that

$$L_t = \left[C_U \sum_{s=0}^{\infty} L'_s + \frac{V'_\Sigma C_U}{2(1 - U'_\Sigma)^2} U_\Sigma\right] t^{-\xi_U}(1 + o(1)). \tag{192}$$

We can compute

$$\sum_{s=0}^{\infty} L'_s = \frac{1}{2}\sum_{s=0}^{\infty}\sum_{t=1}^{s}(1 - \mathbb{U}')_{s-t}^{-1}V'_t = \frac{1}{2}\sum_{s=0}^{\infty}(1 - \mathbb{U}')_s^{-1}\sum_{t=1}^{\infty}V'_t = \frac{V'_\Sigma}{2(1 - U'_\Sigma)}. \tag{193}$$

Substituting in (192),

$$L_t = \left[C_U \frac{V'_\Sigma}{2(1 - U'_\Sigma)} + \frac{V'_\Sigma C_U}{2(1 - U'_\Sigma)^2} U_\Sigma\right] t^{-\xi_U}(1 + o(1)) \tag{194}$$

$$= \frac{(1 - U'_\Sigma + U_\Sigma)V'_\Sigma C_U}{2(1 - U'_\Sigma)^2} t^{-\xi_U}(1 + o(1)). \tag{195}$$

$\square$

## H  PROOF OF THEOREM 4

Recall that if $B, \Delta B$ are linear operators and $B$ has a simple eigenvalue $\mu_0$ associated with a one-dimensional eigenprojector $P$, then for small $\Delta B$ the operator $B + \Delta B$ has a simple eigenvalue $\mu$ given by

$$\mu = \mu_0 + \text{Tr}[P\Delta B] + O(\|\Delta B\|^2). \tag{196}$$

We apply this to $B = A_{\lambda=0}$ and $\Delta B = A_\lambda - A_{\lambda=0}$ with $A_\lambda$ given by (16). Observe that the noisy term in Eq. (16) is $O(\lambda^2)$, so it will only contribute a correction $O(\lambda^2)$ to $\mu_\lambda$. On the other hand, the main term in $A_\lambda$, given by $\mathbf{M} \mapsto S_\lambda \mathbf{M} S_\lambda^T$, acts as a tensor square $S_\lambda \otimes S_\lambda$. Accordingly, the eigenvalue $\mu_{A,\lambda}$ of $A_\lambda$ can be written as

$$\mu_{A,\lambda} = z_\lambda^2 + O(\lambda^2), \tag{197}$$

where $z_\lambda$ is the leading eigenvalue of the matrix $S_\lambda$. The leading linear term in $z_\lambda$ can be found, for example, from the condition

$$\chi(z_\lambda, \lambda) \equiv 0, \tag{198}$$

where $\chi(x, \lambda)$ is the characteristic polynomial of $S_\lambda$ defined in Eq. (29). Differentiating condition (198) over $\lambda$, we get

$$\frac{\partial\chi}{\partial x}\frac{dz_\lambda}{d\lambda} + \frac{\partial\chi}{\partial\lambda} = 0. \tag{199}$$

Using representation (36) of $\chi$ and taking into account that $z_{\lambda=0} = 1$, this implies

$$\frac{dz_\lambda}{d\lambda}(\lambda = 0) = -\left(\frac{\partial\chi}{\partial\lambda}\Big/\frac{\partial\chi}{\partial x}\right)(\lambda = 0) = \frac{\sum_{m=0}^{M} q_m}{\sum_{m=0}^{M} mp_m - M - 1} = -\alpha_{\text{eff}}, \tag{200}$$

where $\alpha_{\text{eff}}$ is given by the second equality in (22). To see that it is equivalent to the first equality $\alpha_{\text{eff}} = \alpha - \mathbf{b}^T(1-D)^{-1}\mathbf{c}$, note from Eq. (35) that

$$\frac{\partial\chi(x,\lambda)}{\partial x}(x=1,\lambda=0) = \det(1-D), \tag{201}$$

$$\frac{\partial\chi(x,\lambda)}{\partial\lambda}(x=1,\lambda=0) = -\det\begin{pmatrix} \mathbf{a}^T\mathbf{c}-\alpha & \mathbf{a}^T(1-D)-\mathbf{b}^T \\ \mathbf{c} & 1-D \end{pmatrix} \tag{202}$$

$$= \det\begin{pmatrix} \alpha & \mathbf{b}^T \\ \mathbf{c} & 1-D \end{pmatrix} \tag{203}$$

$$= \det(1-D)\det\begin{pmatrix} \alpha & \mathbf{b}^T \\ (1-D)^{-1}\mathbf{c} & I \end{pmatrix} \tag{204}$$

$$= \det(1-D)(\alpha - \mathbf{b}^T(1-D)^{-1}\mathbf{c}). \tag{205}$$

We conclude that $z_\lambda = 1 - \alpha_{\text{eff}}\lambda + O(\lambda^2)$ and hence $\mu_{A,\lambda} = 1 - 2\alpha_{\text{eff}}\lambda + O(\lambda^2)$, as claimed.

If $\alpha_{\text{eff}} > 0$, then for small $\lambda$ the leading eigenvalue $|\mu_{A,\lambda}| < 1$, and so $A_\lambda$ are strictly stable.

## I   PROOF OF THEOREM 5

**Asymptotics of $V_t, V_t'$.**   Recall that

$$V_t' = \sum_k \lambda_k c_k^2 \langle \left(\begin{smallmatrix}1\\\mathbf{a}\end{smallmatrix}\right)\left(\begin{smallmatrix}1\\\mathbf{a}\end{smallmatrix}\right)^T, A_{\lambda_k}^{t-1}\left(\begin{smallmatrix}1&0\\0&0\end{smallmatrix}\right)\rangle. \tag{206}$$

Recall by Theorem 4 that for sufficiently small $\lambda > 0$, say for $\lambda < \lambda_*$, the maps $A_\lambda$ have a simple leading eigenvalue

$$\mu_{A,\lambda} = 1 - 2\alpha_{\text{eff}}\lambda + O(\lambda^2). \tag{207}$$

Let $P_\lambda : \mathbb{R}^{(M+1)\times(M+1)} \to \mathbb{R}^{(M+1)\times(M+1)}$ denote the corresponding one-dimensional spectral projector. Then, by the assumption of strict stability of $A_\lambda$,

$$V_t' = \sum_{k:\lambda_k<\lambda_*} \lambda_k c_k^2 \langle \left(\begin{smallmatrix}1\\\mathbf{a}\end{smallmatrix}\right)\left(\begin{smallmatrix}1\\\mathbf{a}\end{smallmatrix}\right)^T, P_{\lambda_k}\left(\begin{smallmatrix}1&0\\0&0\end{smallmatrix}\right)\rangle\mu_{A,\lambda_k}^{t-1} + O(r^t), \quad t \to \infty, \tag{208}$$

for some $0 < r < 1$. The second term will be asymptotically negligible, so we can focus on the first one.

Next, observe that the projector $P_\lambda$ continuously depends on $\lambda$. At $\lambda = 0$ the matrix $\mathbf{M}_0 = \left(\begin{smallmatrix}1&0\\0&0\end{smallmatrix}\right)$ is precisely the leading eigenvector of $A_{\lambda=0}$, so

$$\langle \left(\begin{smallmatrix}1\\\mathbf{a}\end{smallmatrix}\right)\left(\begin{smallmatrix}1\\\mathbf{a}\end{smallmatrix}\right)^T, P_\lambda\left(\begin{smallmatrix}1&0\\0&0\end{smallmatrix}\right)\rangle = 1 + o(1), \quad \lambda \to 0. \tag{209}$$

Since for any fixed $\widetilde{\lambda} > 0$ the contribution of the terms with $\lambda_k > \widetilde{\lambda}$ to the expansion (208) is exponentially small, we get

$$V_t' = (1 + o(1)) \sum_{k:\lambda_k<\lambda_*} \lambda_k c_k^2 \mu_{A,\lambda_k}^{t-1} + O(r^t), \quad t \to \infty. \tag{210}$$

Given power-law spectral assumptions (1) and relation (207), the asymptotic of $\sum_{k:\lambda_k<\lambda_*} \lambda_k c_k^2 \mu_{A,\lambda_k}^{t-1}$ is derived by standard methods (see e.g. Theorem 1 in Appendix B of Velikanov & Yarotsky (2021)):

$$\sum_{k:\lambda_k<\lambda_*} \lambda_k c_k^2 \mu_{A,\lambda_k}^{t-1} = (1 + o(1))Q\Gamma(\zeta+1)(2\alpha_{\text{eff}}t)^{-\zeta}. \tag{211}$$

We conclude that

$$V_t', V_t = (1 + o(1))Q\Gamma(\zeta+1)(2\alpha_{\text{eff}}t)^{-\zeta}, \tag{212}$$

as claimed.

**Asymptotics of $U_t, U_t'$.** We have

$$U_t' = \frac{\tau_1}{|B|} \sum_k \lambda_k^2 \langle (\begin{smallmatrix} 1 \\ \mathbf{a} \end{smallmatrix})(\begin{smallmatrix} 1 \\ \mathbf{a} \end{smallmatrix})^T, A_{\lambda_k}^{t-1}(\begin{smallmatrix} -\alpha \\ \mathbf{c} \end{smallmatrix})(\begin{smallmatrix} -\alpha \\ \mathbf{c} \end{smallmatrix})^T \rangle. \tag{213}$$

We argue as in the previous paragraph, but the difference now is that the matrix $(\begin{smallmatrix} -\alpha \\ \mathbf{c} \end{smallmatrix})(\begin{smallmatrix} -\alpha \\ \mathbf{c} \end{smallmatrix})^T$ is not an eigenvector of $A_{\lambda=0}$, so we need to find the projection $P_{\lambda=0}\big[(\begin{smallmatrix} -\alpha \\ \mathbf{c} \end{smallmatrix})(\begin{smallmatrix} -\alpha \\ \mathbf{c} \end{smallmatrix})^T\big]$. At $\lambda = 0$, this can be factored as

$$P_{\lambda=0}\big[(\begin{smallmatrix} -\alpha \\ \mathbf{c} \end{smallmatrix})(\begin{smallmatrix} -\alpha \\ \mathbf{c} \end{smallmatrix})^T\big] = \big[P_{S_0}(\begin{smallmatrix} -\alpha \\ \mathbf{c} \end{smallmatrix})\big]\big[P_{S_0}(\begin{smallmatrix} -\alpha \\ \mathbf{c} \end{smallmatrix})\big]^T, \tag{214}$$

where $P_{S_0}$ is the spectral projector for the matrix $S_{\lambda=0}$ and its eigenvalue 1. The corresponding eigenvector of $S_{\lambda=0}$ is $(\begin{smallmatrix} 1 \\ 0 \end{smallmatrix})$, so

$$P_{S_0}(\begin{smallmatrix} -\alpha \\ \mathbf{c} \end{smallmatrix}) = x(\begin{smallmatrix} 1 \\ 0 \end{smallmatrix}) \tag{215}$$

with a coefficient $x$ that can be found, for example, from the relation

$$A_{\lambda=0}^t(\begin{smallmatrix} -\alpha \\ \mathbf{c} \end{smallmatrix}) = x(\begin{smallmatrix} 1 \\ 0 \end{smallmatrix}) + o(1), \quad t \to \infty. \tag{216}$$

We have

$$A_{\lambda=0}^t = \begin{pmatrix} 1 & \mathbf{b}^T \sum_{s=0}^{t-1} D^s \\ 0 & D^t \end{pmatrix}, \tag{217}$$

so

$$x = \lim_{t \to \infty} \left( -\alpha + \mathbf{b}^T \sum_{s=0}^{t-1} D^s \mathbf{c} \right) = -\alpha_{\text{eff}}. \tag{218}$$

It follows that

$$\langle (\begin{smallmatrix} 1 \\ \mathbf{a} \end{smallmatrix})(\begin{smallmatrix} 1 \\ \mathbf{a} \end{smallmatrix})^T, P_\lambda\big[(\begin{smallmatrix} -\alpha \\ \mathbf{c} \end{smallmatrix})(\begin{smallmatrix} -\alpha \\ \mathbf{c} \end{smallmatrix})^T\big] \rangle = \alpha_{\text{eff}}^2 + o(1), \quad \lambda \to 0 \tag{219}$$

and hence

$$U_t' = (1 + o(1))\frac{\tau_1}{|B|}\alpha_{\text{eff}}^2 \sum_{k:\lambda_k < \lambda_*} \lambda_k^2 \mu_{A,\lambda_k}^{t-1} + O(r^t), \quad t \to \infty. \tag{220}$$

Applying the power law spectral assumption on $\lambda_k$ and computing the standard asymptotic, we finally get

$$U_t', U_t = (1 + o(1))\frac{(\alpha_{\text{eff}}\Lambda)^{1/\nu}\tau_1\Gamma(2 - 1/\nu)}{|B|\nu}(2t)^{1/\nu - 2}. \tag{221}$$

## J  PROOF OF THEOREM 6

1. Strict stability of $D$ is equivalent to

$$0 < \delta < 2. \tag{222}$$

By Theorem 4, $\mu_{A,\lambda} = 1 - 2\alpha_{\text{eff}}\lambda + O(\lambda)$, so

$$\alpha_{\text{eff}} \equiv -\frac{q_0 + q_1}{\delta} \geq 0 \tag{223}$$

is a necessary condition of strict stability of $S_\lambda$ at small $\lambda > 0$. We can exclude the case of equality $\alpha_{\text{eff}} = 0$ : in this case $q_0 = -q_1$ and so $\chi(1, \lambda) = \det(1 - S_\lambda) = 0$ for all $\lambda$, i.e. 1 is an eigenvalue of $S_\lambda$ for all $\lambda$.

Another necessary condition of strict stability is that the product of both eigenvalues of $S_\lambda$ is less than 1 in absolute value, i.e. $|p_0 + \lambda q_0| < 1$. This holds for all $0 \leq \lambda \leq \Lambda$ iff this holds for $\lambda = 0$ and $\lambda = \lambda_{\max}$. The condition for $\lambda = 0$ is the already established condition $0 < \delta < 2$, while the condition for $\lambda_{\max}$ means $-1 < p_0 + \lambda q_0 < 1$, i.e.

$$-\frac{\delta}{\lambda_{\max}} < q_0 < \frac{2 - \delta}{\lambda_{\max}}. \tag{224}$$

Under condition (224), stability can only fail to hold if $S_\lambda$ has real eigenvalues. At the lowest point $\lambda$ of this failure, one of the eigenvalues of $S_\lambda$ must be 1 or $-1$. Since $\chi(1, \lambda) = -\lambda(q_0 + q_1)$, 1

cannot be an eigenvalue of $S_\lambda$ for $\lambda > 0$ if $\alpha_{\text{eff}} > 0$. On the other hand the condition $\chi(-1, \lambda) = 0$ reads

$$4 - 2\delta + \lambda(q_1 - q_0) = 0. \tag{225}$$

At $\lambda = 0$ the l.h.s. is positive, so $S_\lambda$ does not have the eigenvalue $-1$ for $0 \le \lambda \le \lambda_{\max}$ iff the l.h.s. is also positive at $\lambda = \lambda_{\max}$, i.e.

$$\delta\alpha_{\text{eff}} - 2q_0 = q_0 - q_1 < \frac{4 - 2\delta}{\lambda_{\max}}. \tag{226}$$

Note that adding this inequality to $\alpha_{\text{eff}} > 0$ gives the right inequality (224), so we can drop the latter.

Summarizing, the full set of necessary and sufficient conditions of strict stability is

$$0 < \delta < 2, \quad \alpha_{\text{eff}} > 0, \quad -q_0 < \frac{\delta}{\lambda_{\max}}, \quad \delta\alpha_{\text{eff}} < \frac{4 - 2\delta}{\lambda_{\max}} - 2q_0. \tag{227}$$

2. The characteristic equation $\det(x - S_\lambda) = 0$ is

$$(x - 1)(x - 1 + \delta) - \lambda(q_1 x + q_0) = 0. \tag{228}$$

a) Suppose that $q_1 \le -\alpha_{\text{eff}}$. From the condition $\alpha_{\text{eff}} > 0$ we have $q_0 + q_1 < 0$. We also have $q_1 \le -\alpha_{\text{eff}} < 0$. It follows that

$$-\frac{q_0}{q_1} < 1. \tag{229}$$

On the other hand, $q_1 \le -\alpha_{\text{eff}}$ means $q_1 \le \frac{q_0 + q_1}{\delta}$ or, equivalently,

$$1 - \delta \le -\frac{q_0}{q_1}. \tag{230}$$

Inequalities (229), (230) show that the root $-\frac{q_0}{q_1}$ of the linear $\lambda$-term $\lambda(q_1 x + q_0)$ lies between the roots $1, 1 - \delta$ of the quadratic term $(x - 1)(x - 1 + \delta)$. It follows that at all $\lambda \in \mathbb{R}$ the matrix $S_\lambda$ has two real eigenvalues (possibly coinciding if $-\frac{q_0}{q_1} = 1 - \delta$).

b) Now suppose that $q_1 > -\alpha_{\text{eff}}$. Suppose first that $q_1 > 0$. Then

$$-\frac{q_0}{q_1} > 1, \tag{231}$$

i.e. the root of the term $\lambda(q_1 x + q_0)$ lies to the right of both roots $1, 1 - \delta$. Then, since $q_1 + q_0 < 0$, the polynomial $\chi(x, \lambda)$ has two different real $x$ roots at sufficiently small $\lambda > 0$. Then, as $\lambda$ is increased, the roots become not real and complex conjugate, and then again become real, at even larger $\lambda$.

To find the geometric position of the roots, note that they satisfy the condition

$$\frac{(x - 1)(x - 1 + \delta)}{q_1 x + q_0} \in \mathbb{R}. \tag{232}$$

This can be written as

$$hJ\left(\frac{q_1 x + q_0}{h}\right) \in \mathbb{R}, \tag{233}$$

where

$$h = \sqrt{(q_0 + q_1)(q_0 + q_1 - \delta q_1)} = \delta\sqrt{\alpha_{\text{eff}}(\alpha_{\text{eff}} + q_1)} \tag{234}$$

and $J(z) = z + \frac{1}{z}$ is the Joukowsky function. Since $J^{-1}(\mathbb{R}) = \mathbb{R} \setminus \{0\} \cup \{z \in \mathbb{C} : |z| = 1\}$, the roots $x$ either are real or belong to the circle $|q_1 x + q_0| = h$, as claimed.

If $q_1 > -\alpha_{\text{eff}}$ but $q_1 < 0$, we have

$$-\frac{q_0}{q_1} < 1 - \delta, \tag{235}$$

i.e. the root of $q_1 x + q_0$ is to the left of both roots $1, 1 - \delta$. The same pattern as above then applies – at small $\lambda > 0$ there are two real eigenvalues, then complex eigenvalues, then again real, and the non-real eigenvalues again lie on the circle $|q_1 x + q_0| = h$.

In the special case $q_1 = 0$ the root $-\frac{q_0}{q_1}$ becomes infinite. For sufficiently small $\lambda = 0$ the eigenvalues are real, and then for all larger $\lambda$ become complex. The circle $|q_1 x + q_0| = h$ degenerates into the straight line $\Re x = 1 - \frac{\delta}{2}$.

3. We can compute the value $R_\lambda$ using the identity

$$R_\lambda \equiv \sum_{t=0}^{\infty} \langle (\begin{smallmatrix} 1 \\ a \end{smallmatrix})(\begin{smallmatrix} 1 \\ a \end{smallmatrix})^T, A_\lambda^t (\begin{smallmatrix} -\alpha \\ c \end{smallmatrix})(\begin{smallmatrix} -\alpha \\ c \end{smallmatrix})^T \rangle = \langle (\begin{smallmatrix} 1 \\ a \end{smallmatrix})(\begin{smallmatrix} 1 \\ a \end{smallmatrix})^T, (1 - A_\lambda)^{-1}(\begin{smallmatrix} -\alpha \\ c \end{smallmatrix})(\begin{smallmatrix} -\alpha \\ c \end{smallmatrix})^T \rangle. \tag{236}$$

Here, $A_\lambda$ can be viewed as a linear operator on $\mathbb{R}^{2\times2}$ or, simpler, on the space of symmetric matrices $\mathbb{R}^{2\times2}_{\text{sym}} \cong \mathbb{R}^3$. The expression $(1 - A_\lambda)^{-1}(\begin{smallmatrix} -\alpha \\ c \end{smallmatrix})(\begin{smallmatrix} -\alpha \\ c \end{smallmatrix})^T$ can be found by symbolically solving the $3 \times 3$ linear system $(1 - A_\lambda)Z = (\begin{smallmatrix} -\alpha \\ c \end{smallmatrix})(\begin{smallmatrix} -\alpha \\ c \end{smallmatrix})^T$ for $Z \in \mathbb{R}^{2\times2}_{\text{sym}}$. We perform this computation in Sympy (Meurer et al., 2017) and obtain the reported result.

## K  PROOF OF THEOREM 7

**Part 1.**  Note that the theorem assumption $\lambda_k < 1$ implies that for some $\varepsilon > 0$ all the eigenvalues $\lambda_k \leq 1 - \varepsilon$. This means that the algorithm parameters never approach the boundary of strict stability of $S_\lambda$, given by theorem 4 at $\lambda_{\max} = 1$. This observation will be useful throughout the proof.

We start with showing the necessity of the first two conditions $\delta \to 0$ and $q_0 > 0$ that are agnostic to spectrum $\lambda_k$. As mentioned in the main text the necessity of $\delta \to 0$ to have $\alpha_{\text{eff}} \to \infty$ follows directly from strict stability bound $\alpha_{\text{eff}} < \frac{4}{\delta}$. Next, we observe that at fixed $\delta$, pairs of strictly stable $(\alpha_{\text{eff}}, q_0)$ are given by $q_0 > -\delta$ and $0 < \alpha_{\text{eff}} < \frac{4-2\delta}{\delta} - \frac{2q_0}{\delta}$. These inequalities define a right-angle triangle on $(\alpha_{\text{eff}}, q_0)$ plane with vertices at $\{(\frac{4}{\delta}, -\delta),\ (0, -\delta),\ (0, 2-\delta)\}$, which can serve as a useful geometric picture of relevant $(\alpha_{\text{eff}}, q_0)$ values.

The condition $q_0 > 0$ will follow from noisy stability condition $\sum_k \lambda_k^2 R_{\lambda_k} = O(1)$. Hence, we bound different terms appearing in $R_\lambda$, taking into account $\lambda_k \leq 1 - \varepsilon$, as

$$2 - 2\delta < \left[ \lambda q_0 + 2 - \delta \right] < 4 - 2\delta,$$
$$\varepsilon(4 - 2\delta) < \left[ 4 - 2\delta - \lambda(2q_0 + \delta\alpha_{\text{eff}}) \right] < 4. \tag{237}$$

As $\delta \to 0$, all the bounds above can be replaced by some global $\delta$-independent values. Then, we can simplify $R_\lambda$ up to a multiplicative constant as

$$R_\lambda \asymp \frac{\lambda q_0^2 + \delta\alpha_{\text{eff}}}{\lambda(\delta + \lambda q_0)}, \tag{238}$$

reproducing the first part of eq. (26). Observe that for $q_0 \leq 0$ we have $\delta + \lambda q_0 \geq \delta$, and, therefore, $R_\lambda \geq c\frac{\alpha_{\text{eff}}}{\lambda}$ with some $c > 0$. Then, for the stability sum we would have $\sum_k \lambda_k^2 R_{\lambda_k} \geq c\alpha_{\text{eff}} \sum_k \lambda_k \to \infty$, since $\alpha_{\text{eff}} \to \infty$. This settles impossibility of $q_0 \leq 0$ for noisy stability.

Now, we proceed to the last two stability conditions condition, assuming $q_0 > 0$. Then, both nominator and denominator in (238) have only positive terms, and we can write them as $\delta + \lambda q_0 \asymp \max(\delta, \lambda q_0)$ and $\lambda q_0^2 + \delta\alpha_{\text{eff}} \asymp \max(\lambda q_0^2, \delta\alpha_{\text{eff}})$. The characteristic spectral scales that separate two choices in these maximums are $\lambda_1^{\text{cr}} = \frac{\delta}{q_0}$ and $\lambda_2^{\text{cr}} = \frac{\delta\alpha_{\text{eff}}}{q_0^2}$, as given in the main text. Note that we asymptotically have $\lambda_1^{\text{cr}} < \lambda_2^{\text{cr}}$ since $\alpha_{\text{eff}} \to \infty$ while $q_0$ is bounded. This gives the second part of (26)

$$R_\lambda \asymp \begin{cases} \frac{\alpha_{\text{eff}}}{\lambda}, & 0 < \lambda < \lambda_1^{\text{cr}} \\ \frac{\delta\alpha_{\text{eff}}}{\lambda^2 q_0}, & \lambda_1^{\text{cr}} < \lambda < \lambda_2^{\text{cr}} \\ \frac{q_0}{\lambda}, & \lambda_2^{\text{cr}} < \lambda < \lambda_{\max} - \varepsilon. \end{cases} \tag{239}$$

Substituting the above into the stability sum leads to the condition

$$\sum_k \lambda_k^2 R_{\lambda_k} \asymp \alpha_{\text{eff}} \Big[ \sum_{k:\lambda_k \leq \frac{\delta}{q_0}} \lambda_k \Big] + \frac{\delta\alpha_{\text{eff}}}{q_0} \Big[ \sum_{k:\frac{\delta}{q_0} < \lambda_k < \frac{\delta\alpha_{\text{eff}}}{q_0^2}} 1 \Big] + q_0 \Big[ \sum_{k:\lambda_k \geq \frac{\delta\alpha_{\text{eff}}}{q_0^2}} \lambda_k \Big] = O(1) \tag{240}$$

Here, the third term is always bounded since $\sum_k \lambda_k < \infty$ and $q_0$ is also bounded. The boundness of the first and second term corresponds to the third and forth stability conditions of theorem 7, respectively. From the argumentation presented above, it is clear that those conditions are both necessary and sufficient.

**Part 2.** In fact, we can consider even weaker power-law spectrum assumption $\lambda_k \asymp k^{-\nu}$, while in the theorem statement we opted for spectral condition (1) in the sake of main text consistency. Assuming $\frac{\delta}{q_0} \to 0$, the sums in (240) can be estimated with a standard power-law arguments in terms of characteristic spectral index $k_1^{\mathrm{cr}} \asymp (\frac{\delta}{q_0})^{-\frac{1}{\nu}}$ defined by $\lambda_{k_1^{\mathrm{cr}}} = \frac{\delta}{q_0}$

$$\Big[\sum_{k:\lambda_k \le \frac{\delta}{q_0}} \lambda_k\Big] \asymp \Big(\frac{\delta}{q_0}\Big)^{-\frac{1-\nu}{\nu}}, \qquad \Big[\sum_{k:\frac{\delta}{q_0} < \lambda_k < \frac{\delta\alpha_{\mathrm{eff}}}{q_0^2}} 1\Big] \asymp \Big(\frac{\delta}{q_0}\Big)^{-\frac{1}{\nu}}. \tag{241}$$

Then, both the third and fourth stability conditions become $\alpha_{\mathrm{eff}}\delta^{-\frac{1-\nu}{\nu}} q_0^{\frac{1-\nu}{\nu}} = O(1)$. Substituting the ansatz $\alpha_{\mathrm{eff}} = \delta^{-h}$ and $q_0 = \delta^g$, we get $\delta^{-h-(1-g)\frac{1-\nu}{\nu}} = O(1)$, which is equivalent to $h \le (1-g)(1-\frac{1}{\nu})$ given in the statement of the theorem. As for the case $g \ge 1$, it breaks the previous assumption $\frac{\delta}{q_0} \to 0$. This brings us back to $q_0 \le 0$ case, where the first sum in (240) is estimated as $\Omega(\alpha_{\mathrm{eff}})$, leading to the divergence of $\sum_k \lambda_k^2 R_{\lambda_k}$. Finally, when $h < h_{\max}$ and $g > 0$, we can see that all three sums in (240) converge to zero, which translates to $U'_\Sigma$ at any constant batch size $|B|$.

## L  CONVERGENCE OF NON-STATIONARY ALGORITHM WITH POWER-LAW SCHEDULE

In this section, we derive the convergence rate of signal propagator $V_t$ given by (17) for the power-law ansatz schedule $(\delta_t, \alpha_{\mathrm{eff},t}, q_{0,t}) \sim (t^{-\overline{\delta}}, t^{\overline{\alpha}}, const)$.

**Adiabatic approximation.** For a non-stationary algorithm, evolution operators $A_{t,\lambda}$ essentially depend on $t$, making the analysis of the respective products $A_{\lambda,t}A_{\lambda,t-1}\ldots A_{\lambda,0}$ very challenging. To overcome this, we rely on the fact that in the considered schedule $\delta_t = t^{-\overline{\delta}}, \alpha_{\mathrm{eff},t} = t^{\overline{\alpha}}$, the parameters are changing very slowly at large $t$. Specifically, we assume that a significant change of eigenvectors of $A_{\lambda,t}$ also requires a significant relative change of algorithm parameters (e.g. 2 times increase/decrease). To achieve this change on a time window $(t, t+\Delta t)$ for any power-law relation $f_t = t^{-F}$, a large window size $\Delta t \sim t$ is required. Let $\mu_1, \mu_2, \mu_3, \mu_4$ be the eigenvalues of $A_{\lambda_t}$, ordered as $|\mu_1| > |\mu_2| \ge |\mu_3| \ge |\mu_4|$, and $\mathbf{S} = (\mathbf{s_1}\ldots\mathbf{s_4})$ be the matrix of respective eigenvectors. Then, if we manage to pick window size such that $\Delta t \ll t$ and $|\mu_2|^{\Delta t} \ll |\mu_1|^{\Delta t} \lesssim 1$, for any starting vector $\mathbf{r} \in \mathbb{R}^2 \otimes \mathbb{R}^2$ we have

$$\prod_{i=t}^{t+\Delta t-1} A_{\lambda,i}\mathbf{r} \approx \big(A_{\lambda,t}\big)^{\Delta t}\mathbf{r} = \sum_{e=1}^{4} \mathbf{s}_e(\mu_e)^{\Delta t}\big(\mathbf{S}^{-1}\mathbf{r}\big)_e \approx \mathbf{s}_1(\mu_1)^{\Delta t}\big(\mathbf{S}^{-1}\mathbf{r}\big)_1. \tag{242}$$

Thus, evolving starting vector $\mathbf{r}$ with $A_{\lambda,t}$ aligns the result with eigenvector $\mathbf{s}_1$ corresponding to the slowest eigenvalue $\mu_1$ of $A_{\lambda,t}$. The magnitude of the evolution result is given, up to initial projection $\big(\mathbf{S}^{-1}\mathbf{r}\big)_1$, by the product of slowest eigenvalue $\mu_1$. Extending this argument to many consecutive time windows $(t_j, t_{j+1})$, with $\Delta t_i = t_{i+j} - t_j$ growing with $t_i$, we naturally arrive to the following *adiabatic approximation*[1]

$$\prod_{i=1}^{t} A_{\lambda,i}\mathbf{r} \asymp \Big(\prod_{i=1}^{t} \mu_{1,i}\Big)\mathbf{s}_{1,t}, \quad t \to \infty \tag{243}$$

---

[1] our usage of this approximation has many parallels with adiabatic approximation in quantum mechanics wiki/Adiabatic theorem

We will use this approximation to compute asymptotic of $V_t$. Applying (243) to a single term $V_{\lambda,t}$ in $V_t$ sum gives

$$V_{\lambda,T} \equiv \langle \begin{pmatrix} 1 \\ \mathbf{a} \end{pmatrix} \begin{pmatrix} 1 \\ \mathbf{a} \end{pmatrix}^T, A_{T,\lambda} A_{T-1,\lambda} \ldots A_{1,\lambda} \begin{pmatrix} 1 & 0 \\ 0 & 0 \end{pmatrix} \rangle \asymp \mu_{1,T} \mu_{1,T-1} \ldots \mu_{1,1}$$

$$\asymp \exp \left[ \int_1^T \log \mu_1(t) dt \right] \equiv V_{\lambda,T}^{\mathrm{adi}}. \tag{244}$$

Here in the last transition, we used that our schedule can be smoothly interpolated to non-integer $t$. Then, replacing the sum by the integral can change the result by at most a constant, as soon as $\varepsilon < |\mu_{1,t}| < 1$ for all $t$.

**Evolution eigenvalues.** Before proceeding to the calculation of the signal propagator as $V_T \asymp \sum_k c_k^2 \lambda_k V_{\lambda_k,T}^{\mathrm{adi}}$, let us write down the required properties of eigenvalues of $A_\lambda$. First, we remind that, as in sec. 6, we consider $\tau_2 = 0$, which leads to factorizable evolution operator $A_\lambda = S_\lambda \otimes S_\lambda$ with eigenvalues equal to products of eigenvalues of $S_\lambda$. With a slight abuse of notation, in the rest of the section we will denote $\mu_1, \mu_2$ the eigenvalues of $S_\lambda$. These eigenvalues are given by zeros of characteristic equation $\chi(x,\lambda) = 0$ (see (36)), which in variables $(\delta, \alpha_{\mathrm{eff}}, q_0)$ is written as

$$x^2 - x\big(2 - \delta - \lambda(\delta\alpha_{\mathrm{eff}} + q_0)\big) + 1 - \delta - \lambda q_0 = 0. \tag{245}$$

The respective roots are

$$\mu_{1,2} = 1 - \frac{\delta + \lambda(\delta\alpha_{\mathrm{eff}} + q_0) \pm \sqrt{E}}{2}, \quad E = (\delta + \lambda(\delta\alpha_{\mathrm{eff}} + q_0))^2 - 4\lambda\delta\alpha_{\mathrm{eff}} \tag{246}$$

Let us denote $\widetilde{\lambda}_{1,2}^{\mathrm{cr}}$ the solutions of $E(\lambda) = 0$, as they separate the spectrum into regions either real and complex eigenvalues $\mu_{1,2}$. In the case $\delta \ll \delta\alpha_{\mathrm{eff}} \ll q_0 \sim 1$, they are given by $\widetilde{\lambda}_1^{\mathrm{cr}} \approx \frac{\delta}{4\alpha_{\mathrm{eff}}}$ and $\widetilde{\lambda}_2^{\mathrm{cr}} \approx \frac{4\delta\alpha_{\mathrm{eff}}}{q_0^2}$. These critical spectral scales, although connected, should not be confused with the other scales $\lambda_{1,2}^{\mathrm{cr}}$ appearing in (26) and separating different regions of the stability sum. In particular, we have $\widetilde{\lambda}_2^{\mathrm{cr}} \asymp \lambda_2^{\mathrm{cr}}$ but $\widetilde{\lambda}_1^{\mathrm{cr}} \ll \lambda_1^{\mathrm{cr}}$, suggesting that behaviour of the stability sum changes in the middle of complex region $\lambda \in (\widetilde{\lambda}_1^{\mathrm{cr}}, \widetilde{\lambda}_2^{\mathrm{cr}})$.

In the left real region $\lambda < \widetilde{\lambda}_1^{\mathrm{cr}}$ evolution eigenvalues can be approximated for $\lambda \ll \widetilde{\lambda}_1^{\mathrm{cr}}$ as $\mu_1 \approx 1 - \alpha_{\mathrm{eff}}\lambda$ and $\mu_2 \approx 1 - \delta$, while at the border $\mu_1(\widetilde{\lambda}_1^{\mathrm{cr}}) = \mu_2(\widetilde{\lambda}_1^{\mathrm{cr}}) \approx 1 - \frac{\delta}{2}$ (see also fig. 3 and theorem 4).

As the schedule parameters $(\delta_t, \alpha_{\mathrm{eff},t}, q_{0,t})$ changes with time $t$ so will the spectral scales $\widetilde{\lambda}_{1,2}^{\mathrm{cr}}(t)$ and the evaluation eigenvalues $\mu_{1,2}(\lambda, t)$ within the respective spectral regions. The specific behavior of this change is central to our computation of the signal propagator asymptotic below.

**Signal propagator asymptotic.** Now, we come back to signal propagator $V_T$ and its adiabatic approximation $V_{\lambda,T}^{\mathrm{adi}}$. Let us fix some $\lambda \ll 1$ and increase time $T$. At first, the chosen eigenvalue $\lambda$ will be very small compared to $\widetilde{\lambda}_1^{\mathrm{cr}}(t)$. But as the schedule advances, it will enter the complex region at a transition time $t_*(\lambda) : \widetilde{\lambda}_1^{\mathrm{cr}}(t_*) = \lambda$. By substituting $\delta_t, \alpha_{\mathrm{eff},t}$ from our schedule, we get $t_*(\lambda) \approx (4\lambda)^{-\frac{1}{\alpha+\delta}}$. This time signalizes the end of the adiabatic approximation applicability period: the latter requires $|\mu_1| > |\mu_2|$ but at $t = t_*(\lambda)$ we have $\mu_1 = \mu_2$ since the roots of a quadratic polynomial merge at the border between complex and real regions.

At times $T < t_*(\lambda)$, we can use $\mu_1(\lambda, t) \approx 1 - \alpha_{\mathrm{eff}}(t)\lambda$ and estimate the propagator in adiabatic approximation as (recall that $\mu_1(A_\lambda) = \big(\mu_1(S_\lambda)\big)^2$, giving extra factor of 2)

$$V_{\lambda,T}^{\mathrm{adi}} = \exp \left[ \int_1^T 2\log\mu_1(t) dt \right] \approx \exp \left[ -\int_1^T 2\lambda t^{\overline{\alpha}} dt \right] \approx \exp \left[ -\frac{2}{1+\overline{\alpha}} \lambda T^{1+\overline{\alpha}} \right] \tag{247}$$

Let us look at the convergence reached by the end of adiabatic approximation applicability time $t_*(\lambda) \approx (4\lambda)^{-\frac{1}{\overline{\alpha}+\overline{\delta}}}$. Substituting this value into $V_{\lambda,T}^{\mathrm{adi}}$ leads to

$$V_{\lambda,t_*(\lambda)}^{\mathrm{adi}} \approx \exp \left[ -C_{V^{\mathrm{adi}}} \lambda^{-\frac{1-\overline{\delta}}{\overline{\alpha}+\overline{\delta}}} \right], \quad C_{V^{\mathrm{adi}}} = \frac{2^{-\frac{2+\overline{\alpha}-\overline{\delta}}{\overline{\alpha}+\overline{\delta}}}}{1+\overline{\alpha}}. \tag{248}$$

We see that, for $\overline{\delta} < 1$, the propagator at $t = t_*(\lambda)$ has the form $e^{-c_1 \lambda^{-c_2}}$ with some $c_2, c_1 > 0$. Thus, as $\lambda \to 0$, we have exponential convergence of the propagator $V_{\lambda, t_*(\lambda)}^{\mathrm{adi}} = e^{-c_1 \lambda^{-c_2}} \to 0$. Moreover, after adiabatic approximation applicability time $T = t_*(\lambda)$ the propagator cannot increase back, because we have strictly stable $S_\lambda$ with $|\mu_{1,2}| < 1$ at all times.

If we now combine the contribution to the propagator from all eigenvalues $V_T = \sum_k c_k^2 \lambda_k V_{\lambda_k, T}$ at some large $T$, we can neglect the terms that are exponentially small in $T$ (like $e^{-c_3 T^{c_4}}, c_3, c_4 > 0$) compared to polynomially small terms of the size $T^{-c_5}, c_5 > 0$. The neglected terms include $V_{\lambda_k, T}$ for large eigenvalues $\lambda_k \sim 1$ since large eigenvalues converge exponentially fast with $T$, regardless of whether adiabatic approximation is used. They also include $V_{\lambda_k, T}$ for eigenvalues that have already left adiabatic approximation applicability region $T > t_*(\lambda_k)$. For all such terms we can, in fact, formally use the adiabatic expression (247) because it will also be exponentially converged.

Summarizing, in the limit $T \to \infty$ we can use the expression (247) at all $T$ and $\lambda_k$ and correctly capture the leading contributions into $V_T$. The resulting asymptotic is

$$
\begin{aligned}
V_T &\asymp \sum_k c_k^2 \lambda_k V_{\lambda_k, T}^{\mathrm{adi}} \approx \int_0^{\lambda_{\max}} Q \zeta \lambda^{\zeta-1} V_{\lambda, T}^{\mathrm{adi}} d\lambda \\
&\approx \int_0^{\lambda_{\max}} Q \zeta \lambda^{\zeta-1} e^{-\frac{2}{1+\overline{\alpha}} \lambda T^{1+\overline{\alpha}}} d\lambda \approx Q \Gamma(\zeta+1) \left( \frac{2}{1+\overline{\alpha}} \right)^{-\zeta} T^{-\zeta(1+\overline{\alpha})}.
\end{aligned}
\tag{249}
$$

Here we used continuous approximation for the sum over $\lambda_k$, according to (1). The integral in the second line is computed by reducing it to Gamma function integral with linear change of variables $z = \frac{2}{1+\overline{\alpha}} \lambda T^{1+\overline{\alpha}}$.

Finally, let's comment on the allowed values of momentum exponent $\overline{\delta}$. The derivation above requires $\overline{\delta} < 1$, otherwise, $V_{\lambda, t_*(\lambda)}^{\mathrm{adi}}$ will not be exponentially converged according to (248) and adiabatic approximation will not correctly capture the leading contributions to $V_T$. However, the Jacobi schedule, which is provably optimal in the noiseless setting described by $V_T$, uses $\overline{\delta} = \overline{\alpha} = 1$. This suggests that the requirement $\overline{\delta} < 1$ is reasonable and approaches the optimal scaling of momentum term $\delta$. Yet, we need a method more accurate than adiabatic approximation to treat $\overline{\delta} = 1$ case.

## M  EXPERIMENTS

In all the experiments we consider Memory-1 algorithms with $a = 0$. There are many equivalent ways to implement a Memory-1 algorithm. In practice, we found it the most convenient to use a minimal modification of standard *SGD with momentum* algorithm (referred to as HB in this work) implemented in modern deep learning frameworks[2]

$$
\mathbf{u}_{t+1} = \beta \mathbf{u}_t - \nabla L(\mathbf{w}_t), \qquad
\begin{cases}
\text{HB :} & \mathbf{w}_{t+1} = \mathbf{w}_t + \alpha \mathbf{u}_{t+1} \\
\text{Memory-1 :} & \mathbf{w}_{t+1} = \mathbf{w}_t + \alpha_2 \mathbf{u}_{t+1} - \alpha_1 \nabla L(\mathbf{w}_t),
\end{cases}
\tag{250}
$$

This way, one can use an update step of standard SGD optimizer class with learning rate $\alpha = \alpha_2$, and then manually subtract parameter gradients with new learning rate $\alpha_1$

The new learning rates in parametrization (250) are expressed in terms of our main notations as $\beta = 1 - \delta$, $\alpha_1 = \frac{q_0}{\beta}$, $\alpha_2 = \delta(\alpha_{\mathrm{eff}} - \alpha_1)$. Then, the accelerated region $\delta^{-1} \gg \alpha_{\mathrm{eff}} \gg q_0 = const$ translates to $\delta \ll \alpha_2 \ll \alpha_1 = const$.

This gives another perspective on the accelerated AM1 algorithm we identified in sec. 6. To achieve acceleration, we still want, as in HB, to make the velocity vector more inertive (i.e. making the ball "heavier"). However, while the optimally accelerated HB adds this velocity with $\alpha = O(1)$ learning rate, we add it with vanishing $\alpha_2 = O(\delta \alpha_{\mathrm{eff}}) \to 0$ learning rate but compensate with an extra "kick" term $\nabla L(\mathbf{w}_t)$ with constant $\alpha_1 = O(1)$ learning rate. Apparently, this combination leads both acceleration and bounded effect of mini-batch noise.

Next, we discuss our two experimental setups.

---

[2]See, for example, pytorch/SGD

## M.1 GAUSSIAN DATA

For Gaussian data experiments, we generate inputs $\mathbf{x} \sim \mathcal{N}(0, \Lambda)$ from Gaussian distribution with diagonal covariance $\Lambda = \mathrm{diag}(\lambda_1, \lambda_2, \dots \lambda_M)$, leading to diagonal Hessian $\mathbf{H} = \Lambda$. For the optimal parameters, we simply take the vector of target coefficients $\mathbf{w}_* = (c_1, c_2, \dots, c_M)$. Then, we set ideal power-laws for $\lambda_k = k^{-\nu}$ and $c_k^2 = k^{-\kappa-1}$ which satisfy our asymptotic power-law conditions (1) with $\zeta = \frac{\kappa}{\nu}$. This way, we create a simple setting where all the loss curves are expected to have a pronounced power-law shape, and we will be able to directly compare experimental convergence exponents with theoretically predicted ones. We set $M = 10^4$ to ensure that truncation of the spectrum has negligible finite size effects and the experiments are well described by the asymptotic regime $t, k \to \infty$.

For the experiments in fig. 1 and fig. 4, we pick spectral exponents $\zeta = 0.5, \nu = 3$. For the AM1 schedule exponent we use $\overline{\delta} = 0.95, \overline{\alpha} = \overline{\delta}(1 - \frac{1}{\nu})$, at the edge of allowed AM1 exponents values. In both figures 1 and 4, we compare empirical loss trajectories with power-law curves (dashed lines) with predicted convergence rate, which show a good agreement between each other. In fig. 4 we try more $\overline{\delta}, \overline{\alpha}$ values, and find good agreement of the experimental loss rates with predicted $L_t = O(t^{-\zeta(1+\overline{\alpha})})$.

## M.2 MNIST

We design the MNIST experiment to be the opposite of the Gaussian data setting to test our results in more general scenario beyond quadratic problems with asymptotically power-law spectrum. Specifically, we apply a single-hidden layer neural network to a flattened MNIST images $\mathbf{x} \in \mathbb{R}^{768}$:

$$f(\mathbf{W}_1, \mathbf{W}_2, \mathbf{b}, \mathbf{x}) = \frac{1}{\sqrt{n}} \mathbf{W}_2 \, \mathrm{ReLU}(\mathbf{W}_1 \mathbf{x} - \mathbf{b}) \in \mathbb{R}^{10}, \tag{251}$$

where $\mathbf{W}_1 \in \mathbb{R}^{n \times 768}$, $\mathbf{b}, \mathbf{x} \in \mathbb{R}^{768}$, $\mathbf{W}_2 \in \mathbb{R}^{10 \times n}$, with model width taken as $n = 1000$. Then, we consider one hot encoded vectors $y = (0, \dots, 1, \dots, 0) \in \mathbb{R}^1 0$ as quadratic loss $L_B = \frac{1}{2b} \sum_{i=1}^{b} \|f(\mathbf{x}_i) - \mathbf{y}_i\|^2$ as an empirical loss function to be minimized.

The above setting, although much simpler than modern deep-learning problems, provides an example of non-convex optimization function. In particular, if one linearizes $f(\mathbf{W}_1, \mathbf{W}_2, \mathbf{b}, \mathbf{x})$ with respect to parameters on the current optimization step, the resulting quadratic approximation to the original loss function will evolve during and the respective spectral distributions $\lambda_k, c_k$ are expected to deform significantly from the roughly power-law shapes with $\nu \approx 1.33$ and $\kappa = \zeta\nu \approx 0.33$ reported in Velikanov & Yarotsky (2021) for the limit $n \to \infty$.

Yet, on figure1 we can still see that both Jacobi-scheduled HB and AM1 display qualitatively similar behavior to that of quadratic problems with power-law spectrum, though with slightly different exponents than can be expected from $\nu \approx 1.33$, $\kappa \approx 0.33$ estimation. Most importantly, Jacobi-scheduled HB retains its instability in stochastic settings, while AM1 is stable at least for extremely long $t = 10^5$-steps duration of the experiment.

AM1 algorithm in fig.1 uses $\overline{\delta} = 1$ and $\overline{\alpha} = 0.5$. This chosen effective learning rate exponent is, in fact, larger than the maximal stable exponent $\overline{\alpha}_{\max} = \overline{\delta}(1 - \frac{1}{\nu}) \approx 0.25$ that could be estimated with MNIST eigenvalue exponent $\nu \approx 1.33$. Yet, we don't see any signs of divergence on the whole training duration, but which were observed on Gaussian data experiments in fig. 4. That might suggest that non-quadratic models could exhibit different stability conditions due to some recovery mechanism, with the catapult effect Lewkowycz et al. (2020) being one example of such recovery. We provide MNIST loss trajectories for extra values of Batch size and algorithm parameters in fig. 5.

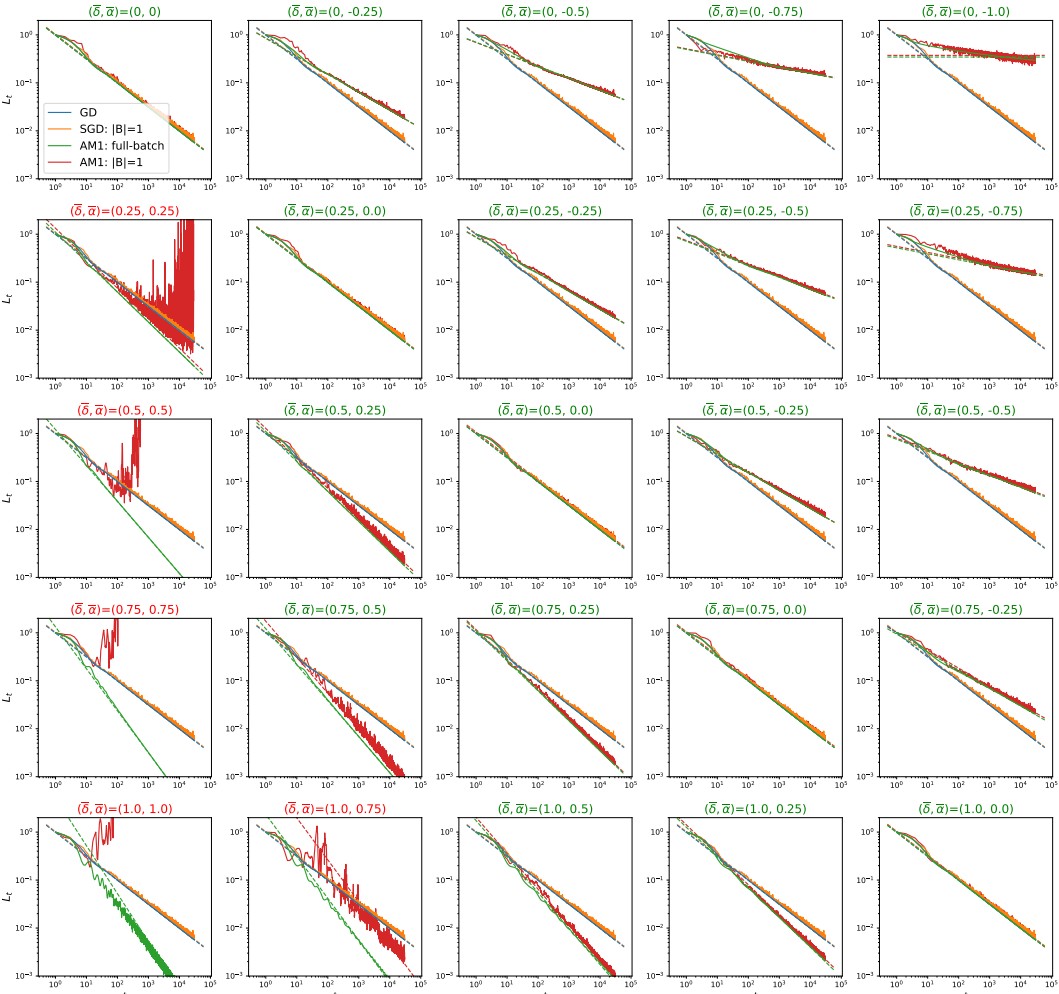

Figure 4: Comparison of plain GD and AM1 on a grid of $\overline{\delta}, \overline{\alpha}$. We run both algorithms in full-batch setting and mini-batch with $|B| = 1$, leading to 4 trajectories on each subplot. Also, we choose $\alpha = 0.1$ for both algorithms, and $\alpha_{\text{eff},t} = 0.1t^{\overline{\alpha}}$ to avoid any non-asymptotic divergence. For each couple $\overline{\delta}, \overline{\alpha}$, we color in green the subplot titles if stability condition $\overline{\alpha} \leq \overline{\delta}(1 - \frac{1}{\nu})$ holds, and in red otherwise. We observe that all empirically diverged trajectories are predicted correctly by this stability condition, while a single trajectory $(\overline{\delta}, \overline{\alpha}) = (1.0, 0.75)$ is expected to diverge but demonstrate only large but bounded noise fluctuations. Finally, along each experimental trajectory, we plot an exact power-law (dashed) line with rate: $t^{-\zeta}$ for GD and $t^{-\zeta(1+\overline{\alpha})}$ for AM1. Again, we observe very good agreement between theoretical prediction and empirical rates, validating adiabatic approximation of sec. L. Moreover, the rate $t^{-\zeta(1+\overline{\alpha})}$ remains valid even for the case $\overline{\delta} = 1$ not covered by adiabatic approximation.

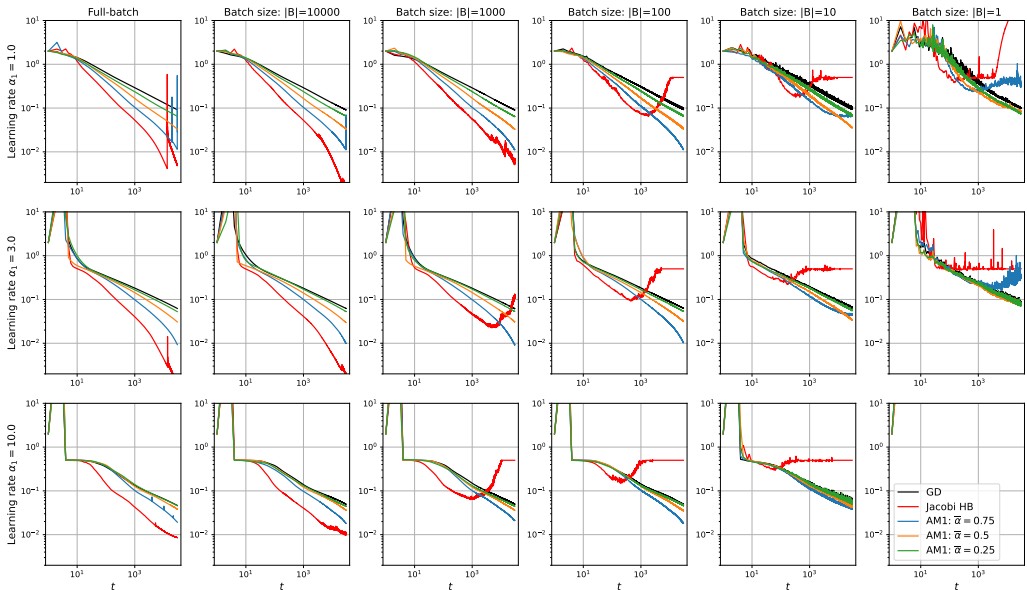

Figure 5: Trajectories of plain GD, Jacobi scheduled HB and AM1 for different learning rates and batch sizes. For each subplot, learning rate $\alpha$ of GD and Jacobi-scheduled HB is equal to "kick" learning rate $\alpha_1$ of AM1. Also, Jacobi-scheduled HB and AM1 have exactly the same values of momentum $\beta_t = 1 - \frac{2}{t}$ on each iteration. We observe that AM1 is indeed much more stable than Jacobi-scheduled HB for a range of learning rates and batch sizes. Moreover, it is stable for larger values of $\overline{\alpha}$ than estimated maximal value $\overline{\alpha}_{\max} \approx 0.25$. Another interesting observation is that the divergent trajectories of Jacobi scheduled HB, unlike for pure quadratic problems, only get stuck around the loss of random prediction instead of diverging to infinity.

