# OpenReview forum: "SGD with memory: fundamental properties and stochastic acceleration"
_ICLR.cc/2025/Conference — ICLR 2025 Poster_

### Official Review · Reviewer_eHN7 · 2024-11-01

**Soundness:** 4
**Presentation:** 2
**Contribution:** 3
**Rating:** 8
**Confidence:** 2

**Summary:**

In this work, the authors address the open problem of theoretical feasible acceleration of mini-batch SGD-type algorithms (i.e. in the stochastic setting) on infinite-dimensional quadratic problems, particularly with power-law spectrum.
The authors introduce memory-M algorithms with M auxiliary velocity vectors and show the equivalence to the multistep recurrence formulation if the loss $L$ is quadratic and hyperparameters of the algorithm are stationary.
By expanding the loss in terms of so-called signal and noise propagators, the authors show that the loss convergence $L_t \sim C_l t^{-\xi}$ of stationary stable memory-M algorithms retain the exponent $\xi$ of plain GD, but can have different constants $C_L$. Finally the authors prove that memory-1 algorithms can make $C_L$ arbitrarily small while maintaining stability. The theoretical results are empirically verified on synthetic Gaussian data and MNIST using a shallow ReLU network.

**Strengths:**

- The authors are able to (heuristically) derive an accelerated and stable rate of convergence in their algorithm AM1 and verify the rate empirically. (However I am not too familiar with the field to judge the novelty of this contribution.)
- The derivation of the stability of stationary stable memory-M algorithms seems rigorous and it is interesting to see that stationary stable algorithms cannot achieve a faster rate than plain GD even using arbitrary numbers $M$ auxiliary vectors.
- The analysis of the convergence constant $C_L$ and the possibility of making it arbitrarily small for stationary stable memory-1 algorithms and how it relates to the effective learning rate $\alpha_{eff}$ was also insightful.

**Weaknesses:**

- The paper seemed quite convoluted with many different symbols, equations and indices due to its mathematic rigor and it could benefit from restructuring it such that it becomes more accessible.
- The empirical validation of the results remains quite limited with only experiments on synthetic Gaussian data and MNIST. Particularly, the spectrally expressible approximation holds exactly for gaussian data and approximately for MNIST classification. It would be interesting to see also experiments where this approximation is less exact.
- It is seems like the optimal rate depends on the right choice of hyperparameters, such as $\alpha_0$, $q_0$ and $\delta$. However it is unclear to me how to select or estimate them in practice. This limits its application to real-world problems.

**Questions:**

1. I do not fully understand the image on the right of Figure 2. What are $t_1, ..., t_4$ supposed to denote or perhaps more generally what does it mean that a signal/noise propagator is "long" or "short"?
2. I have another question regarding the stability of the maps $A_\lambda$: If I understood it correctly, $\lambda$ corresponds to the eigenvalues of the matrix $\mathbf{H}$ (line 201). Thus, it seems to me that we have to assume $\lambda > 0$ in general. Does Theorem 3 then imply that one can only have strict stability of $A_{\lambda}$ if $\lambda_{\max}$ is sufficiently small?
3. What are the values of $\xi_V$ and $\xi_U$ respectively in Eq. (21) and (22)?
4. Have the authors also considered non-quadratic settings and how does it perform here?

---

> ### Author Response · Authors · 2024-11-27
>
> Thank you for your careful reading and positive evaluation of our work!
>
> > The paper seemed quite convoluted...
>
> That's true; we have rewritten one part that was especially convoluted, please see our general response.
>
> > The empirical validation of the results remains quite limited ...
>
> We agree that in our empirical validations we covered a limited set of scenarios. We have validated that the heuristically derived AM1 algorithm actually gives the predicted convergence rate while being stable in numerical simulations, and in the MNIST experiments, we showed that AM1 still works in non-convex setting without exact power-law spectrum and on real-world data. See also a detailed summary of our experiments in the response to reviewer LKb1.
>
> A comprehensive test of AM1 in a variety of modern deep-learning scenarios would be an interesting direction for future research. Yet, we are convinced that the provided set of experiments fits well the current manuscript. We would like to keep the focus of this submission on theoretical analysis and derivation of the algorithm, limiting experiments to validation of main theoretical claims and providing proof-of-concept evidence that the algorithm can work beyond the purely quadratic setting considered in the paper.
>
> > It is seems like the optimal rate depends on the right choice of hyperparameters, such as $\alpha_0$, $q_0$ and $\delta$. However it is unclear to me how to select or estimate them in practice. This limits its application to real-world problems.
>
> Thank you for this remark! We tried to address this question in the paper, although not very directly.
>
> - Although theorems 6 and 7 provide specific choice of $\delta, q_0, \alpha_\mathrm{eff}$ hyperparameters, it is natural to assume that the precise values will change if we deviate from the setting of quadratic models with exact power-law spectrum. Yet, we believe that the region of hyperparamters $\delta^{-1}\gg \alpha_\mathrm{eff}\gg q_0=const$, identified in theorem 7, will be more robust to the change of the setting (non-power law spectrum, non-quadratic loss, etc). See the remark in the penultimate paragraph on page 9.
>
> - Another possible difficulty for practical application is formulation of the algorithm in terms of variables $\delta, q_0, \alpha_\mathrm{eff}$. These variables appear to be the most convenient parametrization for our theoretical analysis,  but not necessarily for the practical implementation of the algorithm. We address this in the beginning of section M, where we rewrite the AM1 update rule as a modification of HB (a.k.a SGD in most deep learning libraries). Specifically, we can just add can an extra ``kick'' term - updating parameters directly with the gradients of the current iteration. Denoting the learning rate of the new kick term as $\alpha_1$, and the learning rate of the classical momentum term as $\alpha_2$, the AM1 schedule becomes $(\delta_t, \alpha_{1,t}, \alpha_{2,t})\sim (t^{-\overline{\delta}}, const, t^{-\overline{\delta} + \overline{\alpha}})$.
>
> - Hyperparameter choice. One can parameterized the whole schedule as $(\delta_t, \alpha_{1,t}, \alpha_{2,t}) = (c_\delta t^{-\overline{\delta}}, c_{\alpha_1}, c_{\alpha_2}t^{-\overline{\delta} + \overline{\alpha}})$ leading to 5 tunable hyperparameters. A quick starting point could be (i) setting reasonable values for constants $c_\delta,c_{\alpha_1},c_{\alpha_2}$ (ii) picking $\overline{\delta}$ close to its theoretically predicted value $\overline{\delta}=1$ to avoid some edge effects, e.g. $\overline{\delta}=0.9$ (iii) doing a sweep on value of $\overline{\alpha}$, since its theoretical optimal prediction $\overline{\alpha} = \overline{\delta}(1-\frac{1}{\nu})$ requires knowledge of the spectral exponent $\nu$ which is typically unknown for real-world problems. We roughly followed this strategy in our experiments, as shown on figures 4,5.
>
> - Finally, we note that there is freedom in choosing schedule exponents $\overline{\delta},\overline{\alpha}$, removing the need to precisely estimate spectrum exponent $\nu$. From our results in sec. 6, the algorithm would be stable as soon as $\overline{\alpha}\leq \overline{\delta}(1-\frac{1}{\nu})$, and acceleration is determined by the value of $\overline{\alpha}$. Thus, a less aggressive choice than theoretically optimal $\overline{\delta}=1, \overline{\alpha}=1-\frac{1}{\nu}$ would still provide accelerated convergence while being stable.

---

> > ### Author Response · Authors · 2024-11-27
> >
> > > I do not fully understand the image on the right of Figure 2. What are $t_1,\ldots,t_4$ supposed to denote or perhaps more generally what does it mean that a signal/noise propagator is "long" or "short"?
> >
> > The meaning of times $t_1,\ldots,t_m$ is central to our propagator expansion. We tried to make this part clearer in the revised version and also added a dedicated section D in the appendix. In the proof of new Theorem 2 (loss expansion), the loss at time $T$ appears by considering the second moments update at each step as the sum of a diagonal operator $A_t$ (mainly, the noiseless part of the evolution) and a rank 1 operator $P_t$ (coming from the $\mathbf{H} \operatorname{Tr}[\mathbf{H}\mathbf{C}]$ term of the minibatch noise variance in eq. (15)). Then, the times $t_1,\ldots,t_m$ correspond to those iterations where we chose $P_t$ when expanding the product $(A_T+P_T)(A_{T-1}+P_{T-1})\ldots(A_1+P_1)$. Taking into account all the ways to expand the parentheses translates into the sum over all propagator configurations in eq. (19).
> >
> > By "long"/"short" we simply mean a propagator $V_t$ or $U_t$ with a large or small $t$. More precisely, when considering the propagator expansion for $L_T$, "long" means length $T-o(T)$ and "short" means length $o(T)$. By inspecting the proof of Theorem 3, only those configurations of propagator factors that include one long propagator an several short ones (as shown in Figure 3) make the leading contribution to the loss expansion (19) in the signal- and noise-dominated convergence regimes. This is the key intuition behind the proofs of the loss asymptotics for these regimes in Theorem 3.
> >
> > > I have another question regarding the stability of the maps $A_\lambda$ ... Does Theorem 3 then imply that one can only have strict stability of $A_\lambda$ if $\lambda_{\max}$ is sufficiently small?
> >
> > No, $\lambda_{\max}$ does not need to be small for $A_\lambda$ to be stable throughout $0<\lambda<\lambda_{\max}$. Theorem 3 (4 after revision) characterizes stability of $A_\lambda$ only for small $\lambda>0$ and does not say anything about larger $\lambda$, where the statement $\mu_{A,\lambda}=1-2\alpha_{\mathrm{eff}}\lambda+O(\lambda^2)$ becomes uninformative.
> >
> > For example, as can be seen in Figure 3, both in Heavy Ball and our memory-1 methods, at some critical value $\lambda$ the leading eigenvalue $\mu_{A,\lambda}$ collides with the second eigenvalue, after which they both start moving along a circle in the complex plane. In the examples shown in Figure 3, all the eigenvalues $\mu_{\lambda}$ of $A_\lambda, 0<\lambda\le \lambda_{\max},$ stay within the unit circle and so $A_\lambda$ are stable for all $0<\lambda\le \lambda_{\max}$. However, the regime associated with the eigenvalues on the circle is not described by Theorem 3 (4).
> >
> > At the same time, in the case of memory-1 algorithms, we do give a complete characterization of stability for all $0<\lambda\le\lambda_{\max}$ (and in particular covering all regimes) in Theorem 6, part 1. This characterization is necessarily significantly more complicated than the condition $\alpha_{\mathrm{eff}}>0$ from Theorem 3 (4).
> >
> > > What are the values of $\xi_v,\xi_U$ in Eq. (21) and (22)?
> >
> > Theorem 3 (2 in the old version) is a combinatorial result that assumes that the propagators are asymptotically power-law, and derives the loss asymptotics based only on the propagator expansion (19) of the loss. So, in Theorem 3 the values $\xi_v,\xi_U$ can be any numbers such that $\xi_U>1, \xi_V>0$. But then, in Theorem 5 we show that for the particular propagators given by Eqs. (17),(18) we have $\xi_U=2-1/\nu, \xi_V=\zeta$. We separate the two theorems because we feel that such structure of the exposition clarifies the origin of the loss asymptotics $L_t\sim t^{-\zeta}$ and $L_t\sim t^{1/\nu-2}$ in the signal- and noise-dominated regimes.
> >
> > > Have the authors also considered non-quadratic settings and how does it perform here?
> >
> > This is a good question. Empirically, we observe our algorithm to work on slightly non-quadratic problems: note that in our MNIST experiments (see Figure 1, right, and Figure 5) the network is finite and so the loss is non-quadratic. We observe both acceleration and stability to be present in these experiments.
> >
> > Theoretically, it is a good future research direction, and there are some interesting challenges. In particular, our work relies heavily on spectral source and capacity conditions: a reasonable extension to non-quadratic problems would need to propose their non-quadratic generalizations while demonstrating that these can be fulfilled in realistic problems.

---

> > > ### Comment · Reviewer_eHN7 · 2024-11-28
> > >
> > > Thank you for your elaborate answers and explanations!
> > >
> > > I particularly appreciated the elaborate explanation about the choice of hyperparameters, which I believe could also be beneficial to the reader in the final version. Perhaps the authors can consider adding a short section on this in the appendix.
> > >
> > > Given that my concerns and questions have been addressed, I have adjusted my score and would suggest the acceptance of this paper to ICLR 2025.

---

### Official Review · Reviewer_LKb1 · 2024-11-03

**Soundness:** 1
**Presentation:** 1
**Contribution:** 3
**Rating:** 3
**Confidence:** 2

**Summary:**

This paper explores how to make stochastic gradient descent (SGD) faster on certain types of optimization problems, specifically those that involve a noisy mini-batch setting. The authors examine how adding "memory" (extra helper variables) to SGD can help. They introduce "memory-$M$" algorithms, where $M$ is the number of extra velocity vectors added. Through mathematical analysis, they show that these memory-$M$ algorithms can achieve better performance by improving certain constants that affect convergence speed, even though they don’t change the basic rate of convergence.

The authors also propose a new "memory-1" algorithm, which has only one extra velocity vector and uses a special time-based schedule for learning. This algorithm is shown to improve performance both theoretically and in experiments.

**Strengths:**

The work is a definite interest to read, and the results seem to hold a lot of promise.

**Weaknesses:**

- Introduction: The reader who is not an expert in this field will find this introduction difficult to understand. I recommend that it be significantly reworked using various welcome tricks, such as tables. In general, the text is difficult to read....

- I don't fully understand about the experimental part. Is it just not there?

**Questions:**

Can authors convince me that their work fits the spirit of this conference? This question is not obvious to me at this point.....

---

> ### Author Response · Authors · 2024-11-27
>
> Thank you for your feedback!
>
> > I don't fully understand about the experimental part. Is it just not there?
>
> While the paper primarily focuses on theoretical analysis, we do provide experiments to validate the derived results, as  presented in figures 1, 4, 5. All the settings are described in the dedicated section M of the appendix. In short, we believe that our current set of experiments covers the following points:
>
> - Figure 1 demonstrates the divergence of Jacobi accelerated HB that was mentioned in the introduction. Importantly, the divergence of Jacobi accelerated HB happens at any batch size, with larger batch size values just delaying the moment of divergence. This serves as a motivation to look into accelerated strategies, leading us to the formulation and analysis of Memory-M algorithms.
>
> - Same figure 1 shows that the proposed AM1 achieves an accelerated power-low convergence rate, although not as high as that of Jacobi accelerated HB in the noiseless setting. Moreover, the AM1 algorithm is stable even at quite small batch sizes ($|B|=1$ and $|B|=10$ in the figure).
>
> - Figure 1 also provides the noisy and noiseless (full-batch) loss curves in otherwise identical settings. This visually illustrates the effect of noise on the loss trajectory in both stable and diverging cases.
>
> - In terms of scope, we consider either synthetic Gaussian data with the manually set power-law spectrum and MNIST classification with a shallow ReLU network. We describe the details of each setting in section M. These two settings serve different goals.
>
> - The Gaussian setting is directly described by the developed theory and was expected to follow our theoretical predictions. Yet, the last step of designing a schedule for AM1 algorithm and obtaining its convergence rate is heuristic and non-rigorous, in contrast to the rest of the paper. Thus it was important for us to validate the intuition used in the heuristic derivation of AM1. Indeed, the predicted convergence rate, depicted by dashed lines, matches the experimental curves.
>
> - The MNIST setting, although basic by modern deep learning standards, is non-convex and thus deviates from the quadratic optimization problem analyzed in the paper. Yet, we see that the main properties of AM1, stability and accelerated convergence, survive deviation from purely quadratic problems. This gives us some hope that AM1 algorithm could have broad practical applications in real-world optimization problems.
>
> - Finally, in figures 4 and 5 we provide more experiments by changing various parameters of both Gaussian and MNIST settings. These experiments show that the conclusions from figure 1 are robust.
>
> > Can authors convince me that their work fits the spirit of this conference? This question is not obvious to me at this point...
>
> We are trying to solve a problem -acceleration of SGD - that is relatively hard and does not seem to have a simple intuitive solution. To this end, we develop and present an original theoretical approach that includes many new elements (general memory-$M$ algorithms, a propagator expansion of the loss, generalized effective learning rate,  stable memory-1 algorithms with unbounded effective learning rate ...) and eventually reveals a solution. We believe our work to advance the ML science and thus fit the spirit of ICLR quite well.
>
> Many of the works directly connected to our research and cited in our list of refernces have also been published either at ICLR or similar conferences like NeurIPS, ICML, COLT. This further confirms that our work fits the spirit of ICLR well.

---

### Official Review · Reviewer_1Va8 · 2024-11-05

**Soundness:** 4
**Presentation:** 2
**Contribution:** 4
**Rating:** 8
**Confidence:** 3

**Summary:**

This paper introduces a framework for analyzing iterative optimization methods, referred to as memory-$M$ algorithms, which allow the gradient to be evaluated at shifted points relative to $w_t$ additionally a shift term is added to the gradient when doing the update. The authors explore both stationary (with fixed update parameters) and non-stationary versions of these algorithms. They establish that memory-$M$ algorithms can achieve the optimal convergence rate for data with a power-law spectrum, even under mini-batch noise. Notably, they propose a memory-1 algorithm with a time-dependent power-law schedule, which provides a heuristic pathway to accelerated convergence, supported by empirical results.


**Justification for the score**

While I think the result is important in establishing convergence results for a large family of methods, I think the presentation of the paper could be significantly improved.

**Strengths:**

The paper's biggest strength is the establishment of strong convergence rates for iterative methods. Mini-batched algorithms are difficult to analyze; hence, establishing the convergence rate for regression in the noiseless regime is interesting.

Additionally, the method allows for controlling the constant and shows that it can also be arbitrarily improved.

I think both of these are important contributions.

In addition, the paper introduces a general framework for analyzing such methods. i think this is an important contribution as well.

**Weaknesses:**

The writing of the paper is a major weakness. It is very dense notationally, not necessarily conceptually. This makes it quite challenging to build intuition as one reads the paper. One possible suggestion is that the paper develops the results initially in the $a = 0$ as the shifts are not needed for the main contributions and then provide extensions, possibly in the appendix. The other is to hide some of the intermediate steps to provide space for more intuition for the different objects.

One thing that threw me off is using capital letters for the propagators as these are scalars.

**Questions:**

1. Could such a framework be extended to non-IID batching, that is, batching with replacement (recent work [1,2])?

2. Could you say anything about the generalization error using [3]?

3. Could we add label noise or regularization?

[1] Lok, Jackie, Rishi Sonthalia, and Elizaveta Rebrova. "Discrete error dynamics of mini-batch gradient descent for least squares regression." arXiv preprint arXiv:2406.03696 (2024).

[2] Beneventano, Pierfrancesco. "On the trajectories of sgd without replacement." arXiv preprint arXiv:2312.16143 (2023).

[3]  Wang, Y., Sonthalia, R. &amp; Hu, W.. (2024). Near-Interpolators: Rapid Norm Growth and the Trade-Off between Interpolation and Generalization. <i>Proceedings of The 27th International Conference on Artificial Intelligence and Statistics</i>, in <i>Proceedings of Machine Learning Research</i> 238:4483-4491 Available from https://proceedings.mlr.press/v238/wang24l.html.

---

> ### Author Response · Authors · 2024-11-27
>
> Thanks very much for your careful reading and a very positive evaluation of our work!
>
> > One possible suggestion is that the paper develops the results initially in the $a=0$  as the shifts are not needed for the main contributions and then provide extensions, possibly in the appendix. The other is to hide some of the intermediate steps to provide space for more intuition for the different objects.
>
> Thank you for these suggestions - we seriously thought about them, especially removing the shifts. We didn't get to implement it, because of the lack of time and because we wanted to keep some discussion of shifts in the main text, but we plan to return to this issue in a future update. However, we have rewritten the excessively dense "loss expansion" section, moving intermediate steps to the appendix as you suggested (see our general response).
>
> > One thing that threw me off is using capital letters for the propagators as these are scalars.
>
> Thank you for pointing this out. We now mention explicitly in the text that the propagators are scalar, to minimize possible confusion. There are several reasons why we use capital letters for the propagators: 1) they denote direct contribution to the averaged loss $L_t$ which is also consistently denoted by a capital letter (e.g., in the noiseless case $L_t=\tfrac{1}{2}V_t$); 2) propagators $U_t$ are extensively treated as entries of a circulant matrix $\mathbb U$ in various derivations in the appendix; 3) propagators are important objects, and most lowercase letters are already used for other purposes in the paper.
>
> > Could such a framework be extended to non-IID batching, that is, batching with replacement (recent work [1,2])?
>
> We expect that yes, it could be extended in this way. Reading these papers, analysis of batching with replacement is certainly technically more involved than of batching without it. But at least for large training sets and small batches, we can expect both kinds of batching to be close, suggesting similar properties of our generalized SGD.
>
> > Could you say anything about the generalization error using [3]?
>
> The paper [3] derives a precise asymptotic characterization of the test error under power-law spectrum. Yet, there seem to be several challenges left to obtain (and then analyze) such characterization for mini-batch gradient-based  algorithms, including memory-M algorithms:
>
> - The test error in [3] is derived for kernel ridge regression (which is especially convenient for tools from random matrix theory thanks to its connection to the resolvent), but not for gradient descent. The case of gradient descent is, however, addressed to some extent in some other papers, e.g. [4,5].
>
> - Regarding the stochasticity of inputs, in [3] and other related works the training set is randomly drawn once before applying the learning algorithm. The SGD setting adds one more layer of stochasticity due to the random choice of mini-batches from the finite-size training set. The joint averaging over these two sources of randomness might be challenging.
>
> - If we want to characterize the generalization error under a label noise, this also needs to be added to the framework (see another question below for more details).
>
> On a positive side, though, the SE approximation used in our framework holds exactly for Gaussian features used in [3].
>
> [4] Yicheng Li, Weiye Gan, Zuoqiang Shi, and Qian Lin. Generalization error curves for analytic spectral algorithms under power-law decay, 2024. URL https://arxiv.org/abs/2401.01599.
>
> [5] M. Velikanov, M. Panov, and D. Yarotsky. Generalization error of spectral algorithms. In ICLR 2024. https://openreview.net/forum?id=3SJE1WLB4M.

---

> ### Author Response · Authors · 2024-11-27
>
> > Could we add label noise or regularization?
>
> Both label noise and regularization could be introduced in our framework by generalizing some of its parts. Basically, we need to modify second moments update (12),(13), and then add new respective terms into propagator expansion (19).
>
> In case of additive i.i.d. label noise with variance $\sigma^2$, second moments update acquires a term (one can use derivation in sec. C.1 as an example)
> $$
> \frac{\sigma^2}{|B|} \mathbf{H} \otimes \begin{pmatrix} -\alpha_t & \mathbf c_t  \end{pmatrix}^T \begin{pmatrix} -\alpha_t & \mathbf c_t \end{pmatrix}.
> $$
> The case of $L_2$ regularization is a bit more involved with more terms changed/added. Since it pulls the parameters to $0$, the optimum will shift to a new location $\mathbf{w}^*_{reg}$, modifying both the main evolution term $A_t$ and the noise term $P_t$ that make up the evolution operator $F_t$ (see section D).
>
> However, the biggest changes would be in the loss propagator expansion. In its current version, we heavily rely on the multiplicative structure of the evolution operator, its form  $F_t=A_t+P_t$, and $P_t$ being rank-1. Adding extra terms of a different type to $F_t$, or introducing additive terms into the second moment's update as in the case of label noise, would require redesigning the loss expansion. This is an interesting direction for future research.

---

### Official Review · Reviewer_kpRS · 2024-11-06

**Soundness:** 4
**Presentation:** 3
**Contribution:** 3
**Rating:** 8
**Confidence:** 2

**Summary:**

This paper establishes a theoretical framework for the so-called memory-M algorithms - first order algorithms that are augmented with storage for M vectors in the gradient space, and uses a linear combination of them to compute the gradients and modify weight updates. The usefulness of this framework is that it is a natural generalization of Heavy Ball and a number of other first order algorithms. The work provides an equivalence of this formulation with recursive multi-step first-order algorithms. The work also provides a generalized loss decomposition into signal and noise propagations. The paper also provides a detailed description of convergence properties of memory-M (and focusing on memory-1) algorithms - including the main parameters controlling this convergence and an algorithm providing equivalent convergence rates for the SGD case.

**Strengths:**

Overall, this is a very good paper. I think it is very methodical in its development of the memory-M framework - starting with existing issues of accelerated HB for SGD, moving to the description of the generalized framework for GD and SGD, loss decomposition and improved algorithm and convergence properties. The topic coverage is very comprehensive, covering the expected questions raising with the development of the generalized framework. There are clearly a lot of efforts made towards the balance between rigor and readability - with moving a lot of the technical details into the appendix (e.g. the phrasing of Theorem 2). The contributions are novel to the field of stochastic optimization

**Weaknesses:**

An expected consequence of the comprehensiveness of the paper is its density - making it relatively hard to read. As mentioned before, there was clearly a lot of effort put towards reorganization and moving details into the appendix - but I wonder if more could be done in that direction. For example, the Loss expansion section with the introduction of propagators seemed too dense - and was lacking intuitions that would facilitate further reading of the paper, since those concepts were heavily relied on for further developments in the paper. Those intuitions were present in the earlier sections and were thus helpful for speeding up understanding of the paper

**Questions:**

As mentioned, I wonder if there are any ways of improving the readability of the technical portions with further addition of intuitions and heuristics. This wouldn’t change the technical contributions of the paper but would make the paper more accessible for the readers and thus facilitate the propagation of ideas in it.

---

> ### Author Response · Authors · 2024-11-27
>
> Thank you very much for your careful reading and a very positive evaluation of our work!
>
> > ...the Loss expansion section with the introduction of propagators seemed too dense - and was lacking intuitions that would facilitate further reading of the paper..
> ... I wonder if there are any ways of improving the readability of the technical portions ...
>
> Thank you for this feedback and particularly pointing out the loss expansion section as too dense - we fully agree and have rewritten this piece, moving technical details to a new appendix. We plan to return to improving the readability of the paper in the future update.

---

### Author Response · Authors · 2024-11-27
**General response to reviewers**

We sincerely thank the reviewers for their positive feedback and many useful suggestions.

All the reviewers point out that the paper is not easy to read, and some indicated the exposition of our loss expansion to be especially convoluted. We agree that the "Loss expansion" subsection in section 4 was excessively compressed. We have rewritten this subsection, emphasizing the intuition and general ideas. The technical derivations were moved to new Appendix D and now are written there with more detail. The loss expansion itself is formulated now as new Theorem 2. This has slightly shifted the numbering of formulas, theorems and sections in the paper; in our responses below we refer to the new numbers.

In addition, we have fixed a number of typos and clarified the exposition in several other places (in particular, the derivation of the convergence rate of the algorithm AM1 in section L).

---

### Meta-Review · Area_Chair_8jWf · 2024-12-22

**Metareview:**

This paper establishes a theoretical framework for the so-called memory-M algorithms - first order algorithms that are augmented with storage for M vectors in the gradient space, and uses a linear combination of them to compute the gradients and modify weight updates. The usefulness of this framework is that it is a natural generalization of Heavy Ball and a number of other first order algorithms. The work provides an equivalence of this formulation with recursive multi-step first-order algorithms. The work also provides a generalized loss decomposition into signal and noise propagations. The paper also provides a detailed description of convergence properties of memory-M (and focusing on memory-1) algorithms - including the main parameters controlling this convergence and an algorithm providing equivalent convergence rates for the SGD case.

- Overall, the work is methodical in its development of the memory-M framework - starting with existing issues of accelerated HB for SGD, moving to the description of the generalized framework for GD and SGD, loss decomposition and improved algorithm and convergence properties.
- The topic coverage is comprehensive.
- The contributions are novel to the field of stochastic optimization

However, the reviewers agreed that the paper is not very well written, mainly due to it being overly dense and technical. The empirical validation of the results is limited with only experiments on synthetic Gaussian data and MNIST; however, since this is a theoretical work, I am putting lower weight on this.

Three reviewers rated the paper with "8"; and one rated it as "3". However, this negative review did not offer substantial reasons for the criticism raised, and I did not see how the score 3 was justified. Hence, I put a lower weight on this review.

Overall, I am happy to recommend acceptance.

---

I wish to add that the authors should consider (I am not forcing them to) citing & discussing these works in relation to their results, which seem closely relevant, since they consider acceleration of SGD for quadratic problems:

1. Nicolas Loizou and Peter Richtarik. Momentum and stochastic momentum for stochastic gradient, Newton, proximal point and subspace descent methods, arXiv:1712.09677, 2017
2. Robert M. Gower, Filip Hanzely, Peter Richtarik and Sebastian Stich. Accelerated stochastic matrix inversion: general theory and speeding up BFGS rules for faster second-order optimization, NeurIPS 2018
3. Peter Richtarik and Martin Takac. Stochastic reformulations of linear systems: algorithms and convergence theory, arXiv:1706.01108, 2017

**Additional Comments On Reviewer Discussion:**

Several questions were asked, e.g., about extension to non-IID batching, hyper-parameters, experiments, and writing. I've read the replies and believe the answers were generally satisfactory.

---

### Decision · Program_Chairs · 2025-01-22

Accept (Poster)